# Connecting CSR theory and LPJmL 5 to assess the role of environmental conditions, management and functional diversity for grassland ecosystem functions

Stephen Björn Wirth[1,2], Arne Poyda[2], Friedhelm Taube[2], Britta Tietjen[3,4], Christoph Müller[1], Kirsten Thonicke[1], Anja Linstädter[5,6], Kai Behn[5,6], Sibyll Schaphoff[1], Werner von Bloh[1], and Susanne Rolinski[1]

[1]Potsdam Institute for Climate Impact Research (PIK), Member of the Leibniz Association, P.O. Box 60 12 03, 14412 Potsdam, Germany
[2]Institute of Crop Science and Plant Breeding, Grass and Forage Science/Organic Agriculture, Kiel University, Hermann-Rodewald-Str. 9, 24118, Kiel, Germany
[3]Freie Universität Berlin, Institute of Biology, Theoretical Ecology, Königin-Luise-Str. 2/4 Gartenhaus, 14195 Berlin, Germany
[4]Berlin-Brandenburg Institute of Advanced Biodiversity Research (BBIB), D-14195 Berlin, Germany
[5]University of Potsdam, Institute of Biochemistry and Biology, Potsdam, Germany
[6]Institute of Crop Science and Resource Conservation, University of Bonn, Bonn, Germany

**Correspondence:** Stephen Björn Wirth (stephen.wirth@pik-potsdam.de)

**Abstract.** Forage offtake, leaf biomass and soil organic carbon storage are important ecosystem services of permanent grasslands, which are determined by climatic conditions, management and functional diversity. However, functional diversity is not independent of climate and management, and it is important to understand the role of functional diversity and these dependencies for ecosystem services of permanent grasslands. Since functional diversity may play a key role in mediating impacts of changing conditions. Large-scale ecosystem models are used to assess ecosystem functions within a consistent framework for multiple climate and management scenarios. However, large-scale models of permanent grasslands rarely consider functional diversity. We implemented a representation of functional diversity based on the Competitor, stress-tolerator, ruderal (CSR) theory and the global spectrum of plant form and function into the Lund Potsdam Jena managed Land (LPJmL) dynamic global vegetation model (DGVM) forming LPJmL-CSR. Using a Bayesian calibration method, we parameterised new plant functional types (PFTs) and used these to assess forage offtake, leaf biomass, soil organic carbon storage and community composition of three permanent grassland sites. These are a temperate grassland and a hot and a cold steppe for which we simulated several management scenarios with different defoliation intensities and resource limitations. LPJmL-CSR captured the grassland dynamics well under observed conditions and showed improved results for forage offtake, leaf biomass and/or SOC compared to the original LPJmL 5 version at the three grassland sites. Furthermore, LPJmL-CSR was able to reproduce the trade-offs associated with the global spectrum of plant form and function and similar strategies emerged independent of the site-specific conditions (e.g. the C- and R-PFTs were more resource exploitative than S-PFTs). Under different resource limitations, we observed a shift of the community composition. At the hot steppe for example, irrigation led to a more balanced community composition with similar C-, S- and R-PFT shares of above-ground biomass. Our results show that LPJmL-CSR allows

for explicit analysis of the adaptation of grassland vegetation to changing conditions while explicitly considering functional
diversity. The implemented mechanisms and trade-offs are universally applicable paving the way for large-scale application.
Applying LPJmL-CSR for different climate change and functional diversity scenarios may generate a range of future grassland
productivity.

# 1   Introduction

Permanent grasslands deliver multiple ecosystem services, one of which is their role as a source of feed for livestock across the
globe (White et al., 2000). Another service is their soil organic carbon (SOC) storage which has the potential to contribute to
climate change mitigation (e.g. Godde et al., 2020; Yang et al., 2019). These two important ecosystem services depend on the
climatic conditions, soil properties, management and functional diversity. The climatic conditions and soil properties determine
the availability of important resources for photosynthesis and plant growth. While irrigation and fertiliser management are
applied to increase the availability of specific resources and thereby productivity, grazing or mowing remove biomass which
can affect leaf and root growth and SOC stocks (Bai and Cotrufo, 2022; Conant et al., 2017). Even though functional diversity
of the vegetation is not an independent factor but depends on environmental conditions (Fei et al., 2018; Grime, 2001) and
management (Guo, 2007), it also affects forage supply and SOC (Yang et al., 2019; Chen et al., 2018). Furthermore, functional
diversity plays an important role for the resistance and resilience of an ecosystem towards the impacts of changing conditions
and might be essential to maintain the ecosystem functioning and ecosystem service provision of permanent grasslands under
climate change (Isbell et al., 2015; Guuroh et al., 2018). Therefore, it is important to understand the role of functional diversity
in permanent grasslands and its role for ecosystem services such as the amount of biomass removed through mowing or grazing
(in the following referred to as forage offtake), above-ground biomass, and SOC storage.

## 1.1   The role of environmental conditions and management for grassland vegetation and SOC storage

Forage offtake, leaf biomass, SOC and plant community composition are dependent on environmental conditions and man-
agement. Important factors for plant growth are atmospheric $CO_2$ concentration, radiation, temperature, water and nutrient
supply. Atmospheric $CO_2$ constitutes the basic resource for photosynthesis and its rising concentration as well as rainfall pat-
terns can shift the competitive balance between C3 and C4 grassland species (Schimel et al., 2015; Xie et al., 2022). Provided
with sufficient water and nutrients, grasslands can produce large amounts of biomass, while drought and nutrient stress lead
to lower productivity. Since large amounts of biomass can lead to high carbon sequestration, this highlights the importance of
temperature and precipitation for SOC storage (Wiesmeier et al., 2019). High precipitation also favours the formation of SOC-
stabilising mineral surfaces (Doetterl et al., 2016; Chaplot et al., 2010) and affects decomposition rates (Meier and Leuschner,
2010). On the other hand, high temperatures can lead to an increase of microbial decomposition and a decrease in SOC stock
(e.g. Koven et al., 2017; Sleutel et al., 2007) if soil moisture levels are sufficient to permit the formation of active microbial
communities. Highest SOC stocks are generally found in cool humid climates but decrease towards warmer and drier climates
(Jobbágy and Jackson, 2000). Additionally, removal of above-ground biomass through grazing or mowing may be beneficial

for grassland productivity depending on its intensity (Ruppert et al., 2015) by removing moribund plant material and triggering growth (over-)compensation. However, mowing and grazing also affect the below-ground biomass and highly intensive management may lead to overgrazing and cause SOC loss (McSherry and Ritchie, 2013). Still, global meta-analyses of grazing effects on SOC did not find consistent trends (McSherry and Ritchie, 2013; Piñeiro et al., 2010).

Together, the environmental factors and the management act as filters for the plant functional types (PFTs) representative of species that are best suited for the specific conditions. Changes in management or climatic and soil conditions may alter this filtering process and lead to the selection of different strategies either indirectly through alterations of the resource limitations that can cause shifts in the competitive balance between functional types (e.g. Yu et al., 2015; Tilman and El Haddi, 1992) or directly in the case of management by manipulating the species pool through reseeding and weeding (Weisser et al., 2017) or

selective grazing (e.g. Wan et al., 2015).

## 1.2   Functional diversity and ecological strategies

Functionally diverse ecosystems contain species that follow different ecological strategies and can be described through a representation of these strategies. We define ecological strategy as the traits a plant or species uses to occupy a certain habitat. Plants have evolved a range of different ecological strategies that influence the performance of different species in different

habitats. Functional diversity which underpins robustness against environmental and management change of certain ecosystem functions is related to the presence or absence of specific strategies. For example, a community in which multiple strategies are present is less vulnerable to fluctuations or changes in environmental conditions or management (Buzhdygan et al., 2020). To distinguish between different ecological strategies, several classification schemes have been developed. The competitor, stress-tolerator, ruderal (CSR) theory (Grime, 2001; Campbell and Grime, 1992; Grime, 1977), distinguishes three main strategies:

Competitive (C), stress tolerant (S) and ruderal (R) strategies can be placed at the nodes of a triangle, while intermediate strategies are placed in between. This scheme can be used to classify the average strategy of a community (e.g. Caccianiga et al., 2006) as well as the strategies of single species (e.g. Grime, 1974). The main strategies are associated with different plant behaviour. C species are efficient resource users and grow fast but do not deal well with resource limitations or frequent disturbances. Opposite are S species which invest resources in more robust tissue which grows slower but enables them to

cope with resource limitations. While both C and S species are vulnerable towards disturbance, R species use periods between disturbances to complete their life cycle and have an advantage in disturbance-prone environments. This different behaviour is expressed through different trait values which in turn can be used to classify plants according to the CSR theory. A prominent example is the global spectrum of plant form and function which explains differences in ecosystem function using traits related to growth economics, stature and life cycle (Díaz et al., 2016) and has been combined with the CSR theory and applied to

single but also multi-species communities (Pierce et al., 2017, 2013). Additionally, several other CSR analysis methods have been developed (Hodgson et al., 1999; Grime et al., 1988) and applied to compare vegetation function (e.g. Schmidtlein et al., 2012; Hunt et al., 2004) and to assess various community processes (Pierce et al., 2017) for example resistance, resilience and coexistence (Lepš et al., 1982), succession (Caccianiga et al., 2006), and the biodiversity-productivity relationship (Cerabolini et al., 2016). Pierce et al. (2017) provided a method to classify and compare the CSR strategies of different vascular plants

at the global scale which is useful to assess community assembly in different environments. However, additional methods are needed to also predict ecosystem functioning and ecosystem service provision of the assembled communities.

## 1.3 Modelling ecosystem functions of permanent grasslands

To assess forage offtake or leaf biomass and SOC storage of permanent grasslands under different environmental conditions and management, models of grassland dynamics can be useful tools (e.g. Jebari et al., 2022; Chang et al., 2021; Rolinski et al., 2018). Models at the community and plot scale that incorporate very detailed approaches to simulate functional diversity in a specific context already exist (e.g. Schmid et al., 2021; May et al., 2009). In contrast, large-scale vegetation models generally use a very simple representation of the community and do not consider the trade-offs described by the global spectrum of plant form and function (Díaz et al., 2016) at all or only partially (e.g. Pfeiffer et al., 2019; Sakschewski et al., 2015). However, large-scale models provide the means to assess functional diversity in a wide range of environmental conditions and management interventions to improve projections of ecosystems functions under future climate change (e.g. Herzfeld et al., 2021; Sitch et al., 2008). In addition, such models could be useful to improve knowledge on the mechanisms underlying the *global spectrum of plant form and function* and help better distinguish local variability from large-scale patterns. To overcome current limitations of large-scale models, simplifications such as the CSR theory provide the opportunity to incorporate ecological strategies and functional diversity into large-scale models.

The dynamic global vegetation model (DGVM) LPJmL is able to simulate different grazing or mowing management (Rolinski et al., 2018), irrigation (Schaphoff et al., 2018), application of manure and synthetic fertiliser (von Bloh et al., 2018) and tillage (Lutz et al., 2019). The CSR strategies and their relationship to specific plant traits provide a simple way to incorporate functional diversity into the LPJmL model to include its effects in the assessment of forage offtake or leaf biomass and SOC storage of grasslands for different environmental conditions and management scenarios. To this end, we implemented the trade-off associated with the three main strategies of the CSR theory (Grime, 1977) for managed grasslands in LPJmL using the *global spectrum of plant form and function* (Díaz et al., 2016) to assess:

- how important functional diversity is for forage offtake or leaf biomass and SOC dynamics in different climates and under different management regimes.

- how changing resource limitations affect forage offtake or leaf biomass, SOC and community composition.

## 2 Methods

We conducted our assessment at three permanent grassland sites in different climates: a temperate meadow in northern Germany with favourable climatic conditions for grassland productivity, as well as a savanna rangeland in South Africa and a cold steppe pasture in Inner Mongolia (China) with less favourable climatic conditions. Throughout the remaining manuscript we refer to the sites as temperate grassland, cold and hot steppe respectively, following the Köppen-Geiger climate classification (Kottek

et al., 2006). At each site, we assessed two levels of management intensity which either differed with respect to the amount of fertiliser applied (temperate grassland) or the defoliation intensity (hot and cold steppe).

We extended LPJmL to account for trade-offs between C-, S- and R-plant species as described by the CSR theory (Grime, 1977) using functional traits. We used two strategy axes to distinguish these three strategies: First, we distinguished between acquisitive (C and R) and conservative (S) strategies using resource economics. Second, we used reproduction strategies and

stature to distinguish between plant species with large investments in reproduction but a small stature (R) from plant species with small investments in reproduction with a wide range of statures (C and S). Both strategy axes are expressed through several model parameters (Sect. 2.4).

To represent the different strategies, we parameterised three herbaceous PFTs - one competitive (C-PFT), one stress tolerant (S-PFT) and one ruderal (R-PFT) - for each site and management intensity (see sect. 2.3 for details). Strategies that are in

between these three main strategies (e.g. competitive ruderal or stress tolerant ruderal) were not reflected by additional PFTs but should be reflected in the fractional cover of the main strategy (e.g. if a competitive ruderal strategy is advantageous in an environment, this results in a higher share of the competitive and the ruderal PFT). We evaluated the new implementation in the following referred to as LPJmL-CSR against forage offtake or leaf biomass and SOC observations for the different sites.

## 2.1 Overview of managed grassland representations in LPJmL

We extended the LPJmL model version 5 (LPJmL 5), which already included the representation of managed grasslands using a daily allocation scheme (Schaphoff et al., 2018), four different management options (Rolinski et al., 2018) and the nitrogen cycle (von Bloh et al., 2018). In this model version the dynamics of a grassland were simulated using three herbaceous PFTs that do not distinguish between forbs and graminoids: one polar C3, one temperate C3 and one tropical C4 herbaceous PFT, which were constrained to the respective climatic regions by bio-climatic limits. Tree PFTs, which are also part of LPJmL,

were not allowed to establish on managed grasslands and all further descriptions provided here of or related to PFTs only concern herbaceous PFTs. As a consequence, all grasslands that are not located at the border between climatic regions were simulated using only one of these PFTs to represent herbaceous vegetation. In theory, however, the number of PFTs that could coexist within a grid cell is not limited. In LPJmL, each PFT represent an entire population of adult plants using the concept of average individuals. The PFT describes the carbon and nitrogen stocks of the leaves and roots of an average individual and

the number of average individuals in a population. It follows, that the carbon and nitrogen stocks of the population can be determined by multiplying the average individual stocks with the number of average individuals. Carbon and nitrogen stocks as well as the number of average individuals are dynamically calculated each day from the simulated processes which are: 1) establishment of new PFTs and reproduction of established PFTs (Sect. 2.3.3), 2) biomass accumulation calculated from gross primary production (GPP) and autotrophic respiration limited by environmental conditions, and 3) plant turnover. LPJmL

represents the response of the vegetation to temperature, water and nitrogen stress but disregards additional causes of stress such as other nutrient deficiencies, salt, heavy metals, ozone or UV radiation. At the core of the model is the representation of growth dynamics including the assimilation and allocation of new biomass through photosynthesis and turnover of senescent tissue. Each day, the GPP is calculated dependent on radiation, temperature, water and nitrogen limitations for each PFT.

Subsequently, NPP is computed by subtracting growth and maintenance respiration from GPP. In a third step, the assimilated
carbon is distributed between leaves and roots to approach the prescribed optimal leaf mass to root mass ratio. Finally, senescent
leaf and root tissue is transferred to the litter layer.

In LPJmL 5, each herbaceous PFT is represented by one average plant individual. The initial community composition is
not prescribed. Instead, upon initialisation, each PFT is established based on the PFT-specific establishment rate and offspring
biomass (Sect. 2.3.3 and 2.4.1). The community composition during each time step emerges from the competition for resources
dependent on the processes described above. Different management options are available for irrigation, fertilisation and grazing
or mowing. In this study, we use the mowing and the daily grazing option to determine forage offtake. While mowing removes
all biomass above a threshold of $50 \, \mathrm{gCm^{-2}}$, the forage offtake from daily grazing depends on the livestock units' feed demand
(details in Appendix A5 and Rolinski et al. 2018). The daily grazing option does not account for animal preferences (Rolinski
et al., 2018). Irrigation options used here are no irrigation (rainfed) or potential irrigation (no water limitation, (Jägermeyr
et al., 2015)). Manure fertilisation options were adapted from the crop module (see SI) and include the amount and timing of
manure application. Manure application can be split over several treatments. In grazed grasslands, 25% of the grazed carbon
(Rolinski et al., 2018) and 50% of the nitrogen are returned to the soil as dung or urine of the grazing animals (Huhtanen et al.,
2008).

## 2.2 Site description

We conducted our assessment at three different sites (Fig. SI 1) which are located in different biomes with substantial differ-
ences in precipitation and temperature, covering the warm temperate fully humid (temperate grassland), the arid hot steppe (hot
steppe) and the arid cold steppe climate (cold steppe) (Kottek et al., 2006) and are subject to different management intensities
(Table 1).

The temperate grassland is located in favourable climatic conditions and provides high forage supply. The vegetation is dom-
inated by C strategists with marginal shares of S and R. It is cut four times each year in May, July, August and September. Data
on two experiments were available: an unfertilised (N0) control and a fertilised (N1) treatment with $240 \, \mathrm{kgN \cdot ha^{-1} \cdot year^{-1}}$
in the form of cattle manure split over four applications at the beginning of the growing season and after the first three cuts
(Reinsch et al., 2018a, b).

Arid conditions lead to a lower forage supply for the hot steppe. S-strategists dominate the vegetation, while the R-strategy
is subordinate and the C-strategy is only marginally present. Data for an ungrazed (C0) control and a rotationally (C1) grazed
experiment with a livestock density of 0.1 cows per hectare with a body weight of around 450 kg were available (Munjonji
et al., 2020).

As a result of the low precipitation and temperatures, the cold steppe is least productive. Similar to the hot steppe, the
S-strategy is dominant and C- as well as R-strategists have marginal shares. We used data of experiments with two different
livestock densities of grazing sheep with a body weight of around 35 kg: the low grazing intensity (S1) of 1.5 sheep $\mathrm{ha^{-1}}$ and
the high grazing intensity (S6) with 9 sheep $\mathrm{ha^{-1}}$ (Hoffmann et al., 2016).

Table 1. Overview of the environmental conditions and management of the investigated grasslands.

| Site | Temperate grassland | | Hot steppe | | Cold steppe | |
|---|---|---|---|---|---|---|
| Location | Lindhof, Germany | | Syferkuil, South Africa | | Xilin, China | |
| Coordinates | $54°27'$N, $9°57'$E | | $23°85'$S, $29°7'$E | | $43°38'$N, $116°42'$E | |
| Mean annual temperature [°C] | 9.4 | | 20.5 | | 0.9 | |
| Mean annual precipitation [mm] | 746 | | 432 | | 329 | |
| Koeppen-Geiger class | Cfb | | BSh | | BSk | |
| Soil type | Sandy loam | | Loamy sand | | Sandy clay loam | |
| Management | Fertilisation | | Cattle grazing | | Sheep grazing | |
| Experiment | unfertilised | fertilised | ungrazed | grazed | low intensity | high intensity |
| Forage offtake [MgDM ha$^{-1}$yr$^{-1}$] | 7.9±1.6 | 9.2±2.0 | - | - | 0.4±0.3 | 0.6±0.2 |
| Leaf biomass [MgDM ha$^{-1}$] | - | - | 1.1±0.6 | 1.5±0.6 | - | - |
| SOC depth [m] | 0.3 | | 0.3 | | 1 | |
| SOC value [MgC ha$^{-1}$] | 69.7±3.7 | 71.9±3.4 | 36±20 | | 273±60 | |
| Literature | (DWD 2021, Reinsch et al. 2018a, b) | | (Munjonji et al. 2020, Scheiter et al. 2023) | | (Hoffmann et al. 2016, Ren et al. 2017; Wiesmeier et al. 2011) | |

## 2.3 Model development

To extend the LPJmL model to simulate different communities in which different ecological strategies are dominant, we focused on three aspects: First, we adapted resource uptake and distribution (Sect. 2.3.1) to improve niche differentiation (see Hardin, 1960). Second, we implemented the trade-off between fast tissue growth at low construction cost and longevity versus slow tissue growth at high construction cost and longevity described by the leaf economics spectrum (LES) (Sect. 2.3.2, Wright et al., 2004). Third, we altered the representation of the plants' life cycle (Sect. 2.3.3) to distinguish different reproductive strategies. We provide a qualitative description of the aspects of recent model development that are important for LPJmL-CSR in the main text and refer to the Appendix A and SI for the technical details and other minor improvements compared to the original code.

### 2.3.1 Resource uptake and distribution

In the LPJmL model, the different PFTs compete for space/light, water and nitrogen. In past model versions these resources were distributed between PFTs based on foliage projective cover (FPC). The FPC is used as a proxy for actual cover, which would require the explicit simulation of the plants' geometries. Distributing these different resources based on one variable neglected the importance of different traits for the uptake of different resources. In particular, water uptake should also be dependent on root traits such as the extent of the root network and the amount of fine root biomass (Tron et al., 2015). Using

root traits to determine access to water enables the model to simulate different strategies for water resource use. Therefore, we adapted the implementation of water supply to make it dependent on root biomass instead of FPC to provide a distinction between the criteria for above-ground and below-ground resource uptake and distribution. Based on the concept of the FPC we implemented a below-ground equivalent based on root instead of leaf biomass (A1). First, the PFTs' access to water from different soil layers is calculated as described in Schaphoff et al. (2018). Second, the amount of water available for the PFT is determined considering its root biomass and the new parameter ($k_{root}$), which is a proxy for root properties associated with morphological properties of the root network (e.g. branching and spread).

### 2.3.2  The leaf economic spectrum

The LES describes correlations between several plant functional traits. Among these are the specific leaf area ($SLA$) and the leaf longevity, which can be used to express the differences between resource acquisitive vs. resource conservative growth strategies (Wright et al., 2004). The resource acquisitive strategy is associated with fast growth of leaves at low construction costs with a high $SLA$ and a short longevity. In contrast, the resource conservative strategy promotes slow growth of long-lived leaves with low $SLA$. Therefore, to represent the trade-offs associated with the differences between these strategies a functional relationship between $SLA$ and leaf longevity can be used.

Despite the importance of $SLA$ and leaf longevity for several processes within LPJmL, the $SLA$ v. leaf longevity trade-off has not been implemented for managed grasslands in LPJmL before. $SLA$ is used to calculate the leaf area index (LAI) of a given grassland area from the dynamically computed leaf biomass, which is important for the interception of light energy and thus for photosynthesis. The leaf longevity was represented through turnover rates, which determine the amount of leaf biomass transferred to the litter layer (Schaphoff et al., 2018). As long as differences between ecological strategies were not considered and only one PFT was used to simulate a managed grassland, this approach was sufficient. However, this means that grasslands along a resource stress gradient only differed in their productivity but not in other aspects of the community. Yet in reality, slow growing, resource conservative plants in stress-prone ecosystems are not only less productive and supply less forage with a lower nutrient content (Lee, 2018; Onoda et al., 2017). Such ecosystems are also more vulnerable to overgrazing (Liu et al., 2013) and recover more slowly from disturbances (Teng et al., 2020). Incorporating the $SLA$ v. leaf longevity trade-off is essential to account for the differences between ecological strategies, which are important to adequately represent ecosystem functions of managed grasslands under different climatic conditions and management.

The $SLA$ v. leaf longevity trade-off has already been implemented in the related LPJmL-FIT model and applied to tropical (Sakschewski et al., 2015) and European forests (Thonicke et al., 2020). For this study, we implemented the $SLA$ v. leaf longevity trade-off for managed grasslands using a functional relationship between the two based on trait observations. Similar to Sakschewski et al. (2015), we derived a power law for $SLA$ and leaf longevity from trait data retrieved from the TRY database (Boenisch and Kattge, 2018). This power law provides a functional relationship between $SLA$ and leaf longevity, which is used to calculate the PFT-specific leaf longevity from predefined $SLA$ values within LPJmL-CSR (A2). Based on the alignment of the resource conservation axis of the root economic space (Bergmann et al., 2020) and the LES (Weigelt et al.,

2021), we assume that leaf and root longevity are not independent from each other and maintain a fixed ratio of the two in LPJmL-CSR.

### 2.3.3 Reproduction and mortality

Herbaceous plants are adapted to different growing conditions and therefore have different reproduction strategies and whole plant - or for graminoids phytomere - longevity. In LPJmL, each herbaceous PFT was simulated using only one average
individual with specified properties. Age mortality was implicitly included in the representation of turnover of leaves and roots and not as a separate process. The only additional cause of mortality was negative leaf and/or root biomass after allocation as a result of prolonged stress. While this may be caused by water stress, additional causes of mortality from water stress such as embolism (Jacobsen et al., 2019) as well as heat stress were not considered.

LPJmL does not simulate seed bank formation and reproduction is not limited by the amount of seeds available in a seed
bank. Instead, the establishment depends on the bare ground area and the PFT-specific establishment rate. Furthermore, in LPJmL 5, reproduction was simulated as a biomass increase of the average individual. We argue that this was not sufficient to simulate different reproduction strategies, which differ in the amount of seeds, seed survival and germination rates, and germination requirements (Thompson, 1987; Brown and Venable, 1986).

In the representation of CSR strategies in LPJmL-CSR, we retained the approach of establishing seedlings instead of seeds
but allowed PFTs to establish different numbers of seedlings in agreement with their reproductive strategy. To achieve this, we abandoned the approach of using only one average individual to simulate each PFT and introduced a dynamic number of average individuals assuming a homogeneous population (i.e. individuals of the same PFT share the same properties) but form the community together. Based on the existing implementation, we modified the reproduction so that additional individuals are established and thereby increase the number of average individuals simulated. Each day, the number of average individuals of
each PFT is increased if there was bare ground area available. The bare-ground area is distributed between established PFTs depending on their establishment rate $k_{est}$. The total amount of seedlings established is calculated based on $k_{est}$, accounting for the bare ground area. Subsequently, the number of average individuals is increased and the individual specific carbon and nitrogen stocks are adjusted. LPJmL-CSR does not consider trait plasticity or evolutionary processes and therefore does not account for phenotypic adaptation. This also means, that already established and newly established average individuals
share the same traits. Since space for plant establishment is limited and age-related mortality is common in natural grasslands (Zimmermann et al., 2010), we prohibit infinite increase of the number of average individuals by adding an age-mortality based on the growth efficiency to reduce the number of average individuals (A3). The growth efficiency is the ratio of the net change in the individual carbon stocks (the result of net photosynthesis and turnover) and the individual carbon stocks. Assuming that old plants grow more slowly this is used as a proxy for population age and resulting age-mortality. We did not implement
additional causes of mortality such as drought and fire (Zimmermann et al., 2010). While the new approach does not simulate individual and phytomere morphology explicitly, it provides some implicit information on community structure and plant size through the number of average individuals, the area covered by them and their biomass. It can be assumed that few individuals that maintain a high cover and biomass must be larger than more individuals that provide a similar cover and biomass.

## 2.4 Defining the C, S and R-PFT

We based our new PFTs on the already existing herbaceous PFTs (Schaphoff et al., 2018), from which we retained the majority of parameter values. For the temperate grassland we used the temperate herbaceous, for the hot steppe the tropical herbaceous and for the cold steppe, the polar herbaceous PFT. To design the new C-, S- and R-PFTs for each of these environments and given management scenarios, we assessed a subset of parameters that represent functional traits inspired by the *global spectrum of plant form and function* (Díaz et al., 2016) and define our trait space using the stress and disturbance gradient to distinguish

the CSR-strategies. Based on past sensitivity analyses (Forkel et al., 2019; Zaehle et al., 2005) and expected behaviour of newly implemented trade-offs, we selected four parameters for each dimension to distinguish the CSR-strategies (Tab. 2).

### 2.4.1 The stress and disturbance gradients

We assumed, that the position of a PFT within the CSR triangle can be determined through trade-offs between plant functional traits along the stress and the disturbance gradients according to the relations described below. Names, descriptions and usage

of the model parameters are based on the model versions LPJmL4 (Schaphoff et al., 2018) and 5 (von Bloh et al., 2018).

    According to CSR theory, stress is defined as constrained metabolic efficiency limiting biomass production and can be caused by a variety of factors (Grime, 1977). The stress gradient expresses the intensity of stress a species is exposed to in a certain habitat. It ranges from unstressed to severely stressed and can include the combined impacts of several stressors. Different traits and their values are associated with the ability of a plant to cope with the different stress levels. The traits of

the LES (Wright et al., 2004) together with different strategies for resource use can be used to distinguish C- and R-strategists (low stress tolerance) from S-strategists (high stress tolerance).

    Since the LPJmL model only represents a subset of possible stress factors (Sect. 2.1), only stress arising from temperature and water as well as nitrogen availability can be considered. Within LPJmL-CSR, some traits are linked to a general response to stress independent of the stressor, while other are used to represent adaptation to specific stressors. Since the grassland

steppe sites simulated by us are predominantly limited by water, we decided to focus on water stress. This allows for a better understanding of the underlying processes and the resulting patterns. To represent the stress gradient, we used functional traits associated with the growth rate and water-resource use. We selected the maximum transpiration rate ($E_{max}$), the minimum canopy conductance ($g_{min}$), the specific leaf area ($SLA$) and the leaf to root mass ratio ($lmro$).

$SLA$: The specific leaf area is the ratio of leaf area to leaf dry mass and a measure of the amount of biomass required

to produce a unit of leave area. It is predominantly associated with the stress gradient in the CSR theory. $SLA$ is used in four processes of LPJmL-CSR: First, it is used to calculate the LAI, which controls light interception and thus productivity determining the area occupied by a PFT in competition with other PFTs. Second, $SLA$ is used to determine the above-ground biomass of newly established seedlings from the seedling LAI (see explanation of $LAI_{sapl}$). Third, it is used to determine the actual mortality rate (A3). Fourth, it is used to calculate the leaf longevity controlling tissue

turnover and litterfall (Sect. 2.3.2). The $SLA$ can be used to determine the trade-off between short-lived, acquisitive

(high $SLA$) and long-lived, conservative (low $SLA$) leaves. In contrast, in LPJmL 5 it was only used in the first and second process.

$lmro$: The leaf mass to root mass ratio ($lmro$) is the target ratio of above- and below-ground biomass. It is predominantly associated with the CSR stress gradient but since it controls investments into above v. below-ground biomass it also affects the PFTs response to the removal of above-ground biomass. $lmro$ is used within two processes of LPJmL 5 and LPJmL-CSR: First, to determine the allocation of the current day's productivity to above- and below-ground biomass pools to approach $lmro$. Second, to calculate the below-ground biomass of newly established seedlings from the above-ground biomass of newly established seedlings (A3). The $lmro$ can be used to differentiate between strategies on investing assimilates for above- (high $lmro$) or below-ground (low $lmro$) growth and the resulting access to resources.

$E_{max}$: The maximum transpiration rate defines the upper limit of transpiration per day. It is predominantly associated with the CSR stress gradient. In LPJmL 5 and LPJmL-CSR, $E_{max}$ is used to calculate the water supply. Here, $E_{max}$ presents the upper limit and actual transpiration is reduced depending on the PFT-specific root distribution and the soil water content. $E_{max}$ can be used to distinguish more (low $E_{max}$) and less (high $E_{max}$) water saving strategies.

$g_{min}$: This defines the minimum canopy conductance in mm per second that is independent of photosynthesis and a result of other processes controlling the lower limit of transpiration. It is predominantly associated with the stress gradient. In LPJmL 5 and LPJmL-CSR, $g_{min}$ is used in the calculation of the total canopy conductance as a part of the photosynthesis routine. $g_{min}$ can be used to distinguish more (low $g_{min}$) and less (high $g_{min}$) water saving strategies.

Similar to the stress gradient, the disturbance gradient ranges from undisturbed to severely disturbed. Reproductive traits and plant stature (Westoby et al., 1996; Grime, 1974; Salisbury, 1943) can be used to distinguish C- and S-strategists (low disturbance tolerance) from R-strategists (high disturbance tolerance). Functional traits associated with reproduction and plant geometry can be used to represent the trade-off associated with the disturbance gradient. We selected the following functional traits involved in the direct interaction of the different PFTs: The root efficiency coefficient ($k_{root}$), the light extinction coefficient ($k_{beer}$), the establishment rate ($k_{est}$) and the leaf area index of a seedling ($LAI_{sapl}$). While seedling is the more intuitive term for herbaceous plants and we will use it throughout the manuscript, the subscript in the parameter name refers to saplings because it was adopted from the tree PFTs in the past.

$k_{beer}$: The light extinction coefficient is a parameter describing the amount of light absorbed by a vegetation layer. It is predominantly associated with the CSR disturbance gradient but since it is used in the calculation of the FPC, which also determines resource access, it is also associated with the CSR stress gradient. In LPJmL 5 and LPJmL-CSR, $k_{beer}$ is used to determine FPC controlling the PFT-specific area share and its access to light. $k_{beer}$ can be used as a proxy to distinguish large (high $k_{beer}$ - rarely shaded by competitors and have high light absorption capacity) from small (low $k_{beer}$ - potentially shaded by competitors and have high light absorption capacity only if dominant) stature plants and is essential for the competition for light and space.

$k_{root}$: The root efficiency coefficient is a parameter used as a proxy for root functional traits such as branching and density of the root network. It is predominantly associated with the CSR disturbance gradient but it also affects PFT-specific water access. $k_{root}$ was introduced in LPJmL-CSR and is used to represent the below-ground morphology controlling the PFT-specific share of the below-ground and its access to respective resources. $k_{root}$ can be used as a proxy to distinguish sparse and constrained (low $k_{root}$) from dense and spread root networks (high $k_{root}$) and is important for the competition for water.

$k_{est}$: The establishment rate describes the maximum amount of seedlings established per day. It is predominantly associated with the CSR disturbance gradient. While in LPJmL 5, $k_{est}$ was used to determine the increase of the biomass of the average individual, in LPJmL-CSR, $k_{est}$ is used to calculate the increase of the number of average individuals per m$^2$ from establishment on bare ground area. $k_{est}$ can be used to distinguish the number of offspring and thus strategies with low (low $k_{est}$) and high (high $k_{est}$) reproductive capacity.

$LAI_{sapl}$: The seedling LAI is the leaf area index of a newly established seedling. It is predominantly associated with the CSR disturbance gradient. In LPJmL 5 and LPJmL-CSR, it is used to calculate the above-ground biomass of a seedling using the PFT-specific $SLA$. It can be used to distinguish the biomass of offspring (low/high $LAI_{sapl}$ lead to low/high offspring biomass) which we use as a proxy for the competitive strength of the offspring of different strategies.

In total, we used eight parameters to distinguish the PFTs and defined plausible ranges for their parameterisation so that different CSR strategies can be represented by the extended set of PFTs. The selected traits affect a variety of processes within the model and differentiate the C-, S- and R-PFT along the stress and disturbance gradients. We assumed all parameters to be independent from each other. While we are aware that $SLA$ and the light extinction coefficient $k_{beer}$ are correlated in reality because the transmissivity of leaves increases with $SLA$ we have to treat them as independent because in LPJmL, the light extinction coefficient does not describe the transmissivity of a single leaf but of the entire vegetation layer. Stacking a high number of high transmissivity leaves may result in the same light extinction compared to a lower number of low transmissivity leaves. In LPJmL-CSR, a similar $k_{beer}$ would be assigned for both cases because it represents the light extinction coefficient of the entire vegetation layer.

### 2.4.2 Parameterisation and evaluation of new PFTs

To parameterise the new PFTs we had to assess the model performance for different parameter sets. We included several variables in the calculation of a likelihood ($logLI$): forage offtake or leaf biomass, SOC and C-, S- and R-strategy cover (Table 3). Data on forage offtake, leaf biomass and SOC were available from several field experiments conducted at the respective sites. For C-, S- and R-PFT cover, data were only available for the hot steppe and we defined values based on our knowledge of the site-specific conditions to agree with CSR theory for the other sites.

We optimised the $logLI$ using a Markov Chain Monte Carlo (MCMC) method with a Metropolis algorithm (Wirth et al., 2021; Van Oijen et al., 2005). This method evaluates the performance of a sequence of sampled parameter sets. In the following, we refer to a sequence as a *chain* and to an iteration as a *link*. At the beginning of the chain, a first parameter set is drawn from

**Table 2.** Parameter names, units, ranges, associated CSR gradient(s) and the hierarchy of the parameters for the C-, S- and R-PFTs.

| Parameter | Abbreviation | Unit | Min | Max | Predominant gradient | Subsidiary gradient | Hierarchy |
|---|---|---|---|---|---|---|---|
| Specific leaf area | $SLA$ | $[\mathrm{m^2 \cdot gC^{-1}}]$ | 0.01 | 0.1 | stress | - | S < C < R |
| Light extinction coefficient | $k_{beer}$ | [-] | 0.2 | 0.8 | disturbance | stress | R < C & S |
| Establishment rate | $k_{est}$ | $[\mathrm{Ind. \cdot m^{-2} day^{-1}}]$ | 3000 | 6000 | disturbance | - | R > C & S |
| Root efficiency coefficient | $k_{root}$ | [-] | 0.005 | 0.025 | disturbance | stress | R < S < C |
| Leaf to root mass ratio | $lmro$ | [-] | 0.6 | 1 | stress | disturbance | S < R < C |
| Maximum transpiration rate | $E_{max}$ | $[\mathrm{mm \cdot d^{-1}}]$ | 4 | 12 | stress | - | S < R < C |
| Seedling leaf area index | $LAI_{sapl}$ | [-] | 0.01 | 0.15 | disturbance | - | R < S < C |
| Minimum canopy conductance | $g_{min}$ | $[\mathrm{mm \cdot s^{-1}}]$ | 0.3 | 2 | stress | - | S < R < C |

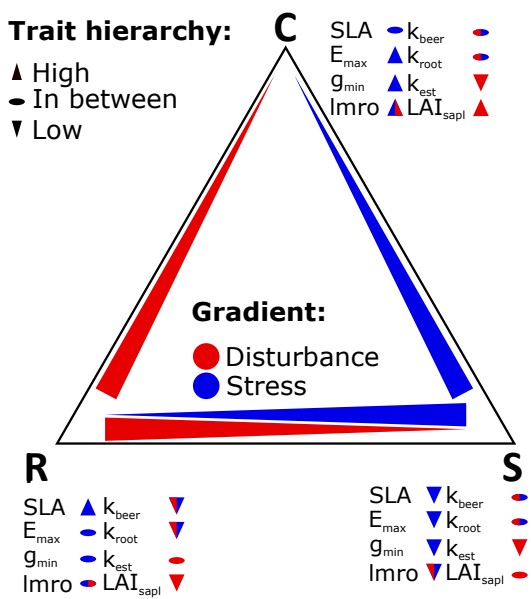

**Figure 1.** Stress (blue) and disturbance (red) gradient and associated traits and their hierarchy (low, in between and high)

a multivariate Gaussian distribution with its modes at the centre of the parameter ranges for each parameter and its variances as a fraction of the parameter ranges. A fraction of the ranges is used to limit the difference between parameter sets of subsequent links which improves the performance of the algorithm. The width of this fraction is controlled through a tuning parameter and is fixed for the entire chain, while the modes of the Gaussian distribution are updated throughout the chain if the model performance calculated as the total likelihood ($logLI$, Eq. 1) improves.

$$logLI_i = logPrior_i + logLink_i \tag{1}$$

**Table 3.** Variables used for parameterisation (para) and evaluation (eval) of the new PFTs at the study sites

| Site | Variable | Resolution | Usage | Source | Literature |
|------|----------|-----------|-------|--------|-----------|
| Temp. grassland | Dry matter yield | per cut | para/eval | field observations | (Reinsch et al., 2018b) |
| Temp. grassland | Soil carbon | annual | para/eval | field observations | (Reinsch et al., 2018b) |
| Temp. grassland | Cover of C-, S- and R-PFT | constant | para | expert estimate | - |
| Hot steppe | Leaf biomass | annual | para/eval | field observations | (Munjonji et al., 2020) |
| Hot steppe | Soil carbon | monthly | eval | field observations | (Munjonji et al., 2020) |
| Hot steppe | Cover of C-, S- and R-PFT | Hot steppe | constant | para | expert estimate | - |
| Cold steppe | Leaf biomass | monthly | para | field observations | (Schönbach et al., 2012) |
| Cold steppe | Grazing offtake | monthly | eval | field observations | (Schönbach et al., 2012) |
| Cold steppe | Soil carbon | constant | para | field observations | (Wiesmeier et al., 2011) |
| Cold steppe | Cover of C-, S- and R-PFT | constant | para | expert estimate | - |

The total likelihood $LogLI$ is calculated for each link $i$. It consists of a prior likelihood ($logPrior_i$, Eq. 2) and the likelihood of the current link ($logLink_i$, Eq. 3).

$$logPrior_i = \sum_j B(\theta_{i,j}, p, q) \tag{2}$$

The prior likelihood $logPrior$ is calculated from the prior distribution, which represents an initial guess on the resulting posterior distribution. We chose a geometrical prior distribution (B) with the shape parameters $p = 1 + 4 \cdot (\hat{\theta}_{centre} - \hat{\theta}_{min} / \hat{\theta}_{max} - \hat{\theta}_{min})$ and $q = 6 - p$. Here, $\theta_{i,j}$ are the parameter values of each parameter $j$ of the current link $i$, $\hat{\theta}_{centre}$ are the values at the centre of the parameter space, and $\hat{\theta}_{min}$ and $\hat{\theta}_{max}$ are the lower and upper limits of the parameter space, respectively.

$$logLink_i = \sum_k -0.5 \cdot \frac{y_{sim,i,k} - y_{obs,k}}{\sigma_{obs,k}}^2 -$$

$$0.5 \cdot log(2\pi) - log(\sigma_{obs,k}) \tag{3}$$

The likelihood of the current link, $logLink_i$, is a measure of the model performance of a simulation using $\theta_{i,j}$. $logLink_i$ incorporates the difference between simulation results ($y_{sim}$) and observations ($y_{obs}$) for all variables $k$ including also the uncertainty of the observations ($\sigma_{obs}$). The overall likelihood ($logLI_i$) is compared to the highest likelihood that was achieved so far ($LogLI_{max}$) to decide about the acceptance of the current parameter set. If the difference between the current likelihood

and the highest likelihood ($\Delta LogLI = LogLI_{max} - LogLI_i$) is positive, the parameter set is always accepted. For negative $\Delta LogLI$, it is only accepted if it exceeds the natural logarithm of a random number between 0 and 1. This mechanism prohibits that the algorithm is trapped in local optima. At the end of the chain, the algorithm returns a posterior parameter distributions whose modes are the parameter values with the best model performance.

We used the same parameter space for all three new PFTs but ensured that we select parameter values consistent with the traits associated with CSR theory. We prescribed a hierarchy based on our expertise (Tab. 2) for each parameter that defines whether a PFT has to obtain a higher or lower value compared to the other PFTs. For example S-PFTs must have a lower SLA than C- and R-PFTs because the S-strategy is associated with slower growth and longer living tissue than the C- and R-strategies.

Our approach included two steps represented by two subsequent chains. The first chain was short and used a large tuning parameter so that the sampling covered the entire parameter space and an area of good model performance could be identified. The second chain started in the area discovered by the first chain, was longer and used a smaller tuning parameter to find the optimal parameter values within the area.

We evaluated the new PFTs using the mean square error (MSE) and its components (see Appendix A6). For the evaluation, we either used a different data set or split the data into different sets for parameterisation and evaluation if the number of replicates was at least eight for the majority of observations (Table 3). Observations with less than eight replicates were only used for the parameterisation. For the hot steppe we used the difference in SOC between the ungrazed and grazed scenario for the evaluation because the current representation of the processes listed in Sect. 4.1.2 made it impossible to simulate the overall SOC level adequately. For the cold steppe, SOC data were only available for one year and the common management for the examined region (Wiesmeier et al., 2011). While this is comparable to our extensive grazing intensity, for the intensive grazing intensity we assumed a 25% lower SOC level. We based this assumption on Kölbl et al. (2011) who reported around 25% lower SOC content of the topsoil under heavy grazing compared to areas without or with periods of moderate grazing.

## 2.5 Modelling protocol

Simulations with LPJmL are driven by data on climate variables and management. If available, we used climate data obtained at the sites (see SI). For missing climate variables we supplemented data from the GSWP3-ERA5 data set for the temperate grassland and bias-adjusted data from the MRI-ESM2-0 (Lange and Büchner, 2022) for the hot and cold steppe. To design the new PFTs and evaluate the model development, we reproduced the management under which the experiments were conducted (Sect. 2.2 and Table 1).

LPJmL-CSR simulates all processes and provides all outputs with a daily resolution. If necessary, outputs are aggregated to a monthly or annual resolution in the postprocessing. Before simulating managed grasslands, the model was run for 30000 years with natural vegetation to obtain an equilibrium of the carbon and nitrogen cycle during a spinup simulation. Afterwards, a second spinup of 390 years was conducted to account for the effects of historical land-use change on soil conditions. For none of the sites, data on the land use history were available and we assumed livestock grazing with a moderate density for the second spinup period to account for the transition from natural vegetation to managed land. A detailed list of all inputs and settings to reproduce the conditions of the sites and experiments is provided in the SI.

In addition to the simulations done for the parameterisation of the new PFTs, we simulated several scenarios to analyse forage offtake, leaf biomass and SOC for different water or nitrogen limitation levels. For each site, we simulated the two management schemes also used to derive the new PFTs. To evaluate the changes of forage offtake, leaf biomass, SOC and

**Table 4.** Scenarios names and management (mowing/grazing intensity, irrigation, fertilisation) used for the simulations at the Lindhof (temperate grassland), Syferkuil (hot steppe) and Xilin (cold steppe) site.

| Name | Mowing/grazing | Irrigation | Fertiliser application |
|---|---|---|---|
| Temperate grassland N0 | Mowing (4 Cuts) | rainfed | unfertilised |
| Temperate grassland N1 | Mowing (4 Cuts) | rainfed | fertilised $240 \, \mathrm{kgN \cdot ha^{-1} \cdot year^{-1}}$ |
| Hot steppe C0 R U | Grazing ($0.0 \, \mathrm{Cows \cdot ha^{-1}}$) | rainfed | unfertilised |
| Hot steppe C1 R U | Grazing ($0.1 \, \mathrm{Cows \cdot ha^{-1}}$) | rainfed | unfertilised |
| Hot steppe C0 I U | Grazing ($0.0 \, \mathrm{Cows \cdot ha^{-1}}$) | irrigated | unfertilised |
| Hot steppe C1 I U | Grazing ($0.1 \, \mathrm{Cows \cdot ha^{-1}}$) | irrigated | unfertilised |
| Cold steppe S1 R U | Grazing ($1.5 \, \mathrm{Sheep \cdot ha^{-1}}$) | rainfed | unfertilised |
| Cold steppe S6 R U | Grazing ($9 \, \mathrm{Sheep \cdot ha^{-1}}$) | rainfed | unfertilised |
| Cold steppe S1 I U | Grazing ($1.5 \, \mathrm{Sheep \cdot ha^{-1}}$) | irrigated | unfertilised |
| Cold steppe S6 I U | Grazing ($9 \, \mathrm{Sheep \cdot ha^{-1}}$) | irrigated | unfertilised |
| Cold steppe S1 I F | Grazing ($1.5 \, \mathrm{Sheep \cdot ha^{-1}}$) | irrigated | fertilised |
| Cold steppe S6 I F | Grazing ($9 \, \mathrm{Sheep \cdot ha^{-1}}$) | irrigated | fertilised |

community composition in response to different resource limitations, we simulated our three sites additionally without the prevailing site-specific limitations. For this, we removed water limitation for the hot steppe and water and nitrogen limitation
separately for the cold steppe (Table 4).

Pre- and postprocessing of the data and figure creation were conducted using R (R Core Team, 2019). A list of all R packages used is provided in the SI.

## 3 Results

We evaluated LPJmL-CSR for the selected variables (Sect. 3.1) - results of the parameterisation are shown in the SI. Afterwards,
we assessed the effect of removing the resource limitations (Sect. 3.3), compared the traits and trade-offs within and across sites (Sect. 3.2) and analysed the community composition (Sect. 3.4).

### 3.1 Evaluation of new PFTs

For each site and management scenario, the new PFTs led to improved model results for forage offtake/leaf biomass and a reduced mean square error (MSE) compared to a simulation using LPJmL 5 (Fig. 2 a, d and g), which did not include the
changes described in Sect. 2.3. A major improvement was the capability of LPJmL-CSR to distinguish between CSR-strategies using different PFTs. For all sites and strategies we were able to find parameter sets for the new PFTs that enable LPJmL-CSR to represent the community well. Annual averages of the C-, S- and R-PFT cover simulated by LPJmL-CSR compared well to the expected cover which we used for the parameterisation. MSEs for the FPC were below 0.02 (Fig. 2 c, f and g) across sites

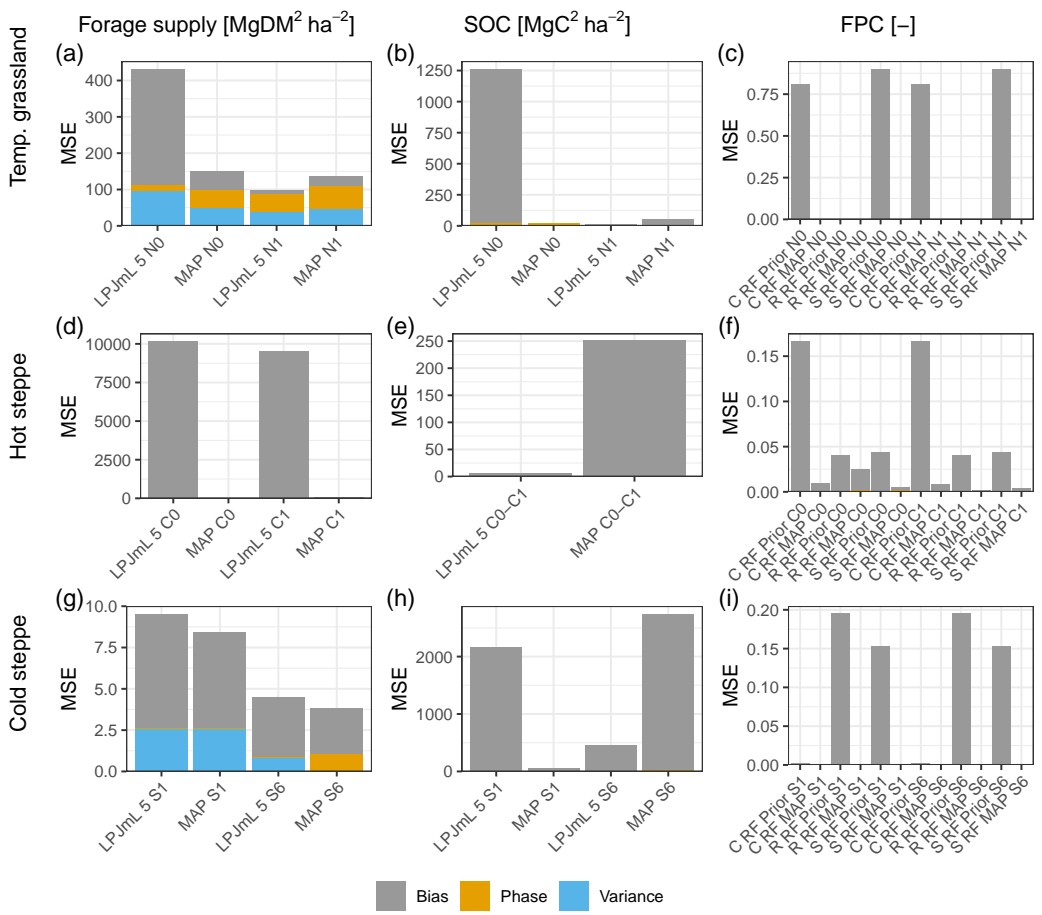

**Figure 2.** Mean square error (MSE) for the different management scenarios (x-axis) for forage offtake/leaf biomass in $\mathrm{MgDM} \cdot \mathrm{ha}^{-1} \cdot \mathrm{yr}^{-1}$, SOC in $\mathrm{MgC} \cdot \mathrm{ha}^{-1} \cdot \mathrm{yr}^{-1}$ and FPC (columns, left to right) for the temperate grassland, hot and cold steppe (rows, top to bottom). For forage offtake/leaf biomass and SOC, MSEs for the old (LPJmL 5) and new (LPJmL-CSR) model version are shown. For FPC MSEs are shown for each PFT separately for LPJmL-CSR before and after the calibration. The colours separate the MSE into three components: the bias (grey) showing the systematic error for each variable, the phase (yellow) showing the temporal shift against observations and the variance (blue) which is the random error not attributable to bias and phase compared to observations.

and scenarios. Simulation results for forage offtake/leaf biomass improved at all sites (Fig. 2 a, d and g). For the temperate grassland and the extensive grazing scenario in the cold steppe, the MSE of SOC was lower in LPJmL-CSR (Fig. 2 b and h) but similar for the hot steppe and moderately higher for the intensively grazed cold steppe (Fig. 2 e and h).

### 3.1.1 Temperate grassland

Forage offtake of the temperate grassland for the unfertilised scenario was strongly underestimated by LPJmL 5 (Fig. SI 2 a) and the MSE improved from 431.7 to 112.2 $(\mathrm{MgDM} \cdot \mathrm{ha}^{-1})^2$ in LPJmL-CSR (Fig. 2 a). For the fertilised scenario, LPJmL 5 underestimated forage offtake less severely (Fig. SI 2 b) and the MSE was similar with 96.4 $(\mathrm{MgDM} \cdot \mathrm{ha}^{-1})^2$ in LPJmL 5 to 105.3 $(\mathrm{MgDM} \cdot \mathrm{ha}^{-1})^2$ in LPJmL-CSR. For the unfertilised scenario, the representation of SOC improved as well. For the unfertilised scenario LPJmL 5 strongly underestimated SOC stocks (Fig. SI 3 a) and the MSE was reduced from 1262 to 21.4 $(\mathrm{MgC} \cdot \mathrm{ha}^{-1})^2$. However, it remained similar with 11 and 37.9 $(\mathrm{MgC} \cdot \mathrm{ha}^{-1})^2$ for the fertilised scenario (Fig. 2 b SI 3 b).

### 3.1.2 Hot steppe

Simulation results for the hot steppe presented a mixed picture showing lower MSEs for leaf biomass but higher MSEs for SOC in LPJmL-CSR compared to LPJmL 5. For the ungrazed (C0) scenario, the MSE of leaf biomass improved from 10154.1 to 1.9 $(\mathrm{MgDM} \cdot \mathrm{ha}^{-1})^2$ (Fig. 2 d). Similarly, for the grazed (C1) scenario, the MSE of leaf biomass improved from 9522.5 to 40.1 $(\mathrm{MgDM} \cdot \mathrm{ha}^{-1})^2$. The MSE for the difference in SOC between the ungrazed and grazed scenario was lower in LPJmL 5 and increased from 6.3 to 251.2 $(\mathrm{MgC} \cdot \mathrm{ha}^{-1})^2$ (Fig. 2 e). LPJmL 5 already simulated the SOC difference between the scenarios well impeding improvements through LPJmL-CSR. Furthermore, improvements in leaf biomass outweighed degradation in SOC stocks and LPJmL-CSR fits the observations better overall. However, compared to observations, LPJmL 5 severely overestimated leaf biomass and LPJmL-CSR underestimated leaf biomass (Fig. SI 4) and both model versions overestimated SOC in the ungrazed and grazed scenario (Fig. SI 5).

### 3.1.3 Cold steppe

For the cold steppe, animal feed demand was met in both model versions for the low grazing intensity (S1). Still, the MSE for forage offtake improved from 9.5 to 8.4 $(\mathrm{MgDM} \cdot \mathrm{ha}^{-1})^2$ (Fig. 2 g). For the high grazing intensity (S6), the feed demand was not always met in both models versions. Here, the MSE improved from 4.5 in LPJmL 5 to 3.8 $(\mathrm{MgDM} \cdot \mathrm{ha}^{-1})^2$. Both LPJmL 5 and CSR underestimated observed forage offtake for both grazing intensities but the dynamics at the high grazing intensity were captured better by LPJmL-CSR (Fig. SI 6). Unfortunately, replicates for forage offtake were not sufficient to split the data and no additional data were available for evaluation. Similarly, only data on SOC for one year, which did not distinguish between areas of different grazing intensity, were available. Since these data were already used for the parameterisation, we were not able to properly evaluate SOC. While LPJmL 5 strongly underestimated SOC for the low grazing intensity (S1), LPJmL-CSR captured the observations better but still underestimated observations (Fig. SI 7). Values were within the standard deviation of the observations for the low grazing intensity. For the high grazing intensity (S6), we assumed that 75% of the observed SOC

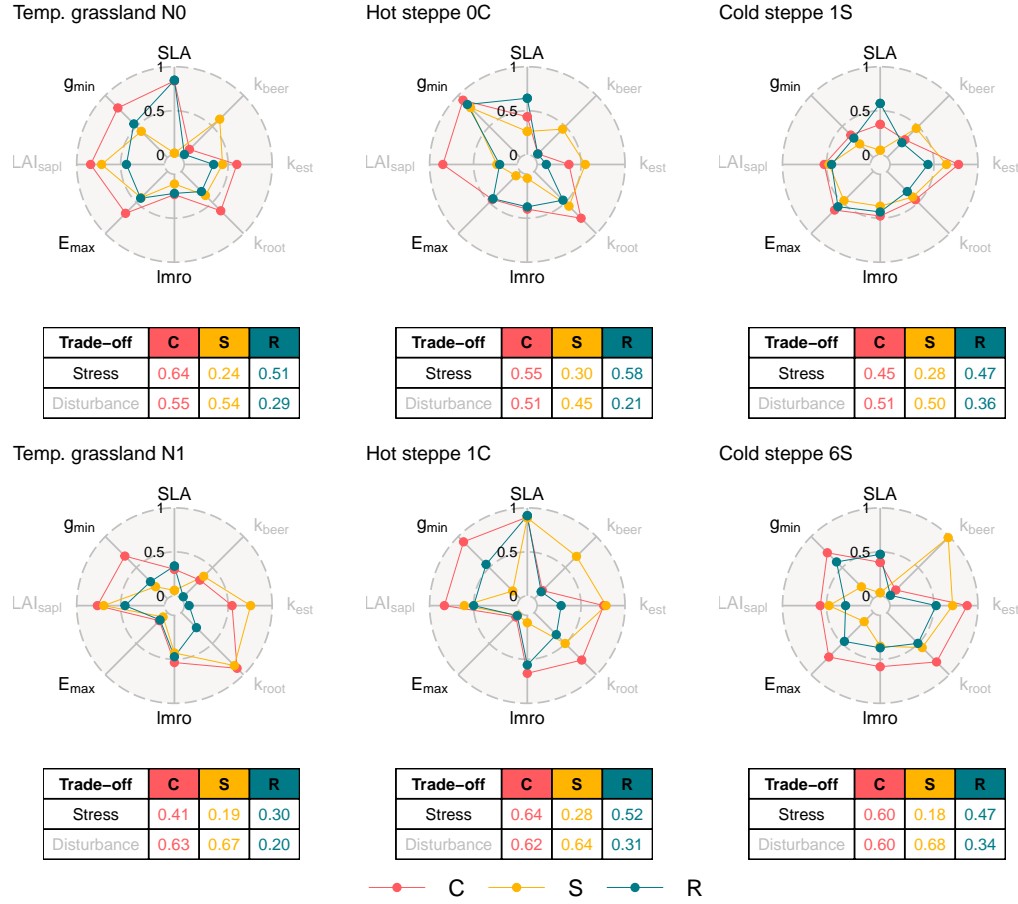

**Figure 3.** Spiderplots of normalised parameter values after calibration for each site (columns) and management scenario (rows). The centre and the edge represent the low and high end of the stress (black labels) and disturbance (grey labels) gradients. The colours of the points distinguish the three PFTs. The tables show the mean of normalised parameter values for each PFT and the two trade-off dimensions.

to be an appropriate estimate for calibration (Sect. 2.4.2). However, both LPJmL 5 and LPJmL-CSR overestimate this reduced calibration estimate (Sect. 4.1.3). The MSE was reduced from 2157.5 to 60.5 $(\mathrm{MgC} \cdot \mathrm{ha}^{-1})^2$ for the low and increased from 456.7 to 2741.5 $(\mathrm{MgC} \cdot \mathrm{ha}^{-1})^2$ for the high grazing intensity (Fig. 2 h).

### 3.2 Comparison of parameterisations between sites and different management intensities

The environmental conditions, the management and the communities at the examined sites were different, and each site and management could be placed at a different location within the CSR-triangle. Therefore, we expected different parameterisations across and within the sites for our new PFTs reflected through the PFTs' positions along the stress and disturbance gradients. Throughout this study, we focus on the two dimensions and discuss parameters in the context of these dimensions.

### 3.2.1 Management intensities

At all sites, the different management scenarios resulted in different parameter values for the three PFTs. For the temperate grassland, the calibration selected a less resource exploitative strategy for the C and R-PFT in the fertilised scenario indicated by the lower value for the stress gradient which resulted from higher leaf longevity (lower $SLA$), while the S-PFTs' strategy remained similar (Fig. 3). Additionally, all PFTs showed a lower maximum transpiration rate ($E_{,}max$) and higher investments into above-ground biomass (higher $lmro$). For the disturbance gradient, the C- and S-PFT had a higher value in the fertilised scenario. For the C- and S-PFT this indicated that the calibration selected a strategy with less offspring (lower $k_{est}$) and a more efficient root network (higher $k_{root}$). The R-PFT had a lower value caused by an increase in number of offspring (higher $k_{est}$).

For the hot steppe, the S- and R-PFT showed a lower value for the stress gradient for the grazed scenario (C1) and the calibration selected a more water saving (lower $E_{max}$ and/or $g_{min}$) strategy. These differences were counteracted to some extent by an increased investment in above-ground biomass (higher $lmro$) and a more resource exploitative strategy (higher $SLA$). The C-PFT showed similar differences except for the reduction of the minimum canopy conductance ($g_{min}$). However, this is likely an artefact of the parameterization. As stated in Sect. 2.4.1, both $SLA$ and $lmro$ do not only underpin the compensation of defoliation but can also play a role for resource uptake and distribution. In the ungrazed scenario (C0), no defoliation has to be compensated and both parameters only play a role for resource uptake and distribution which likely affected the selection of $g_{min}$. In contrast in the grazed scenario (C1), $g_{min}$ and $E_{max}$ become more important for resource uptake and distribution. For the disturbance gradient, all PFTs had higher values from different causes: The C-PFT established less offspring (lower $k_{est}$). The S-PFT increased its stature (higher $k_{beer}$), and seedling size (higher $LAI_{sapl}$). The R-PFT only increased its seedling size (higher $LAI_{sapl}$).

Consistent with the findings for the other sites, for the cold steppe all PFTs showed different strategies for the different management intensities. While the value for the stress gradient was the same for the R-PFT and only differed for the C- and S-PFT, the calibration selected different trait values for all PFTs. For the C-PFT the calibration selected a less water-saving strategy (higher $E_{max}$ and $g_{min}$) and for the S-PFT a more water-saving strategy (lower $E_{max}$ and $g_{min}$). For the disturbance gradient, the C- and S-PFT showed a higher value and the R-PFT a lower value in the intensively grazed scenario (S6). While for the C-PFT this was the result of an increase in the efficiency of its root network (higher $k_{root}$), for the S-PFT this was a result of an increase in stature (higher $k_{beer}$ Sect. 2.4.1). In contrast, the R-PFT had a smaller stature (lower $k_{beer}$) and seedling size (lower $LAI_{sapl}$).

### 3.2.2 Site-specific conditions

Across sites, we found a large variation within both dimensions which ranged from 0.30 to 0.64 for the stress and from 0.18 to 0.68 for the disturbance gradient (Fig. 3). As a consequence of our assumptions for the parameterisation, the sorting of the parameter values for the three PFTs had to match the hierarchy defined in Table 2 (Sect. 2.4.2) for each site. Between sites however, we did not make any assumptions that would predetermine an order, meaning that each site could occupy a different area of the two dimensions. For example, an R-PFT had to have a higher value for the stress gradient compared to the S-PFT

for the same site, but could have a lower value compared to the S-PFT of another site, as is the case when comparing the temperate grassland to the hot steppe. For the disturbance gradient, the same case can be made.

However, if averaged over all sites and management scenarios, the C-PFT still was the most resource exploitative with a value of 0.55 for the stress gradient, while the R- and S-PFT were more resource conservative with values of 0.48 and 0.25. Similarly, the R-PFT produced most offspring and had the smallest stature with a value of 0.29 compared to 0.57 and 0.58 for the C- and S-PFTs. While this general pattern emerged clearly for the two dimensions, there were substantial differences between the sites when comparing the contributing parameters. Most similar was the $lmro$ determining investments into above- versus below-ground biomass, which contributed to high values of the C- and R-PFT for the stress gradient for several scenarios. For the steppe sites, there was some alignment within the S-PFTs, which all had a larger stature (higher $k_{beer}$). The remaining parameters were not discernibly aligned across sites.

### 3.3 Effects of resource limitation

To assess the effect of resource limitation, we compared different scenarios with LPJmL-CSR. In addition to the scenarios using the prevailing climatic conditions (resource limited), we simulated scenarios where we removed the limitation of water or nitrogen supply. For the temperate grassland and the hot steppe, the different management of the unfertilised and fertilised and ungrazed and grazed scenario led to differences in soil carbon before the first year shown in Fig. 3.

#### 3.3.1 Temperate grassland

The temperate grassland already is a productive site where water and nitrogen are not limiting productivity and we did not simulate any additional scenarios but focused on comparing the two fertilisation levels (N0 and N1). For both scenarios total annual forage offtake was similar and between 5.3 and 7.4 $\mathrm{MgDM}\ \mathrm{ha}^{-1}\ \mathrm{year}^{-1}$ for the unfertilised and between 4.7 and 8.9 $\mathrm{MgDM}\ \mathrm{ha}^{-1}\ \mathrm{year}^{-1}$ for the fertilised scenario (Fig. 4 a). The first cut was the most productive, yielding between 1.8 and 2.8 $\mathrm{MgDM}\ \mathrm{ha}^{-1}\ \mathrm{year}^{-1}$ for the unfertilised and between 2.5 and 3.9 $\mathrm{MgDM}\ \mathrm{ha}^{-1}\ \mathrm{year}^{-1}$ for the fertilised scenario. The subsequent cuts contributed substantially to the overall forage offtake except in 2018, which was a drought year. Here, the forage offtake from all cuts was reduced. In all cuts the dominant C-PFT contributed the majority of the forage offtake. In both scenarios, the S- and R-PFT barely contributed - 2 and 8% share of forage offtake on average - in all cuts (Fig. SI 8). Overall, DMY was more stable between years (except 2018) in the fertilised scenario because of higher yields during the regrowth stages (cuts 2 to 4). These compensated the slightly lower DMY of the first cut compared to the unfertilised scenario.

The SOC showed no significant trend for the unfertilised scenario, where the annual average decreased by 0.02 $\mathrm{MgC}\ \mathrm{ha}^{-1}\ \mathrm{year}^{-1}$ on average ($\tau$=0.09, p-value 0.1). In contrast, SOC in the fertilised scenario increased by 0.96 $\mathrm{MgC}\ \mathrm{ha}^{-1}\ \mathrm{year}^{-1}$ on average ($\tau$=0.56, p-value <0.001), respectively (Fig. 4 d). Intra-annual SOC dynamics, which are driven by the litter production and C input from manure, were stronger in the fertilised scenario.

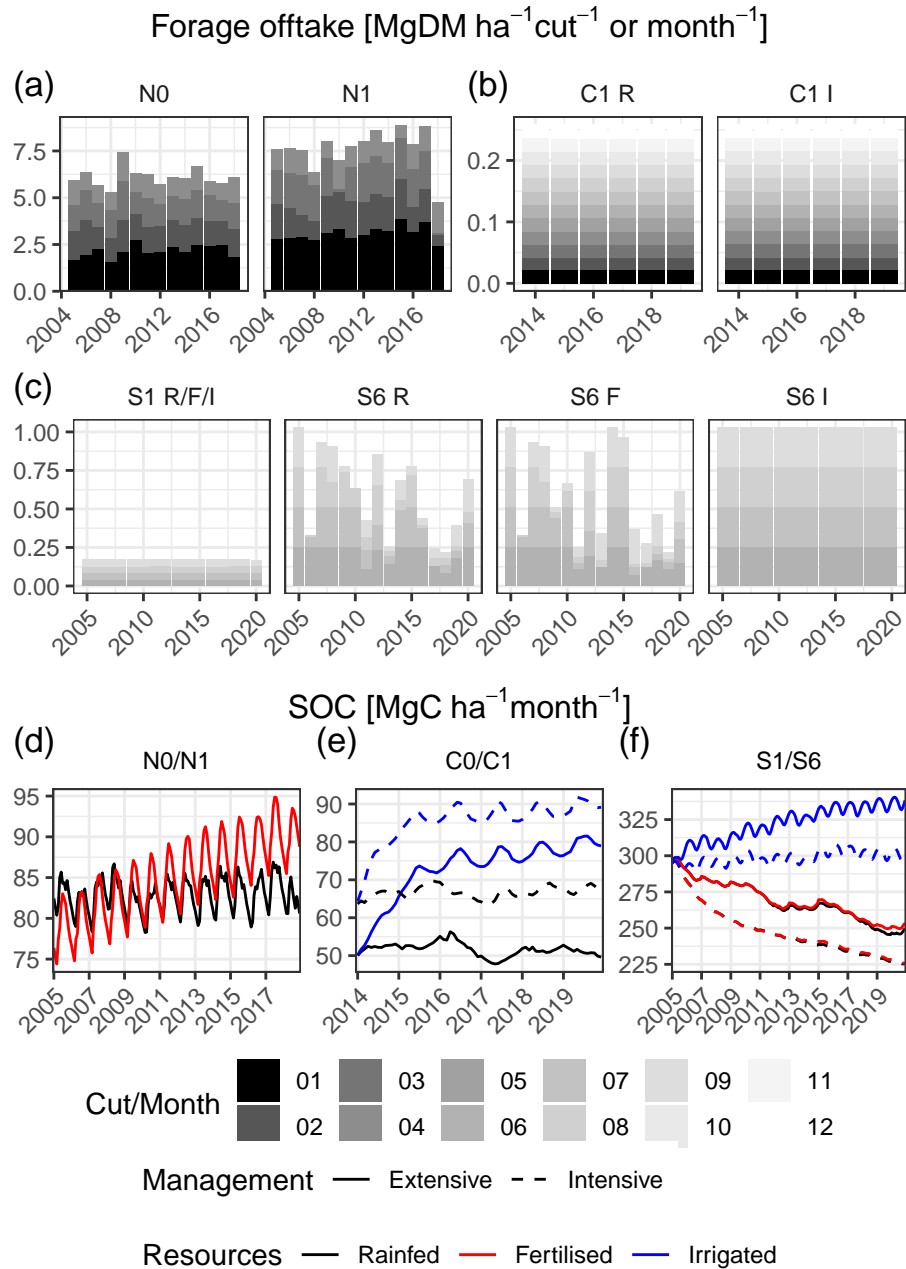

**Figure 4.** Simulated forage offtake/leaf biomass (a,b,c) and SOC (d,e,f) for all sites, management levels and resource limitation scenarios. Bars show the annual forage offtake and coloured segments the forage offtake for each cut/month. Line colours differ between rainfed (prevailing conditions, black), rainfed fertilised (red) and irrigated unfertilised (blue) while line-types show the grazing management intensity low (ungrazed/C0 or extensively grazed/S1, solid) and high (grazed/C1 or intensively grazed/S6, dashed).

### 3.3.2 Hot steppe

For the hot steppe, we simulated an irrigated (I) scenario in addition to the rainfed (R) scenario which was used for the calibration of our PFTs for the ungrazed (C0) and grazed (C1) management. Annual forage offtake was 0.26 MgDM ha$^{-1}$ year$^{-1}$ in the rainfed scenario (Fig. 4 b) and animal feed demand was always met (Fig. SI 9 a). Similarly, the feed demand was always met in the irrigated scenario (Fig. SI 9 b). However, between the two scenarios the composition of forage offtake strongly differed: In the rainfed scenario, the S-PFT contributed the majority in most years whereas in the irrigated scenario, the community composition changed and all PFTs contributed to forage offtake similarly. A shift also occurred in the ungrazed scenario, which was still dominated by the S-PFT but showed a higher share of the C- and R-PFT after several years as well. This change was related to the changing community composition (Sect. 3.4.2) and increased leaf biomass in the irrigated scenario. In the ungrazed scenario, 55% of the leaf biomass increase from irrigation resulted from elevated growth of the S-PFT, 21% from the C- and 23% from the R-PFT. This was different in the grazed scenario, with -18% (S-PFT), 82% (C-PFT) and 37% (R-PFT) respectively.

The SOC of the rainfed scenarios did not show strong trends (Fig. 4 e). However, the negative trend in the ungrazed scenario (C0) was still significant ($\tau$=-0.27, p-value <0.001). In the irrigated scenario, SOC increased strongly with little differences between the grazing scenarios - on average by 4.9 MgC ha$^{-1}$ year$^{-1}$ in the ungrazed and 4.2 MgC ha$^{-1}$ year$^{-1}$ in the grazed scenario. However, SOC did not increase linearly but showed a much stronger increase which was unrealistically high in the first one to two years after the start of irrigation (10.4 MgC ha$^{-1}$ year$^{-1}$ in the ungrazed and 10.6 MgC ha$^{-1}$ year$^{-1}$ in the grazed scenario) than in the remaining time-series (2.1 MgC ha$^{-1}$ year$^{-1}$ in the ungrazed and 1.0 MgC ha$^{-1}$ year$^{-1}$ in the grazed scenario).

### 3.3.3 Cold steppe

For the cold steppe, we simulated an irrigated (I) and a fertilised (F) scenario in addition to the rainfed (R) scenario used for the parameterisation for both the low (S1) and high (S6) grazing intensities. Total forage offtake was 0.17 MgDM ha$^{-1}$ year$^{-1}$ for all scenarios with low grazing intensity because the feed demand of the animals was always met (Fig. 4 c). In all scenarios the forage offtake was almost entirely attributed to the dominant S-PFT (Fig. SI 10 a-c). For the high grazing intensity, total forage offtake was 1.03 MgDM ha$^{-1}$ year$^{-1}$ if the feed demand was met. This was always the case in the irrigated scenario but not in the rainfed and fertilised scenarios. In the latter two, the model simulated very similar forage offtake, indicating that nitrogen addition was not sufficient to increase productivity because water was the main limiting factor. In all three scenarios, the S-PFT was dominant (Fig. SI 10 d-f). However, in the rainfed and fertilised scenarios the share of the S-PFT decreased in months where the feed demand could not be met and mainly the share of the C-PFT increased. In the irrigated scenario, only the S-PFT contributed to the forage offtake (explanation see Sect. 4.1.3).

SOC was similar for the rainfed and fertilised but differed for the irrigated low and high grazing intensity scenarios (Fig. 4 f). Both the rainfed and fertilised scenarios showed a significant negative trend for SOC, which was similar between the high grazing intensity where SOC decreased by roughly 4 MgC ha$^{-1}$ year$^{-1}$ on average ($\tau$=-0.99, p-value <0.001) and the low

grazing intensity with SOC losses of 3 $\mathrm{MgC\ ha^{-1}\ year^{-1}}$ on average ($\tau$=-0.87, p-value <0.001). For the irrigated scenarios, SOC increased by 2.5 $\mathrm{MgC\ ha^{-1}\ year^{-1}}$ on average ($\tau$=0.79, p-value <0.001) for the low and 0.4 $\mathrm{MgC\ ha^{-1}\ year^{-1}}$ on average ($\tau$=0.47, p-value <0.001) for the high grazing intensity.

## 3.4 Community composition

We compared expected and realised shares of the C-, S- and R-PFT for the three sites using leaf biomass, explored seasonal and inter-annual dynamics and analysed shifts under different resource limitations.

As already evidenced by the low MSE values for the FPCs of all PFTs after calibration (Fig. 2), LPJmL-CSR captured our expert estimates on C-,S- and R-PFT cover, which defined the position of the ecosystem within the CSR triangle, well. However, these were annual averages and did not prescribe any intra-annual variability. Since above-ground biomass and FPC are directly related and above-ground biomass is the less abstract variable to interpret, we present results based on above-ground biomass from here on.

### 3.4.1 Intra-annual variability

Each site showed substantial intra-annual dynamics of total above-ground biomass (Fig. SI 11 a, b, 12 a, c, 13, a, g) and the monthly average of the above-ground biomass share of the C- S- and R-PFTs (Fig. 5). However, the intra-annual dynamics were different between sites. In the temperate grassland, the C-PFT was dominant throughout the year, however, after the end of a growing season, the marginal PFTs had an increasing share until after the first cut (Fig. SI 11 c, d). While in the unfertilised (N0) scenario, the share of the S-PFT increased, the share of the S- and R-PFT increased in the fertilised (N1) scenario (Fig. 5 a).

In the hot steppe, the community was dominated by the S-PFT in both management scenarios (Fig. 5 b and d). In the ungrazed (C0) scenario, the C-PFT made up almost the entire remainder of the above-ground biomass (Fig. SI 12 a). However, the C-PFT was replaced by the R-PFT in the grazed (C1) scenario (Fig. SI 12 c).

For the cold steppe, PFT shares of above-ground biomass did not show strong intra-annual variation for the extensive (S1) grazing scenario (Fig. 5 c). However, for the intensive (S6) grazing scenario the C- and R-PFT strongly contributed to the overall leaf biomass and the C-PFT was even dominant in during and after the grazing period (Fig. SI 13 h).

### 3.4.2 Effects of irrigation and fertilisation

Removing resource limitations led to a shift of the community composition for the hot and cold steppe.

The hot steppe transitioned from an S-dominated community to a community with more balanced CSR shares that was still dominated by the S-PFT (Fig. 5 b and d). This transition occurred within the first one to two years after the beginning of irrigation for both scenarios (Fig. SI 12 e-g), which was reflected through the shift of the community average in Fig. 5 b and d. Whether or not this is the new equilibrium state or the community is still transitioning is crucial (Sect. 4.1.2).

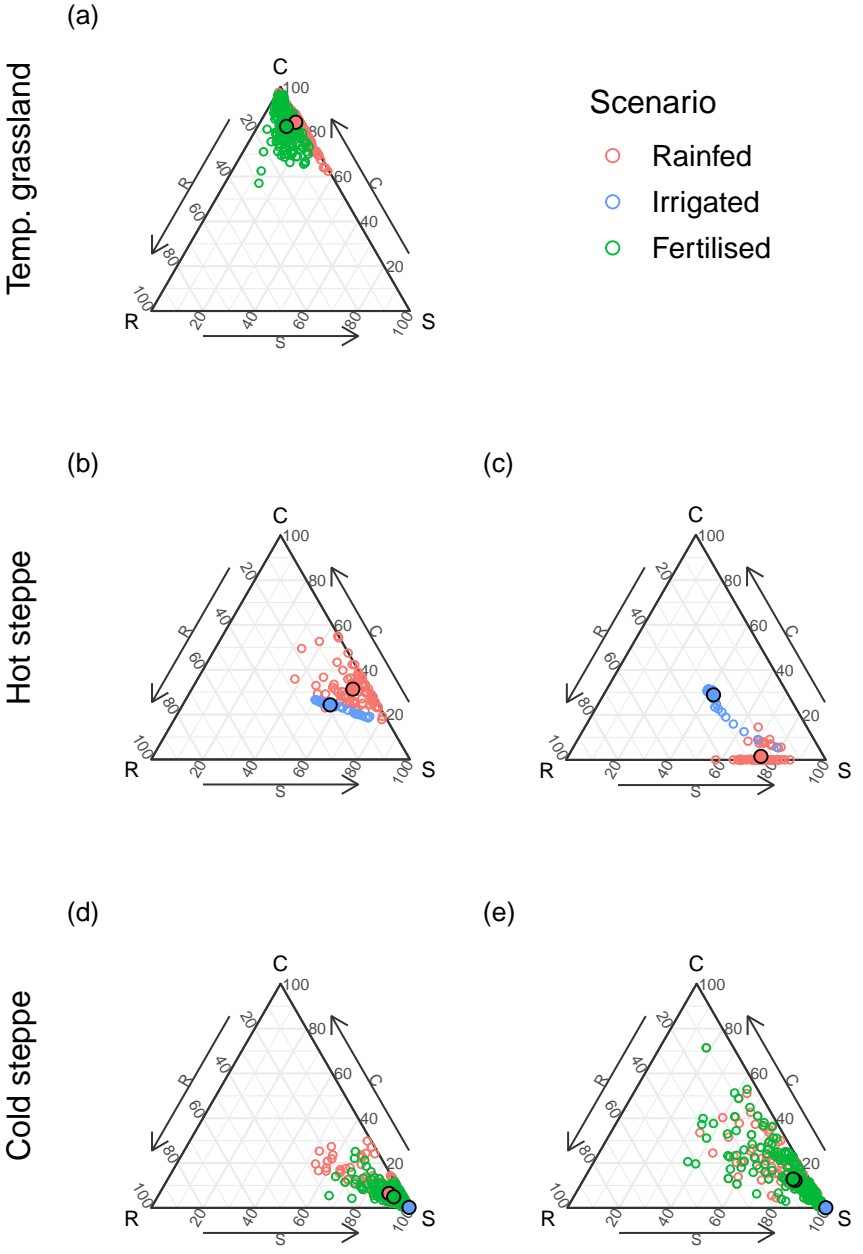

**Figure 5.** Ternary plots of the share of standing above-ground biomass of the C-, S- and R-PFT for the temperate grassland (a), the ungrazed (b) and grazed hot steppe (c), and the extensively (d) and intensively (e) grazed cold steppe. Colours differ between the rainfed (red), irrigated (blue) and fertilised (green) scenarios. Points with a black border show the mean composition of the time-series.

While removing the nitrogen limitation did not alter the community composition of the cold steppe under extensive (S1) and intensive (S6) grazing, irrigation had an effect (Fig. 5 c and e). The S-PFT out-competed the other PFTs entirely in the both grazing scenarios throughout the time-series (Fig. 5 c and e and SI 13 e, f, k, l).

## 4 Discussion

### 4.1 Forage offtake, SOC and community composition under different management and resource limitations

At all sites, forage offtake, SOC, and community composition differed between the different management intensity and resource limitation scenarios. The implemented model extension enabled the model to successfully simulate differences between C-, S- and R-strategists (Sect. 4.2). We were able to define new PFTs using a Bayesian calibration method that led to improved simulation of forage offtake and/or SOC at three sites under different environmental conditions and management. Our implementation is a major advancement because:

1. it allows for explicit analyses of the adaptation of the vegetation to changing conditions compared to the model version in which only productivity changed.

2. changes in the productivity of the community caused by changing conditions are the result of a changing community composition and should therefore not only be quantitatively different to those in LPJmL 5 but also more reliable.

3. this allows assessment of the adaptive capacity under different levels of functional diversity by adding or removing specific strategies.

Furthermore, in LPJmL-CSR the initial community composition is not dependent on additional data which facilitates the application at different sites or at larger scales.

#### 4.1.1 Temperate grassland

While the fertilised scenario for the temperate grassland was already well simulated in LPJmL 5, the unfertilised scenario underestimated forage offtake (Sect. 3.1.1). In LPJmL-CSR, growth of the vegetation was faster than in LPJmL 5 which led to higher yields for all cuts. We identified two reasons for the faster growth. First, the new implementation for biological nitrogen fixation (Appendix A4) reduced nitrogen stress and promoted higher photosynthesis rates. Second, while the parameters used for LPJmL-CSR were tuned for performance under the site-specific environmental conditions and management, the parameters used in LPJmL 5 were defined for large scale simulations with different management.

The temperate grassland is neither water nor nutrient limited and since we only assessed scenarios with reduced resource limitations, we only compared the fertilised and unfertilised scenarios. Despite the additional nitrogen input in the fertilised scenario, the unfertilised scenario achieved a similar forage offtake. Missing nutrients were acquired through biological nitrogen fixation, which was much higher in the unfertilised scenario which is in line with the higher share of legumes observed in the field experiments (Reinsch et al., 2020). Despite the higher share of legumes in the unfertilised experiments, the share of

C-, S- and R-strategists was similar and both fertilisation levels were dominated by C-species, which was well-represented by the model.

The simulated SOC was strongly dependent on the land use history for which available data were limited. For simplicity we did not simulate crop rotations for the land use history but selected a livestock density of $1.0\ \text{LSUs} \cdot \text{ha}^{-1}$ for the land use spinup simulation (see Sect. 2.5 and SI) to prescribe a fixed grazing pressure, which led to an underestimation of observations in unfertilised scenario in LPJmL 5. This indicated, that carbon inputs into the soil were too low in LPJmL 5. LPJmL-CSR showed smaller deviations from observations and an adequate representation of the trends (Sect. 3.1.1). The increased soil carbon input had three reasons. First, the trade-off between SLA and leaf longevity lead to higher turnover rates and in turn higher litterfall compared to LPJmL 5. Second, accounting for mortality explicitly constituted an additional input into the litter layer. Third, our simulation included manure application which provided an additional carbon input into the system.

The community composition showed some intra-annual variability, and higher shares of the marginal PFTs at the end and the beginning of a growing season in the unfertilised and fertilised scenarios (Sect. 3.4.1). The S-PFT gained higher shares in the unfertilised scenario showing an advantage of the S over the R-PFT despite the fact that strong nitrogen stress was avoided through biological nitrogen fixation. In contrast, if nitrogen stress was removed entirely, the S-PFT lost its advantage and the R-PFT could increase its share. After the first cut, these shares of the S- and R-PFT became smaller because a cut is a disturbance that directly removes part of the above-ground biomass. One strategy to cope with this is grazing (or in this case mowing) tolerance (Briske, 1986; Stuart-Hill and Mentis, 1982), which requires fast regrowth of the leaves to compensate for the removed biomass as is typical for a C strategist (Grime, 1977), and Sect. 4.2.2).

### 4.1.2 Hot steppe

For the hot steppe, LPJmL 5 performed better for SOC while LPJmL-CSR performed better for forage offtake. We identified several reasons for these inconsistent results: First, LPJmL does not distinguish between leaves of different age classes and therefore not between alive, senescent and moribund tissue (Schaphoff et al., 2018). All tissue is either alive and associated with the plant or moribund and part of the litter layer. However, observed forage offtake also included senescent biomass (Munjonji et al., 2020). This predisposed the model to underestimate forage offtake when accounting for realistic turnover rates, which was observed in the low biomass values simulated in LPJmL-CSR. Second, litter decomposition is a function of soil moisture, temperature and litter composition (Schaphoff et al., 2018). However, the PFTs do not differ in their persistence of the litter which is the case for different plant species and across ecological strategies (Brovkin et al., 2012). Considering this may help to improve the simulation of SOC dynamics in the future. Third, the vegetation was described as an open thornbush savanna (Acocks, 1994; Scheiter et al., 2023) which includes a woody component. However, in LPJmL managed grassland vegetation does not include bushes or trees and therefore only partially represents the observed community.

The S-PFT was dominant in the grazed and ungrazed scenario, while the remainder of the above-ground biomass was contributed by different PFTs depending on the scenario (Sect. 3.2.1). The dominance of the S-PFT independent of grazing is plausible considering the pronounced dry vs. wet season dynamics at the site that impose water stress (Scheiter et al., 2023)

and potentially also nitrogen stress. The R-PFT was more tolerant towards grazing disturbances and gained dominance in the grazed scenario, replacing the C-PFT which had a lower ability to deal with disturbance.

Removing the water limitation led to an increase of forage offtake and SOC which can be expected when removing the main resource limitation. However, the majority of the SOC increase occurred in the first two years after the start of irrigation which is not realistic. This can be explained by the missing representation of senescent tissue in combination with the adaptation of the community composition: Removing the water limitation led to a strong increase of leaf biomass, which was substantially higher than the feed demand of the simulated grazing intensity and increased the input to the litter layer. Furthermore, the share of the R- and C-PFT which have a lower leaf longevity than the S-PFT increased leading to faster inputs into the litter layer. After one to two years the community composition reached a new equilibrium and inputs into the litter layer decreased. Introducing senescent tissue would increase the competition for light due to self-shading effects (Zimmermann et al., 2010) and likely slow down this transition.

In addition, irrigation led to a shift in the community composition (Sect. 3.4.2) and an increase in leaf biomass to which the C- and R-PFT together contributed more than the S-PFT (Sect. 3.3.2). We cannot determine whether or not increases under irrigation would be lower for a S-PFT monoculture which does not contain other ecological strategies but strongly suspect so.

In both the ungrazed and the grazed scenario, the community transitioned from strongly S-dominated to a community with higher shares of the C- and R-PFT that was still S-dominated. According to the CSR theory, this type of community emerges in somewhat stressed and disturbed habitats (Grime, 1977). While this case can easily be made for the grazed scenario, where the disturbance is caused by the animals, the ungrazed scenario does not include such a clear disturbance. The success of both the C- and the R-PFT is likely determined by the similarity of their $SLA$, $k_{beer}$ and $lmro$ which become more important compared to $E_{max}$ and $g_{min}$ if there is no water limitation. Potentially larger differences in these parameter would lead to the success of one of the two instead.

Less than two years is a very fast transition and while the shares of the leaf biomass seem to have reached a new equilibrium after one or two years of irrigation, it is likely that the soil carbon and nitrogen pools are not in equilibrium yet. This is especially interesting when considering that the overall increase in leaf biomass may promote litterfall and the formation of inorganic nitrogen. This in turn may lead to reduced nitrogen limitation and additional changes in the community composition. Furthermore, biological nitrogen fixation is dependent on soil moisture and may therefore also contribute to decreasing nitrogen stress under irrigation. However, irrigation also leads to increased leaching and could therefore also decrease inorganic nitrogen availability. Future analysis considering longer time scales may help to identify intermediate and final transition states.

Regardless of the equilibrium state of the transition, its velocity is likely overestimated by LPJmL for two reasons. First, the C- and R-PFT can establish quickly despite their limited presence before the onset of irrigation because LPJmL does not simulate a seed bank which would in reality be small at least for the C-PFT limiting its establishment. Second, in reality growth of established individuals is limited and a transition as simulated is strongly controlled by reproduction and dispersal, which slow down population biomass increase. In LPJmL, already established individuals continue to grow and the population biomass increases even without additional establishment.

### 4.1.3 Cold steppe

LPJmL 5 underestimated the observed forage offtake of the cold steppe because the feed demand, which was originally designed to represent large cattle (Rolinski et al., 2018), was scaled down linearly with animal body weight. This led to an unrealistically low feed demand because the feed demand body weight relationship is not linear but follows a power law (Cordova et al., 1978). Our new calculation of feed demand (appendix A5) led to a higher feed demand and forage offtake simulations were improved for low and high grazing intensities.

Under observed conditions, the high grazing intensity severely reduced above-ground biomass and feed demand was not met in all years except the year directly after the increase in stocking density indicating overgrazing. The reduced biomass availability was also observed by Schönbach et al. (2012) in their field experiment. Additionally, LPJmL simulates a different community composition compared to the low grazing intensity. The relative share of the C- and to some extent also the R-PFT is higher for the high grazing intensity (Fig. SI 10 b and 13 h) because such strategies are better suited to tolerate grazing.

During and after the grazing period, the C- and R-PFT had a higher share of the community above-ground biomass. Both these PFTs can regrow faster and invest more into above-ground biomass which gave them an advantage over the S-PFT under grazing. In addition to the observed environmental conditions, we simulated two scenarios where we removed the water and nitrogen limitations separately. Removing the nitrogen limitation barely affected biomass availability, and forage offtake was similar compared to the rainfed scenario (Sect. 3.3.3). The additional soil nitrogen could not be utilised by the plants because water was the main limiting factor (Li et al., 2011; Bai et al., 2004). In contrast, removing the water limitation led to an increase in leaf biomass, and forage offtake met the demand in all years even for the high grazing intensity. This is in line with irrigation and fertilisation experiments conducted in the cold steppe (Li et al., 2011) and other sites with similar conditions (e.g. Shi et al., 2022). Contrary to the results of Li et al. (2011), who reported a lower share of annuals and bi-annuals – that are more likely C than S-strategists – in the rainfed treatments, the S-PFT was dominant in the irrigated scenarios. One reason for this could be that LPJmL does not simulate seedbanks, which play a major role for the establishment and success of the annuals and bi-annuals (Thompson, 1987; Brown and Venable, 1986). Instead, LPJmL simulates establishment of additional seedlings dependent on available space assuming that resources for reproduction are available at any time and not dependent on past investments into seed production.

Despite the fact that we did not have separate data on SOC under the two grazing intensities, our results showed a lower SOC storage for the high grazing intensity typical for overgrazed steppes (e.g. Wiesmeier et al., 2012) compared to the low grazing intensity which constituted the typical livestock density for the region (Hoffmann et al., 2016). Wiesmeier et al. (2012) investigated the effect of high grazing intensities on the SOC, observing significant SOC losses within three years of increased grazing, which is in line with our simulation results. Fertilisation had no effect on SOC because leaf biomass and in turn carbon inputs into the soil did not increase. In contrast, irrigation led to an increase of SOC, which was stronger for the low grazing intensity. This is likely because more biomass was produced and the surplus of the feed demand was not removed but contributes to the litter layer. However, these gains would not justify the effort that would be necessary to irrigate large areas.

Removing the water limitation led to a transition from S-dominated to a S monoculture community under both grazing intensities (Sect. 3.4.2). Since the site was still severely nutrient-limited and exposed to low temperatures, it seems that an S strategy remained advantageous. Furthermore, the S-PFT showed trait values associated with large investments in roots and more persistent root tissue (Sect. 3.2.1) which provides a likely explanation for its increased dominance: It had an advantage in the competition for the additional water. Similar to the hot steppe, it is possible, that our time frame is too short for the soil pools to have reached a new equilibrium. As described in Sect. 4.1.2, irrigation alone already affects processes that could increase nitrogen supply by biological nitrogen fixation and litterfall, but also decrease it by leaching. Both biological nitrogen fixation and mineralisation are dependent on soil moisture as well as on temperature which is low in the cold steppe limiting the increase of inorganic nitrogen. Therefore, it is possible that only an intermediate state emerges during our simulation period. Especially when also considering the increased leaching, we expect that the cold steppe is still nitrogen limited under irrigation, therefore combining irrigation with fertilisation could further reduce nitrogen limitation leading to increased productivity and changes in the community composition. However, the leaf biomass increase may also be limited by higher maintenance respiration which is connected to leaf nitrogen content. Additional analysis is needed to enhance the understanding of these complex interactions.

## 4.2 Stress and disturbance gradients across sites and management

### 4.2.1 Across sites

We used a Bayesian calibration method to find suitable parameter values of eight parameters assigned to two trade-off dimensions for the new PFTs. Due to lacking data on starting values and ranges for the three new PFTs, we used the same ranges and starting values for each PFT but prescribed an order of the parameters. Within a site and management scenario, the prescribed hierarchy for specific parameters also predefined the ranking of the PFTs along the stress and disturbance gradients. Across sites and management we did not constrain the PFTs to positions within the two dimensions. Theoretically, all PFTs of the temperate grassland could have been associated with a more conservative strategy for the stress gradient compared to the PFTs of the hot steppe. However, while there were some differences between the sites and management, on average the C- and R-PFTs occupied a more resource exploitative position for the stress gradient and the S-PFTs a more conservative one (Sect. 3.2.2). Similarly, for the disturbance gradient the C- and S-PFTs occupied a position associated with less but larger offspring and a larger stature compared to the R-PFT. It is an emergent property of the model, that not only the relative position of the PFTs of a site and management scenario determined community composition but also the overall positions along the stress and disturbance gradients (which we derived from the *global spectrum of plant form and function* Díaz et al. 2016) were important. Our experiences from these three sites showed similar strategies independent of environmental conditions, indicating that LPJmL-CSR is capable of reproducing the empirically derived trade-offs associated with the *global spectrum of plant form and function* (Díaz et al., 2016). However, LPJmL-CSR will benefit from additional testing on larger scales in the future.

#### 4.2.2 Across management

While missing processes such as the representation of seedbanks as at the hot steppe (Sect. 4.1.2) and poor data as at the cold steppe (Sect. 4.1.3) may have led to biased model dynamics to some extent, we clearly demonstrated the importance of representing different ecological strategies.

The calibration selected different strategies along the stress and disturbance gradients for the different management intensities (Sect. 3.2.1), which were related to changes in resource limitations or disturbance level: In LPJmL-CSR, a change in

resource availability does only change the conditions for the establishment of a community but does not directly affect the established vegetation (changes in environmental filters Bazzaz, 1991; Woodward and Diament, 1991). In reality however, a change in resource availability may also increase the mortality for specific strategy types affecting the already established community as well. In temperate grasslands, manure application increases N supply and reduces the number of available niches that can be occupied by different ecological strategies. In the unfertilised experiment, species could satisfy their N demand

through two different strategies: Competition for the limited resource in the soil or biological N fixation (BNF). In the fertilised experiment, only the first strategy was advantageous as BNF creates additional costs. In the field experiment this was evidenced through the substantially different amount of legumes between the two experiments (Reinsch et al., 2020). In the model, N-fixing and non-N-fixing species are both collated within each PFT. Therefore, in the unfertilised scenario, a PFT had to apply a strategy combining N uptake and fixation, whereas it could focus on N uptake in the fertilised scenario. Since we

calibrated the unfertilised and fertilised scenarios separately using the same data for C-, S- and R-PFT cover, the difference in strategy between the two scenarios is expressed through the different position of the PFTs along the stress and disturbance gradients: Higher investments into below-ground biomass ($lmro$) provide an advantage in the competition for plant available nitrogen (Johnson and Biondini, 2001). In the model this led to a reduced need of fixing additional nitrogen and in turn a reduction of the investment costs associated with biological nitrogen fixation. (Sect. 3.2.1).

In contrast to resource availability, a disturbance directly affects the vegetation. In the case of grazing, it also influences resource availability indirectly through removal of nutrients from and spatial redistribution within the system (Liu et al., 2023; Chuan et al., 2018; Wan et al., 2015). In LPJmL, the grazing of the animals at the steppe sites constituted a direct reduction of leaf biomass proportional to the cover of each PFT (Rolinski et al., 2018). Under intensive grazing, strategies of grazing tolerance or avoidance are essential (Briske, 1986; Stuart-Hill and Mentis, 1982). While grazing tolerance is mainly associated with

fast regrowth (Briske 1986; Hyder 1972, stress gradient), grazing avoidance strategies can operate in time and space. Grazing avoidance in time is possible through the completion of the life cycle between grazing intervals (Noy-Meir 1990,disturbance gradient). Grazing avoidance in space is contingent on reducing plant size (Rechenthin, 1956; Branson, 1953). However, since plant size is not explicitly represented in LPJmL, we do not discuss this strategy further (Sect. 2.3.3). In the hot steppe, we simulated a daily grazing system, which makes grazing avoidance through the life cycle impossible and the PFTs had to follow

a grazing-tolerance strategy. This was expressed through changes in the stress gradient: All PFTs increased their investment into above-ground biomass and faster tissue growth (Sect. 3.2.1). Because LPJmL does not account for differences in the

palatability of different strategy types the parameterization could not select for such likely successful strategies leading to a potentially biased community composition.

At the cold steppe site, grazing only happened during the growing season and both grazing tolerance and avoidance could be useful strategies. However, grazing avoidance in time, which is the only type simulated by LPJmL will not be successful as it would mean shifting biomass production to the non-growing season where the environmental conditions do not allow growth. Still, between the extensive and intensive grazing scenario the differences between the PFTs in both dimensions do support different strategy adjustments (Sect. 3.2.1). The C-PFT increased its investment into above-ground biomass to tolerate grazing, while the S- and R-PFT did not show any adjustment. However, since the high grazing pressure caused degradation of the above-ground biomass, differences between the two management scenarios do not only reflect different strategies to deal with the disturbance, but also for survival outside the grazed period. As such, all PFTs constructed long living tissue to survive unproductive conditions outside the growing season in the intensive grazing scenario. This was not necessary in the extensive grazing scenario because the PFTs retained substantial above-ground biomass at the end of the growing season and did not need to be as resource conservative.

## 4.3    Limitations and further need for research

The representation of different CSR-strategies is a new feature in LPJmL, a model, which is mainly used at large to global spatial scales. Past explorations have pointed out the difficulties of adding new PFTs to DGVMs in general (Yang et al., 2015) and also to LPJmL (Wirth et al., 2021). We therefore decided to only add a small number of PFTs which should represent the three main CSR-strategies and no sub-strategies. We used expert estimates to determine the shares of the three strategies. These three strategy shares sum up to 100% and also encompass species that would be added to a sub-strategy in a less coarse approach. Consequently, our results show a very simplified representation of the different strategies within a community and across sites, which might be better represented using a small scale model such as IBC GRASS (May et al., 2009) or GRASSMIND (Taubert et al., 2020a, b, 2012). However, large-scale applications also benefit from the inclusion of universally applicable trade-offs between different ecological strategies and the improved representation of productivity changes.

Furthermore, we reduced the trade-offs between C-, S- and R-strategists to fit into two dimensions and used a limited amount of parameters to express these. While this simplification was necessary, this also means that we do not represent all effects, advantages and trade-offs of functional diversity. However, as LPJmL is a global model, our aim was not to optimise performance for specific sites, but to evaluate and test an approach, which can easily be applied at the global scale without the need of a global data set on community composition of grasslands. Keeping this in mind, and considering the difficulties of adding PFTs to a DGVM as well as the global heritage of the model, we find that representing even only the three main CSR-strategies constitutes a major improvement of LPJmL.

Generally, the approach of using a small number of PFTs with a fixed set of parameters has been criticised (Quillet et al., 2010) leading to the development of next generation DGVMs that apply an individual based approach such as LPJmL-FIT (Sakschewski et al., 2015) or aDGVM (Scheiter et al., 2013). These models simulate the competition between individual

plants for which parameter values are drawn from predefined ranges upon establishment. Given sufficient time, only successful strategies will survive. Such models provide a much more nuanced representation of functional diversity compared to classic DGVMs with their coarse division into fixed PFTs but are also computationally substantially more expensive because of the high number of individuals for which all processes have to be calculated. Past studies have therefore often focused on specific

regions such as the Amazon rainforest (Sakschewski et al., 2015), European forests (Thonicke et al., 2020) or South African semi-arid rangelands (Pfeiffer et al., 2019). In contrast, classic DGVMs are still widely applied on the global scale for example to calculate the global carbon budget (Friedlingstein et al., 2022) and we see the need to continue their development for the foreseeable future. Combining our approach of distinguishing between PFTs that follow the main strategies of the CSR theory with an individual based approach making use of the full parameter range instead of single points provides an interesting

opportunity for future research of diverse grasslands.

For this study, we only assessed three sites at which our approach worked well. We did not include a site dominated by R-strategists since this is not common for managed grasslands, but also did not include CS and CSR habitats which are typical for unfertilised and fertilised pastures, respectively (Grime, 1974). Additional research including these intermediate habitats might provide more insight on the newly-implemented strategies and trade-offs. While separate calibrations are feasible for a

small number of sites and scenarios, for large scale or global assessments the lack of data and the computational requirement for the calibration make a site-specific calibration infeasible. However, using a more efficient calibration method and remote sensing data instead of on-site experiments can be used to derive a set of PFTs which are representative of the entire globe or at least climatic regions. For LPJmL, a genetic optimisation algorithm has been used to successfully calibrate the phenology (Forkel et al., 2014) and vegetation dynamics (Forkel et al., 2019) of natural ecosystems. Following this approach, we believe

it is possible to identify C-, S- and R-PFTs for the tropical, temperate and polar regions ending up with nine PFTs in total.

In LPJmL, herbaceous plants are represented as average individuals of a number of different PFTs, without an explicit representation of geometry. Therefore, we used the light extinction coefficient as a proxy for stature, assuming that small stature plants would be less competitive for light. We here deviate from the common interpretation of the light extinction coefficient, which is usually defined as the light absorption of a layer of leaves. However, as explained in Sect. 2.4, LPJmL represents the

entire vegetation as a single layer and we therefore define the light extinction coefficient not for a single leaf but a stack of leaves. Taller plants likely produce more layers of leaves corresponding to a larger stack and a thicker vegetation layer with a higher light extinction. However, thickness of the vegetation layer is not explicitly represented in LPJmL and we represent the described differences by using lower light extinction coefficients for small stature plants for which we assume a lower thickness of the vegetation layer and higher light extinction coefficients for large stature plants. However, this is not sufficient

to simulate grazing avoidance in space (Sect. 4.2.2) and an explicit representation of plant height and area could further improve the representation of ecological strategies (Wirth et al., 2021). Furthermore, the coexistence of trees and grass species, which is typical for savanna sites, is not implemented in the LPJmL model. However, this is crucial to adequately represent such ecosystems (Rolinski et al., 2021) and should be a focus of future model development. Another important aspect in savanna and other dryland ecosystems is the distinction between annual and perennial plants. In LPJmL, this distinction is not explicitly

made. While the R-PFT has a higher replacement rate of average individuals, it is not constrained to a specific growing season,

after which it is completely killed to be reestablished the following growing season. Incorporating this distinction into the model is an option to add additional functional diversity and will likely improve model results.

LPJmL-CSR only represents age mortality, i.e. the effects of mortality from other causes such as frost, heat and embolism are not represented. Especially under changing climatic conditions, specific strategy types may show increased mortality and lose their advantage to the advantage of other strategy types. Including additional causes of mortality may introduce additional trade-offs and enhance the differentiation between strategy types.

Plant species have adapted to grazers in manifold ways, one of which is grazing avoidance by being less or even unpalatable. This is a successful strategy in grazing systems because in contrast to mowing, which is indiscriminate, grazing animals show preferences for plants with a higher palatability (Tribe and Gordon, 1950; Deg, 1954). Selective grazing and grazing avoidance through palatability are currently not represented in LPJmL but can have a strong effect on the community composition (Parsons et al., 1994; Newman et al., 1995). Including preferences for example for high SLA PFTs may improve simulation results further. Additionally, LPJmL-CSR does not consider mechanical stress caused by trampling of animals and potential strategy dependent damage. Incorporating this may add another dimension of stress to distinguish different PFTs.

Data coverage for the temperate grassland site was good and observations were available for multiple years and with sufficient replicates. For the two steppe sites, data on SOC were scarce. Especially data on trends and equilibria under specific management conditions might promote further improvement of the model and help with the parameterisation of new PFTs. We based our parameterisation of the new PFTs on expert estimates for the C-, S-, and R-PFT cover. While we are confident that these estimates were adequate, data on a small number of traits would be sufficient to calculate the shares for each PFT following Pierce et al. (2013), and we would like to encourage including such data as a standard in sampling procedures for future experiments.

The scenarios we examined here only involved the reduction of stress by removing either water or nitrogen limitations. Additional insight might be gained from doing the opposite and imposing additional limitations or looking into gradual changes of environmental conditions.

## 5 Conclusions

We presented a new approach for large-scale models and DGVMs to simulate the three main CSR-strategies of managed grassland PFTs. In addition to improving the simulation of forage offtake or leaf biomass and SOC at three different sites, the approach successfully simulated the dynamic community composition at these sites and reproduced the *spectrum of plant form and function* (Díaz et al., 2016). This is a major improvement allowing to explicitly assess how the presence or absence of specific plant strategies affects ecosystem functioning and thus ecosystem service provision of managed grasslands. Using this new feature, scenarios for projections of forage offtake, leaf biomass and SOC under climate change can be complemented with different constraints on the adaptive capacity of the vegetation. Such projections can provide a range of future grassland productivity as decision-support for policy-makers. To further improve these projections, extending the sites by considering habitats with intermediate environmental conditions as well as the scenarios by including additional resource limitations (e.g.

droughts) or gradual changes of environmental conditions (e.g. temperature increase) could be useful to gain additional insights on the model and to study the complex interactions of climate change, management and functional diversity.

*Code and data availability.* The source code is publicly available under the GNU AGPL version 3 license. An exact version of the code described here and the data used to create the figures is archived under https://zenodo.org/records/10217244 (Wirth et al., 2023).

## Appendix A: Model description

We provided a qualitative description of new model development in the main text (Sect. 2.3), for which we supplement the underlying equations and additional minor developments here.

### A1  Water uptake

To make resource uptake of different resources dependent on different plant traits, we adapted the water uptake routine of the LPJmL model. Available soil water is now distributed between PFTs dependent on their root carbon ($C_{root,PFT}$) and a PFT-specific parameter ($k_{root,PFT}$), which is used as a substitute for information on root functional traits (e.g. branching of the root network, amount of fine roots, number of root tips). These traits cannot directly be incorporated because either the simplified representation of below-ground plant organs hinders their representation or data are not sufficiently available.

$$f_{root,PFT} = w_{PFT} \cdot (1 - exp(-k_{root,PFT} \cdot C_{root,PFT})) \tag{A1}$$

Eq. (A1-A3) describe an exponential function which follows the approach used for the calculation of the foliage projective cover (FPC, see (Schaphoff et al., 2018)), which was used to distribute water between PFTs in previous model versions.

$$w_{PFT} = (1 - exp(-k_{root,PFT} \cdot \sum_{i}^{Number\ of\ PFTs} C_{root,i})) \cdot f_{root,sum}^{-1} \tag{A2}$$

Each PFT's access to plant available soil water ($f_{root,PFT}$) is weighted using Eq. (A2). Here $w_{PFT}$ is calculated as the fraction of the respective PFT's potential access to the plant available soil water if the entire community root carbon would belong to it and the sum of all PFTs' access to plant available soil water if now weighting would be applied (Eq. A3).

$$f_{root,sum} = \sum_{i}^{Number\ of\ PFTs} 1 - exp(-k_{root,i} \cdot C_{root,i}) \tag{A3}$$

### A2  The leaf economic spectrum

To incorporate the trade-offs associated with the LES, we implemented a power law relationship between SLA and leaf longevity (LL) described by Eq. A4

$$LL = a \cdot SLA^b \cdot 12^{-1}, \tag{A4}$$

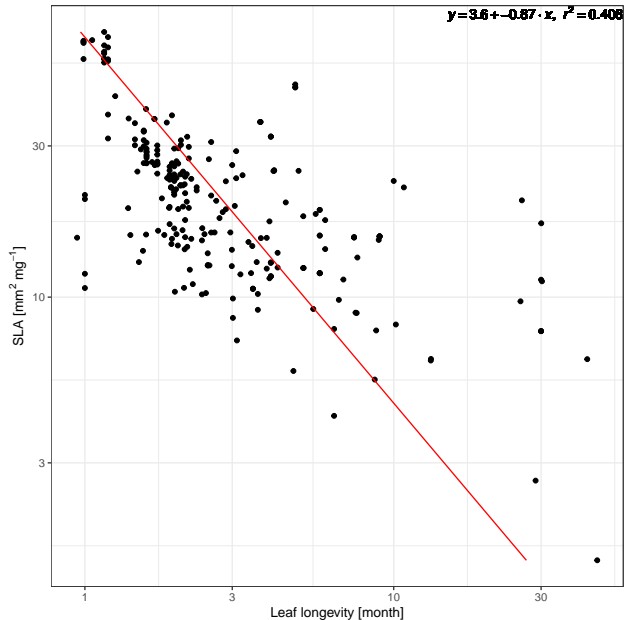

**Figure A1.** Linear regression of log SLA and log LL using trait data for herbaceous species from the TRY database.

where $a = 36.3753$ and $b = -0.85384$. $a$ and $b$ were derived from a regression (A1) using trait data for SLA and LL retrieved
from the TRY database (Boenisch and Kattge, 2018; Kattge et al., 2011). A detailed listing of the data sets used is provided
in SI Tab. 1. The leaf turnover rate is calculated as the inverse of the leaf longevity ($\tau_{leaf} = 1/LL$) and is linearly related to
root turnover ($\tau_{root} = k \cdot \tau_{leaf}$ with $k = 2$) assuming that the LES and the conservation gradient (Bergmann et al., 2020) of
the root economic space are aligned (Weigelt et al., 2021). Plant biomass is transferred to the litter pools each day only if
one of two conditions is met: Under grazing, we assume that depending on the stocking density, leaf tissue is grazed before it
becomes senescent, and we define a threshold ($\xi_{leaf} = 5 \mathrm{gCm}^{-2}$) for leaf biomass below which no senescent tissue for turnover
is available; For mowing, we assume that senescent leaf biomass has to build up again after a mowing event, and we define a
threshold for the leaf-to-root mass ratio ($\xi_{lmtorm} = 0.7 \cdot lmtorm_{opt}$) beyond which senescent tissue is built up again.

### A3 Reproduction and mortality

To improve the representation of different reproduction strategies and lifecycles, we adapted the establishment and mortality
routine of the model. Both establishment and mortality are executed daily. In the new establishment routine, the number of
average individuals ($n_{ind}$) and the carbon ($C_{ind,pool}$) and nitrogen ($N_{ind,pool}$) pools of the leaves and roots for the average

individuals are increased following Eq. (A5-A7).

$$\Delta n_{ind,PFT} = k_{est,PFT} \cdot 365^{-2} \cdot$$
$$(1 - exp(-5 \cdot (1 - FPC_{sum}))) \cdot$$
$$(1 - FPC_{sum}) \cdot (k_{est,PFT} \cdot$$
$$(\sum_{i}^{Number \ of \ PFTs} k_{est,i})^{-1}) \tag{A5}$$

$$\Delta C_{ind,PFT} =$$
$$(C_{seedling,leaf,PFT} + C_{seedling,root,PFT}) \cdot \Delta n_{ind,PFT} \tag{A6}$$

$$\Delta N_{ind,PFT} = \Delta C_{ind,PFT} \cdot NC_{ratio,leaf,PFT} \tag{A7}$$

Here, $k_{est}$ is the PFT-specific establishment rate, $FPC_{sum} = \sum_{i}^{Number \ of \ PFTs} FPC_i$ is the sum of the $FPC$ of all PFTs, $C_{seedling,pool}$ is the PFT-specific leaf and root pool size of a seedling and $NC_{ratio,leaf,PFT}$ is the PFT-specific nitrogen to carbon ratio. The new individual properties are calculated following Eq. (A8) and (A9)

$$C_{ind,pool,PFT} = (C_{ind,pool,PFT} \cdot n_{ind,PFT} +$$
$$C_{seedling,pool,PFT} \cdot \Delta n_{ind,PFT}) \cdot$$
$$(n_{ind,PFT} \cdot \Delta n_{ind,PFT})^{-1} \tag{A8}$$

$$N_{ind,pool,PFT} = (N_{ind,pool,PFT} \cdot n_{ind,PFT} +$$
$$C_{seedling,pool,PFT} \cdot NC_{ratio,pool,PFT} \cdot \Delta n_{ind,PFT}) \cdot$$
$$(n_{ind,PFT} \cdot \Delta n_{ind,PFT})^{-1} \tag{A9}$$

Mortality was implemented as an age mortality using the concept of growth efficiencies (Waring and Schlesinger, 1985; Waring, 1983) using Eq. (A10)

$$mort_{PFT} = mort_{max,PFT} \cdot 365^{-1} \cdot$$
$$(1 + k_{mort} \cdot \Delta bm \cdot C_{ind,leaf,PFT}^{-1} \cdot$$
$$SLA_{PFT}^{-1})^{-1}, \tag{A10}$$

with

$$\Delta bm = C_{inc,PFT} \cdot n_{ind,PFT}^{-1} - C_{turn,PFT} \tag{A11}$$

where $C_{turn,PFT}$ is the amount of carbon that was transferred to the litter pool since the last allocation and $C_{inc,PFT}$ is the biomass increment from photosynthesis since the last allocation. The growth efficiency $\Delta bm \cdot C_{ind,leaf,PFT}^{-1}$ is the ratio of the net carbon change and the carbon stock of the leaves, which is lower for old plants. The SLA influences the maximum age of the different strategies assuming that plants with a low SLA and faster metabolism reach a lower age compared to high SLA plants. The number of average individuals is decreased following Eq. (A12).

$$n_{ind,PFT} = n_{ind,PFT} \cdot (1 - mort_{PFT}) \tag{A12}$$

In grasslands with a high growth efficiency and frequent defoliation establishment may lead to a continuous increase of the number of average individuals. To avoid numerical errors that could results from this, we prohibit the number of average individuals to exceed $250 \, \text{indm}^{-2}$.

## A4 Biological nitrogen fixation

Symbiotic biological nitrogen fixation (BNF) is an important source, especially in unfertilised grassland systems. We implemented an approach adapted from published models of grain legumes (e.g. LPJ-GUESS, CROPGRO, EPIC, APSIM see Ma et al., 2022; Liu et al., 2011), which considers the potential N fixation rate, the soil temperature and the soil water status. The consideration of the growth stage had to be omitted because LPJmL represents herbaceous vegetation using only leaves and roots, not allowing for a determination of growth stages. The nitrogen fixation rate $N_{fix}$ is calculated using Eq. (A13)

$$N_{fix} = N_{fix,pot} \cdot f_T \cdot f_W \tag{A13}$$

with $N_{fix,pot} = 0.1 \, \text{gNm}^{-2}\text{day}^{-1}$ (Yu and Zhuang, 2020). The soil temperature limitation is modeled linearly outside the optimal temperature range (Eq. A14):

$$f_T = \begin{cases} 0, & \text{if } T_{soil} < T_{min} \text{ or } T_{soil} > T_{max} \\ \frac{T_{soil} - T_{min}}{T_{opt,low} - T_{min}}, & \text{if } T_{min} \leq T_{soil} < T_{opt,low} \\ 1, & \text{if } T_{opt,low} \leq T_{soil} \leq T_{opt,high} \\ \frac{T_{max} - T_{soil}}{T_{max} - T_{opt,high}}, & \text{if } T_{opt,high} < T_{soil} \leq T_{max} \end{cases} \tag{A14}$$

with $T_{min} = 0.5$, $T_{op,low} = 18.0$, $T_{opt,high} = 35.0$ and $T_{max} = 45.0$ (Yu and Zhuang, 2020). The soil water limitation is linearly dependent on the relative soil water content $SWC$ (Eq. A15):

$$f_W = \begin{cases} 0, & \text{if } SWC \leq SWC_{min} \\ \varphi_1 + SWC \cdot \varphi_2, & \text{if } SWC_{low} < SWC < SWC_{high} \\ 1, & \text{if } SWC \geq SWC_{high} \end{cases} \tag{A15}$$

with $SWC_{low} = 0$, $SWC_{high} = 0.5$, $\varphi_1 = 0$ and $\varphi_2 = 2.0$ (Yu and Zhuang, 2020).

BNF only happens if the nitrogen uptake from other sources is insufficient and the net primary productivity (NPP) is larger than

zero. The costs of BNF are set at a moderate constant value of 6 gCgN$^{-1}$ (Boote et al., 2009; Kaschuk et al., 2009; Patterson and Larue, 1983; Ryle et al., 1979). If the costs exceed the maximum costs which are set at 50% of the NPP (Kull, 2002) the

985 nitrogen fixation is reduced to the amount achievable with the maximum costs. A full description of the original approach is provided in Ma et al. (2022). While in reality, biological nitrogen fixation is a feature restricted to legume species, in LPJmL we decided to not distinguish in fixing and non-fixing PFTs to keep the number of PFTs as small as possible. This is reasonable because a PFT can be representative of multiple species and will only fix additional nitrogen if its demand cannot be fulfilled by other sources of nitrogen uptake and if its NPP is sufficient. One could say, the PFT has the ability to fix nitrogen only if

needed comparable to a community containing legumes only if they are advantageous.

## A5 Feed demand

We implemented a relationship between metabolic body weight ($MBW$) and feed demand following (Cordova et al., 1978). This is the same relationship used to calculate the feed demand in LPJmL 5, but we replaced the constant $650 \ \text{kg} \cdot \text{animal}^{-1}$ with a parameter $BW$ (Eq.A16) while preserving $intake_{MBW} = 31.07$ (Rolinski et al., 2018).

$$feed \ demand = BW^{0.75} \cdot intake_{MBW} \tag{A16}$$

## A6 MSE components

We calculated the mean square error and it components, the bias, phase and variances following A17 to A20. $x$ and $y$ are the time-series of simulated and observed values of a variable, $\overline{x}$ and $\overline{y}$ are the time-series mean, $\sigma_x, y$ is the time-series standard deviation and $N$ is the number of values in the time-series.

$$MSE = (\overline{x-y})^2 \tag{A17}$$

$$MSE_{Bias} = (\overline{x} - \overline{y})^2 \tag{A18}$$

$$MSE_{Phase} = 2 \cdot (\frac{N-1}{N} \cdot \sigma_x \cdot \sigma_y \cdot (1 - corr(x,y))^2 \tag{A19}$$

$$MSE_{Variance} = ((\frac{N-1}{N}) \cdot (\sigma_x - \sigma_y))^2 \tag{A20}$$

*Author contributions.* SBW, SR and AP designed the study in discussion with AL, BT, FT and CM. SBW designed and conducted the model

implementation with inputs from SR and AP. WvB and SSch contributed to general model development and evaluation. SBW conducted the model simulations and wrote the original draft of the manuscript. All authors discussed the simulation results and the original draft. AL,

BT, SR, CM, AP, SSch, FT, and KB reviewed and edited the manuscript. AL, KB, AP and FT contributed unpublished empirical data from grassland sites and experiments. SR, AP and FT supervised this study.

*Competing interests.* At least one of the (co-)authors is a member of the editorial board of Biogeosciences.

*Acknowledgements.* SBW acknowledges financial support from the Evangelisches Studienwerk Villigst foundation, under the research program: "Third Ways of Feeding The World", the Global Commons Stewardship (GCS) project funded by the University of Tokyo / Institute for Future Initiatives and the German Federal Ministry for Education and Research (BMBF) within the projects CLIMASTEPPE (grant no. 01DJ8012), EXIMO (grant no. 01LP1903D) and ABCDR (grant no. 01LS2105A). SR acknowledges financial support from the BMBF within the projects CLIMASTEPPE (grant no. 01DJ8012), AGrEc (grant no. 01DG21039), MAPPY (grant no. 01LS1903A) and ABCDR
(grant no. 01LS2105A). BT acknowledges funding by the BMBF in the framework of the ValuGaps project (grant no. 01UT2103C). AL and KB acknowledge funding by the BMBF within projects of the SPACES initiative ("Limpopo Living Landscapes" project – grant no. 01LL1304D; "SALLnet" grant no. 01LL1802C). We thank Stefan Lange for providing the GSWP3-ERA5 data set and Edwin Mudongo, Vincent Mokoka, and the Risk and Vulnerability Center at the University of Limpopo, South Africa, for data acquisition at the hot steppe site.

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
