# Peer review of "Connecting CSR theory and LPJmL 5 to assess the role of environmental conditions, management and functional diversity for grassland ecosystem functions"

_Biogeosciences, 2023_

## Author Comment (AC1)

**General evaluation of the research paper**

The paper presented by the authors addresses a very relevant and important topic in the field of DGVM model development. For far too long, the representation of grasses and the herbaceous layer have been given far too little focus in most DGVMs with respect to structural and functional diversity. Only recently, development of more detailed grass layer representations in DGVMs are starting to emerge but compared to tree-layer representation this work is still at a comparatively early stage of development. Grassland ecosystems and savannas cover a substantial fraction of the land surface and provide important ecosystem functions and services to a multitude of people while simultaneously being threatened by the effects of climate change and resource over-exploitation. Therefore, developing vegetation models that are capable of representing within-grass layer dynamics, diversity and processes is crucial to assess the impact of different management strategies and environmental change. I therefore deem the paper a relevant and important scientific contribution.

The CSR theory is a widely known concept and therefore a valid approach to implement functional diversity and trade-offs within the herbaceous layer of the model. One may question whether the implementation in its current form using a Bayesian calibration method to parameterize the new PFTs for three specific sites can be generalized for large-scale application, but in the given context of the study, the approach seems sound and justified to me. The shown results in many cases match ecological expectations and improve results compared to the old model version, further corroborating the chosen approach.

The paper is well-written and clearly structured. I therefore recommend publication pending minor revisions and clarifications detailed below.

We cordially thank the reviewer for their thorough and constructive feedback as well as the positive evaluation of our manuscript.

Below we provide a response to all detailed comments including proposals to achieve the suggested improvements.

**Detailed comments**

**Introduction:**

line(s) 36/37: You might also add the role of atmospheric $CO_2$-concentration. $CO_2$-fertilization effects can shift the competitive balance in grassland communities in locations where both C3 and C4 grasses are present.

We agree, even though we do not look into the effects of changing CO2 concentrations it should be a part of this overview and we will add a brief description of its role.

line(s) 42: "high temperatures can lead to an increase of microbial decomposition". Only in combination with sufficient moisture. In arid regions, decomposition comes more or less to a stand-still during the dry season due to the water limitation that affects the microbial community. Rains at the beginning of the wet season then lead to peak emissions when microbial decomposition picks up again.

We will add a phrase to highlight that moisture is a necessary condition independent of temperature.

line(s) 44/45 "...may be beneficial for grassland productivity depending on its intensity".
Maybe add: "by removing moribund plant material and triggering growth (over-
)compensation."

We will add the phrase the reviewer suggested.

line(s) 49: "for the species" – "for the functional types". I'd rather consistently keep the focus
on functional types.

We agree and will make the amendments throughout the manuscript.

line(s) 52: "indirectly through alterations of the resource limitations" – add: "…that can cause
shifts in the competitive balance between functional types".

We will add the phrase suggested by the reviewer.

**Methods**

line(s) 105: "hot-steppe pasture in South Africa": this is a somehow unusual terminology /
vegetation classification. The Syferkuil site usually is referred to as savanna rangeland in
other publications.

The terminology for the naming of all sites was derived from the Koeppen Geiger climate
zones (in this case hot steppe). At the first mention we decided to add the form of grassland
management (pasture). We therefore will keep the naming as is but will add a phrase pointing
towards the term savanna rangeland in L105.

line(s) 107/108: That means no tests of fertilizer X defoliation intensity combinations? That
could be another interesting experiment to add, at least for the simulations.

Thank you for this interesting suggestion. In this part of the manuscript, we only mention the
managements for which experimental data were available and that could therefore be used to
parameterize the sites. Not knowing experiments including fertilizer X defoliation
combinations, we would be grateful for information and very interested to include such data
and combinations in further studies. The additional scenarios are described in 2.5. With this
separation we distinguish between the scenarios that were predefined by the data and those we
selected for further analysis. When defining the scenarios for further analysis, we decided to
use extreme cases to test the effect of different limiting resources (e.g. infinite nutrient
availability) instead of choosing different fertilizer levels. Regarding the defoliation intensity,
we agree that analyzing a gradient of different intensities provides another interesting
experiment. However, we decided to put our main focus on the resources and believe that the
defoliation intensities of the experiment already cover a sufficient range.

line(s) 115/116: Are the trait values you use to describe the strategies from within a
continuous range, or discrete fixed values? For example, if you use SLA as a trait to
distinguish between acquisitive and conservative strategies, then you will automatically cover
the extremes as well as in-betweens if you allow SLA to be a continuous trait that can range
between a minimum and maximum value (see, e.g., Scheiter et al., 2013, Langan et al., 2017).

The reviewer raises a very interesting point. LPJmL-CSR follows the concept of using a small
number of PFTs with fixed parameters. Therefore, for example SLA is fixed and each PFT

only covers one point of the continuum. We also see the potential for interesting future work following an individual based approach drawing trait values from a continuum similar to LPJ-FIT (Sakschewski et al., 2015) or aDGVM2 (Scheiter et al., 2013). However, the currently implemented management routines of such models are less detailed compared to "classic" DGVMs that include an agricultural component. We therefore see the necessity to continue to improve grassland representation in both model types for the foreseeable future.

line(s) 120 "Overview of managed grasslands in LPJmL" – "Overview of managed grassland representations in LPJmL" seems a more fitting title for this section.

We will adopt the recommendation of the reviewer.

line(s) 123/124: one polar, one temperate and one tropical grass: C4-type photosynthesis for the tropical grass? Knowing classic LPJ, I deem it likely that this is the case, but good to mention explicitly.

We thank the reviewer for pointing this out. Indeed the tropical grass is a C4-type and we accept their proposal to explicitly mention this here.

line(s) 130/131: (no water limitation, ref). – forgot to add the actual reference here.

We will add the reference to Jägermeyr et al., (2015) describing the water management routines.

Table 1: Forage supply [MgDM ha-1]: Terminology not entirely clear: Peak standing biomass? Annual withdrawal quantity (through mowing / grazing)? What is the temporal reference frame – annual?

We will change the unit to MgDM ha-1 yr-1 and add a phrase defining forage supply as annual quantity removed through defoliation from mowing or grazing.

line(s) 166-168: Does this new scheme also account for root biomass distribution in different soil layers, and therefore varying water availability between different soil layers? So that the total water uptake is the biomass-weighted uptake sum across soil layers? Or is it simpler than that?

We thank the reviewer for pointing out that this could be described more clearly. Root distribution between different soil layers was already used to determine the water supply from the different layers in the previous model version (Schaphoff et al., 2018). Our scheme retains this approach and only distributes the sum over the supply from all soil layers based on the root biomass. We will include this in the explanation of our approach.

line(s) 186: I suppose that means that SLA as a trait is a PFT-specific constant? I.e., it cannot vary over the lifetime of individual, or between different individuals of the same PFT?

Yes, it is a constant but as stated in our reply to a previous comment (L78-85), we agree with the reviewer that there is great potential in exploring the entire continuum.

line(s) 191/192: Does LPJmL distinguish between forbs and grasses, and if so, how is this implemented? And for grasses: does it distinguish between C3 and C4 photosynthetic pathway? Is age-mortality the only reason for mortality, or are there other causes implemented as well (e.g., due to negative annual C-balance, due to water stress, due to fire,
etc.)?

LPJmL does not distinguish between forbs and grasses and the herbaceous PFTs can include
both. C3 and C4 photosynthetic pathways are distinguished and we will add a description in
the methods section. In addition to age mortality, the model checks if a PFTs overall root or
leaf biomass becomes negative and kills the respective PFTs. Excessive water stress from
prolonged drought may be a cause of this. However, additional causes of mortality from water
stress such as embolism (Jacobsen et al., 2019) as well as heat stress are not included. Fire on
managed grassland has been implemented both as a disturbance (unpublished) and a
management practice (Brunel et al., 2021) but is not considered here. We will extend the
section on mortality to provide this additional information.

line(s) 193: "a biomass increase of the average individual dependent on the available area" –
rephrase? "the area-specific biomass increase of the average individual"

Using "area-specific" as suggested by the reviewer is in our opinion less explicit since it does
not define which area. We propose instead to replace "available area" with bare ground area.

section 2.3.3: general question on mortality: does the model distinguish between annual and
perennial herbaceous PFTs? I.e., do you have a PFT with enforced death after one growing
season? Enforcing annual types should implicitly strongly select for fast resource acquisition
at the expense of durable structural components, and a strong focus on reproductive
performance (see, e.g., Pfeiffer et al., 2019).

Currently, LPJmL does not explicitly distinguish perennial and annual PFTs and death is not
enforced at any time. Implicitly, the establishment as well as the mortality rate control the life
cycle of the PFT. High establishment and mortality rates lead to a fast turnover of the
population. We see potential in explicitly distinguishing annual and perennial PFTs for
example through constraining the period of establishment for annuals to the growing season.
We will pick up on this in the discussion.

line(s) 197: "we retained the approach of establishing saplings instead of seeds" – I assume
that refers to the tree PFTs? A bit unusual to refer to establishing grasses or herbs as
"saplings". I assume that you must have excluded tree PFTs from the simulations of the
grassland sites, allowing grasses/forbs only? Otherwise, it is likely that a forest type or
savanna type would have established as potential natural vegetation at least at the German and
South African sites. You should add the information of how you handled the tree component
of the model in the section where you describe your simulation protocol. Also clarify how
establishment is done specifically for the grasses / herbaceous layer.

Indeed, only herbaceous PFTs are allowed to establish on managed grassland stands. We will
add this to the model description. We agree with the reviewer that the term sapling is
misleading in this context and will replace it with the term seedling throughout the
manuscript. In addition, since this may create some confusion regarding the sapling LAI
parameter, for which we have to keep the term, we will explain the origin of the parameter
name and its purpose.

line(s) 199/200: So just to make clear that I understand correctly: the average individuals are
clones, i.e., all of the same PFT, but you introduced the clone-concept to be able to account
for PFT-specific reproduction aspects, such as seed numbers, germination rates, and seedling survival probability? If so, you should make it clearer than it is currently. It goes in the
direction of the problems faced by models that simulate actual, true individual plants and their
reproduction and establishment.

The reviewer raises an important point here. Indeed, the concept of the average individual
should be explained in more detail to prevent confusion with individual based approaches. We
will add a section in the methods explaining that each PFT can be seen as a representative for
a population with certain attributes that describe the population (e.g. number of average
individuals, individual biomass). In addition, we will discuss our approach in comparison to
an individual based approach to show advantages and disadvantages.

line(s) 203: age-dependent mortality: hard set (at a specific age), or based on an age-
dependent likelihood? And: the age-dependency differs between the different strategy types?

Thank you for this comment. Actually neither is the case. Depending on the growth
efficiency, the number of average individuals is reduced (Appendix A3 L745-753). Actual
mortality is derived from the maximum mortality rate - which is the same for all strategy
types - and the growth efficiency. The growth efficiency is dependent on SLA, which differs
between the strategy types (Appendix A3 Eq. A10). We will extend the description in
Appendix A3.

And what is the allowed maximum number of average individuals, and the maximum number
of grass-layer PFTs that can now coexist within one grid cell?

We thank the reviewer for this question. It made us realize that we did not include this in
Appendix A3. There is no hard maximum number of individuals. However, if the total
number of individuals exceeds 250 /ind/m2, 5% of the individuals die. We will add a
qualitative description in the method section and update the equations and explain the
underlying reasoning in Appendix A3. The number of PFTs per grid cell is in theory not
limited, however we decided to use one PFT for each main strategy for the purpose of this
study. For future studies this number can be increased, however this will also increase the
computation requirements. We will mention this in the model description.

line(s) 205/206: "It can be assumed that few individuals that maintain a high cover and
biomass must be larger…" – I assume all individuals that are part of one PFT have the same
size and biomass, given that you are still using the average individual concept? So, adding
new young individuals will lower the size and decrease the age of all clone individuals within
the PFT due to the averaging. But this implies that a strongly reproduction-oriented PFT
strategy would automatically have a smaller average individual size, a young average age, and
a larger number of clone individuals representing the PFT. This has implications for the age-
dependent mortality, as highly reproductive strategy types are then less likely to reach the age
where age-dependent mortality hits. Did you consider this aspect?

The reviewer raises an important point. We do not simulate the age of the average individual.
Our implementation of mortality depends on the growth efficiency. This describes the change
in carbon from photosynthesis and turnover per average individual compared to the average
individual carbon pools. In this ratio, the number of average individuals cancels out and the
key aspect is the GPP to turnover ratio, which should be smaller in older populations leading
to a higher mortality. We will also include this explanation in the method section on the
mortality.

Table 2: Maybe add a column that specifies the predominant gradient associated with the
parameter. You mention it in the text of this section, but it would be helpful to also have it as
a brief overview in the table. I find the distinction between biotic and abiotic dimension a bit
arbitrary/confusing with respect of the definition. Referring directly to the respective gradient
(stress gradient for biotic, disturbance gradient for abiotic) would seem more intuitive for me.

We will abandon the terminology abiotic and biotic gradient. When writing the original draft,
we found that it provides a clear distinction between the parameters related to each gradient.
However, as the reviewer correctly noted, this creates an additional layer of terminology to
understand when reading the manuscript. For this purpose we will modify the manuscript to
follow the terminology stress and disturbance gradient as proposed by the reviewer and add a
column to the table.

Table 2: Hierarchy: How did you determine the hierarchy? Based on your expert assessment?

We will add a phrase stating that the qualitative hierarchy of the parameter values for each
PFT was derived from expert assessment by all co-authors.

Table 2: Light extinction coefficient: Independent from SLA, or correlated? High-SLA leaves
should have more transmission than low-SLA leaves.

We agree with the reviewer that transmissivity of single leaves and their SLA are correlated.
However, we had to deal with the challenge that LPJml does not simulate multiple leaf layers
and cannot distinguish between the transmission of single leaves and the entire vegetation
layer. To account for the difference between leaf and entire vegetation transmission at least
implicitly, here the light extinction coefficient is not a measure of the transmissivity of a
single leaf. Instead it is the transmissivity of the entire vegetation layer of a PFT. Therefore,
we assume that PFTs, which have a high SLA can still have a high light extinction if many
high transmissivity leaves are stacked. In the current version of the manuscript this is only
touched upon in the discussion (L663-666). We will describe this in more detail in the
methods section.

Table 2: Maximum transpiration unit [mm] – if this is to be a rate, then the time part of the
unit is missing. [mm/day]?

We will change the unit to [mm d$^{-1}$]

line(s) 237/238: The root efficiency coefficient does affect the competitiveness between plants
(biotic interaction), but it also relates to the stress gradient (abiotic) with respect to water
uptake capacity. This is an example illustrating why using "biotic" and "abiotic" as
dimensions is maybe not the best way to make the distinction.

We agree that there are cases were the distinction between biotic and abiotic is not so clear.
As already proposed earlier (reply in L209-214) we will abandon the terms and only retain the
terms stress and disturbance gradient.

line(s) 240/241: The light extinction coefficient describes the fraction of light intercepted by
each additional leaf layer, right? As the amount of light that can transmit a leaf layer depends
on the thickness of the leaf, one would expect kbeer to be correlated with SLA, which, unlike
kbeer, you define as abiotic dimension. It would be good if you sort this out more clearly.

We agree and refer to our proposal from the related comment in the reply in L220-229. We will also describe more clearly, which parameters play a role for the stress or the disturbance gradient or for both gradients.

line(s) 241/242: the leaf area index of a sapling represents the offspring size - What do you define as "offspring size"? The height of the offspring, or its starting biomass, or its projected foliar coverage? I'm not sure LAIsap is a good description of offspring size, as its meaning is rather vague without a clearer definition. Whether a seedling/sapling of given leaf biomass has a high or low LAI is a function of its SLA, so LAIsap for a given unit of leaf biomass essentially is nothing else as another way to refer to SLA.

In LPJmL, the leaf area index of a sapling is only used to calculate the sapling biomass using SLA. So instead of assuming a given leaf biomass, we assume a given SLA and calculate the leaf biomass. Using the same SLA, a higher sapling LAI is equal to a higher sapling biomass. We will change offspring size to offspring biomass and add an explanation of the relationship to SLA. We will also revise the discussion to reflect both SLA and sapling LAI when discussing offspring biomass.

Table 3: Flip order of columns "variable" and "site", as site is unique and variable is tied to site and non-unique.

We agree and will apply the proposed change.

line(s) 287/288: "the current representation of some processes within the model" – which processes specifically?

We here refer to section 4.1.2 where these processes are listed. We will change "some processes within the model" to "the processes, listed in sect. 4.1.2," and remove the reference to section 4.1.2 at the end of the sentence.

line(s) 299: 390 years - your spin-up duration? Did you add a transient simulation period after the spin-up (how long? For what time-period?). One can only guess based on the time-axis labeling in the figures that follow in the results section. Please specify this with some more detail.

We agree that additional information is needed. We first conducted a potential natural vegetation spin-up simulation of 30000 years followed by a spin-up including land use of 390 years after which the transient simulation start. We will add this to the modelling protocol section.

Modelling protocol: What is the temporal resolution the CSR-model version runs on? Monthly, or daily?

All processes are executed on a daily time scale. We also compute the outputs on a daily timescale but aggregate to a monthly or annual resolution for some of the results. We will add a sentence on this to the modelling protocol.

How do you initialize community composition with respect to present PFTs and shares of PFTs at the beginning of the simulation? Based on the field-based observations? If so, how would you do it in a situation where you did not know the field situation of sites, e.g., for a large-scale or global simulation? (Question for the discussion, I guess).

Upon initialization, each PFT is established dependent on the respective establishment rate
and biomass (derived from sapling LAI, SLA and leaf to root ratio). Therefore, initially a PFT
with high values in both has a higher share in the community. However, if its strategy is not
suitable this will change over time. This means, that no data on initial community
composition or similar is needed. We will add this explanation to the model description.

**Results**

Figure 1: Please specify temporal reference frame for panels a, d, and g - is it the annual sum
(yield), the peak season leaf biomass (leaf biomass), the grazing period duration offtake
(grazing offtake)?

We will add a more thorough explanation in the caption.

General question on all scenarios that included animal grazing: Is preferential grazing, i.e.,
selection of more palatable over less palatable PFTs, accounted for by the new CSR model
version? Unlike mowing or biomass removal by fire that is indiscriminate, biomass removal
by herbivores can alter community composition quite substantially, especially under high
grazing pressure. If preferential grazing is not yet implemented, this should be added as a
limitation in the respective section of the discussion, and could be pointed out as a future need
for development.

The reviewer raises an important point. Indeed the current implementation (Rolinski et al.,
2018) does not consider preferences for specific PFTs. We will briefly mention this when
describing the model and add it to the section on future need for development.

line(s) 365-368: Ecologically, the shift towards more investment into above-ground biomass
(growth (over-)compensation) and towards a more resource-exploitative strategy
(construction of "cheaper" leaves with reduced life duration is plausible. However, I do not
see right away why the minimum canopy conductance should decrease due to grazing?

We agree that the decrease of the minimum canopy conductance is unlikely to be related to
grazing directly. More likely, the high and similar minimum canopy conductance of the
ungrazed scenario (C0) is an artefact of the parameterization. All parameters can be assigned
primary and secondary processes that they affect. The leaf to root ratio and the SLA are
different in the two scenarios and act as a compensation of defoliation from grazing (primary
process). However, to some extent these parameters also control access to and distribution of
resources (secondary processes). In the ungrazed scenario, these do not need to be adjusted to
compensate for the defoliation but can still play a role in the competition for water. Therefore,
more parameters can control resource access and distribution and it is likely that this will
affect the parameterisation of minimum canopy conductance. We will amend the description
of the parameters to account for the primary and secondary processes affected and add the
explanation to the discussion.

line(s) 406/407: How does the relative contribution of the S- and R-PFT to the forage supply
compare to their relative abundance or relative contribution to FPC? I.e., did they contribute
more or less than could be expected according to their relative abundance within the
community?

Thank you for the interesting question. We did not look into this in detail. Since biomass is an
important variable when calculating FPC, we believe it is likely that forage supply and growing season FPC are similar. However, there might be differences when averaging over the entire year. We will analyze our results regarding this and amend the manuscript if we discover additional interesting results.

line(s) 442/443: "In the irrigated scenario, only the S-PFT contributed to forage supply." - That is a bit surprising? One would expect that irrigation reduces stress resulting from water limitation, therefore opening the community more strongly for the C-PFT.

This was also surprising and counterintuitive to us. We already provide an explanation in the discussion in L579-583, which we will now reference in the sentence in L442f.

line(s) 473/474: "…still dominated by the S-PFT." - Is this a legacy effect from the pre-irrigation time period's community composition? If run long enough without resource limitation (i.e., with irrigation on), would the S-PFT type be replaced by the C-PFT type, and if yes, how long do you expect this would take? Can be part of the discussion, if not already discussed there.

We already touch upon this in L542-547 but agree that this can be discussed in more detail. We will add a reference in L473f and extend the discussion in section 4.1.2.

**Discussion**

General remark: how do you intend to use the CSR-model in the future, if you ideally need an a-priori determination of the ideal PFT parameterization depending on site, community, and management? And how can communities respond to changing management or environmental conditions if the parameterization of the PFTs cannot be dynamically adjusted during the simulation based on a selection mechanim that filters for the best-suited parameterization under the given circumstances?

The reviewer raises several interesting questions that go beyond this study. We are currently working on a globally applicable set of PFTs, which will form the basis of another study in the near future. For that study, we retain the fixed PFT parameterization of classic DGVMs. However, we are generally open and very much interested in further developing the model. As already mentioned in the reply in L78-85, it would be very interesting to combine the approach of LPJmL-CSR and aDGVM2 or LPJ-FIT.

line(s) 494/495: "IN LPJmL-CSR, growth of the vegetation was faster than in LPJmL 5.2, which led to higher yields for all cuts." – Elaborate briefly on the causes for the faster growth in the new model version.

The faster growth compared to LPJmL 5 has two reasons: First, the new implementation of biological nitrogen fixation led to less nitrogen stress and higher photosynthesis. Second, this is also a result of the new parameterization, which was tailored to this site. We will add this explanation after L494f.

line(s) 504: "but selected a livestock density of 1.0 cows ha-1" – use "livestock units" rather than cows (how about steers, heifers, etc.); And: Is this to determine the amount of manure input? The temperate grassland was not grazed but mowed, so livestock density does not make much sense with respect to grazing off-take?

The livestock density refers only to the spin-up and the historical periods for which no data on
actual land use were available. Therefore, it is entirely unrelated to the transient simulations
that reproduce the mowing experiments. We will rephrase this paragraph to make this clear.

line(s) 506: Briefly describe the processes / mechanisms that lead to increased carbon input to
the soil in the CSR-version compared to the old version.

We identified three causes for the increased carbon input: First, the SLA longevity trade-off
we implemented led to an increase in turnover supplying more carbon to the litter layer.
Second, implementing explicit mortality of average individuals created an additional input
into the litter layer. Third, accounting for the carbon added through the application of manure
fertilizer also constituted an additional carbon input into the system. We will add this
explanation after L506f.

line(s) 526/527: Here finally the information that I was missing in the methods section. You
should add this information to the modeling protocol section (that you did exclude the tree
PFTs from your site-scale simulations.

We will adopt this suggestion (see also reply in L152-157).

line(s) 528/529: You should try to give a reason for the "why" of this, instead of simply
repeating the result. For example, an explanation could be that grazing was not the only / the
main stress for herbaceous vegetation at this savanna site. The site has a pronounced dry-vs-
wet season dynamics, and therefore water limitation as a stress factor, maybe also N-
limitation, may be causes for the dominance of the S-type irrespective of the grazing
management.

We agree with the reviewer that this should be explained and share their opinion of the
underlying reasons. We will add a sentence to explain the dry wet dynamics of the site and
that these are independent of grazing, which therefore does not affect the water stress level
allowing the S-PFT to remain advantageous.

line(s) 540/541: You could test this by specifically allowing no other PFT than the S-type to
enforce a monoculture.

We discussed the possibility to investigate this further, but decided against because LPJmL
would limit us to simulating an S-PFT monoculture already before the beginning of the
irrigation, which would likely lead to different initial conditions when starting irrigation. This
would make it difficult to interpret the results.

line(s) 544/545: Was your simulation time period with irrigation long enough to allow
establishment of a new steady state with respect to community composition? In my
experience, community composition shifts are one of the slower processes and can take quite
a number of years before reaching a new steady-state after a change in forcing has occurred.

We touch upon this in section 3.4.2 L475f by saying that "the transition occurred within the
first one to two years", which is much faster than we would expect. We mention this when
discussing the change in soil organic carbon (L532-538) but we agree that this is very brief
and will add more detail and highlight the transition time more prominently. We will also
provide an explanation for the fast transition, which is related to the removal of competition
for water. In a water scarce environment, the S-PFT as a water saver was advantageous and the C- and R-PFT were subordinate. Under irrigation, the S-PFT's slow growth becomes a
disadvantage and the C- and R-PFT can exploit resources more efficiently. Both increase their
biomass rapidly until a different limitation prevents further increase, while the biomass of the
S-PFT remains similar. This is comparable to real world ecosystems. However, existing
individuals cannot grow infinitely and need to reproduce producing new individuals. This
process of reproduction and dispersal may slow down the transition. In LPJmL, the PFTs
increase their biomass independent from the establishment of additional individuals which
speeds up the transition. We will add this explanation to the methods and discussion sections.

line(s) 545/546: "However, periods of drought can induce and additional disturbance." –
Correct, but not in this case, because due to the irrigation you had drought eliminated.

The reviewer is correct. A plausible explanation is that the parameterization allows the R-PFT
to coexist with the C-PFT if the main resource limitation is removed. We will update the
explanation.

line(s) 549: "LPJmL 5.3 underestimated the observed forage supply…" – I'm not sure about
your usage of the term "forage supply" (generally throughout the manuscript) - is forage
supply, according to your definition, the potentially available biomass offered by the
rangeland, or do you actually rather mean "the amount of feed required by the animals"
(which should then be termed as "forage demand"?

We agree that our use of forage supply was ambiguous because we use it to define the amount
of biomass removed through mowing or grazing for the temperate grassland and the cold
steppe but also for the amount of leaf biomass available for grazing for the hot steppe. This
was an attempt to use common terms for all sites, which appears to be confusing instead of
helpful. We will therefore change the term forage supply to forage offtake for the temperate
grassland and the cold steppe and use the term leaf biomass for the hot steppe. We will add a
definition of forage offtake in the methods section and explain why we use a different term
for the hot steppe.

line(s) 552/553: I do not understand: how does feed demand change forage supply? Forage
supply is a biomass potential offered by the plant community. Increased feed demand, as
described here by your correction, should not increase the forage supply of the plant
community (unless through growth overcompensation), but rather reduce the supply due to
the increased demand from the animal side?

As in the previous comment we acknowledge that using the term forage supply creates some
confusion and refer to our proposal to change this (reply in L424-431)

line(s) 554/555: The fact that animal demand could not be met AND above-ground biomass
collapsed is a rather clear indication of over-grazing / exceeding of rangeland carrying
capacity. In this context, maybe also discuss changes in the PFT community composition, i.e.,
changes in the prevailing strategy types. It can be expected that such a shift in strategy types
occurs under such circumstances.

We agree with the reviewer that the model results provide strong evidence for overgrazing
and will add a phrase explicitly stating so. We will also add a sentence discussing the change
in community composition which shows an increase of the C-PFT (and also to some extent
the R-PFT) as shown in Fig SI 9 and 12.

line(s) 562/563: You did not combine fertilization with irrigation, right? Do you expect that fertilization in combination with irrigation would increase leaf biomass beyond the level reached with irrigation alone?

Generally, irrigation alone already affects processes related to inorganic N inputs and losses. Biological N fixation and mineralization increase with increasing soil moisture. However, irrigation also leads to higher leaching. We therefore expect that the PFTs are still N limited even though irrigation may already increase but could also decrease inorganic N availability. Additional inorganic N from fertilization may remove the N limitation leading to an additional leaf biomass increase but may also lead to higher maintenance respiration limiting leaf biomass growth. Therefore, we cannot give an unambiguous answer and will add this explanation in section 4.1.3.

line(s) 575: "Fertilization had no effect on SOC" – Not surprising, given that fertilization without irrigation did not increase leaf biomass and therefore C-input to the soil.

We agree with the explanation of the reviewer and will add this to the sentence.

line(s) 580/581: "it seems that an S-strategy remained advantageous" - Again, I wonder about the turnover time required by the model to let a community transition from one steady-state to a new steady-state.

While for the hot steppe we can provide clear evidence, that a new steady state was reached, for the cold steppe the reviewer raises an interesting point. Increased soil moisture from irrigation may lead to an increase of the NO3 and NH4 pools from mineralization and biological nitrogen fixation which may take longer than the simulated time frame (see also reply in L420-423). We will add this to the discussion.

line(s) 600: And it may be interesting how grass-tree coexistence (typical for savanna sites as the one one in South Africa) will affect grass layer community composition compared to a situation where trees are excluded from the simulation.

Indeed an improved representation of Savannahs would be a major step for DGVMs. In order to achieve this, we see the need for additional model development as discussed in Rolinski et al., (2021). We will add a sentence on this in the limitations and further need for research section.

line(s) 606/607: "Generally, a change in resource availability does only change the conditions for the establishment of a community but does not directly affect the established vegetation" – Environmental filtering can also affect the established community by increasing mortality for specific strategy types within the community, not only by changing establishment success of given strategy type. Since you seem to have no other mortality causes aside from age-dependent mortality in the model (at least not for the grass layer), you will not see this effect, but it does exist, nonetheless.

We agree with the reviewer and will extend this sentence to reflect the limitation of our model to age mortality and to discuss potential effects of other causes of mortality.

line(s) 614: Why are N-fixers not separate PFTs in the model? I'm a bit surprised that they are not.

Facing the challenge of adding new PFTs to a classic DGVM, our aim was to reduce complexity as much as possible at first. This included restricting ourselves to add as little PFTs as possible. Grouping N-fixers with non-fixers halved the number of PFTs. We believe this is reasonable because the model will only fix additional N if the demand is not fulfilled. In an approach with two separate PFTs, this would mean a change in community composition and an increase of the N-fixer PFT at the expense of the non-fixer. In our approach, this simply means an increase in biological nitrogen fixation. One could say, that implicitly the PFT is a fixer if needed and not if not needed and could determine this status using the biological nitrogen fixation output. We will add the necessary detail to the description of biological N fixation in Appendix A4.

line(s) 622/623: So the assumption is that grazing is non-preferential, correct? I.e., grazers do not favor one PFT over another, for example based on criteria that characterize palatability / nutrition value. This is a simplification in the model that should be discussed briefly, as herbivores usually do not function the same way as mowing (or fire) that removes biomass indiscriminatingly.

Yes, grazing is not preferential. As proposed in our reply in L301ff we will include this in the model description and include a brief discussion in the limitations.

line(s) 624: "tolerance or avoidance" – Avoidance would for example (aside from temporal avoidance) be realized by being unpalatable. As your grazing is non-preferential, being a grazing avoider type based on palatability would not make a difference in your model as the animals would not discriminate against the avoider. This is a limitation you should mention.

We thank the reviewer for raising this point and will include grazing avoidance through palatability in the limitations together with preferential grazing (reply in L503f).

line(s) 629/630: "… and the PFTs had to follow a grazing-tolerance strategy." - The fact that grazing avoidance can only be achieved through life cycle adaptation and not through palatability likely causes a bias in your community composition. You should at least mention this possibility.

We thank the reviewer for their suggestion and will add a sentence on this at the end of the section (L631).

line(s) 632/633: "At the cold steppe site, grazing only happened during the growing season and both grazing tolerance an avoidance could be useful strategies." – Well, likely not avoidance in the way you can represent it in the model (temporal avoidance). If grazing happens during the growing season, and your only way to implement avoidance is through life cycle adaptation, i.e., temporal avoidance, this will push avoiders to the non-growing season as time when no grazing happens. But I don't see how avoiders could succeed by shifting their existence focus to exactly the season when growth is not possible?

We will add a phrase acknowledging that the model will not be able to simulate the type of avoidance that is likely successful.

line(s) 643-645: This challenge could be circumvented by moving away from a PFT-concept with fixed pre-defined parameter values for each PFT, which implicitly limits the number of strategies that can be realized, for example by defining typical value ranges for the given parameters of a strategy type. Within these continuous ranges, a strategy type can assume many trait value combinations that define its location within the trait space occupied by the strategy type, and therefore allows more plasticity within a strategy type, e.g., a plant could be a moderate, intermediate, or extreme S-strategy type.

We agree with the reviewer, that moving away from the fixed PFT approach is a suitable way to circumvent many of these issues. As discussed in previous comments one necessity is to follow an individual based approach as in aDGVM2 or LPJ-FIT. We see this as a promising and intriguing topic for future model development of LPJmL-CSR and will emphasize this more in the discussion.

line(s) 645/646: The challenge will be to expand this site-scale-focused approach to a generalized large-scale / global approach, because it will not be possible to parameterize suitable PFTs for all imaginable locations and circumstances. I think the value of what you show in this study is to prove that the CSR-concept can work within a DGVM and is ecologically sound in many points. But to make it general, you will have to move away from the discrete parameterization of your PFT approach, for example by allowing an evolutionary approach that self-selects successfull strategies via environmental filtering from a pool of potential trait value combinations, where each trait is represented by a continuous range of allowed values.

The generalization for a global application indeed poses a challenge. However, for the tree PFTs, researchers managed to find a set for classic DGVMS that represents the broad range of environmental conditions possible. We believe that for herbaceous PFTs it will also be possible to find a suitable set that will improve the representation of grasslands in current DGVMs We hope to present this in a separate study in the near future. In the long term, additional model development including the step towards dynamic PFTs will further improve the representation of different growth strategies.

line(s) 664/665: I do not really agree with this approach. The light extinction coefficient (as I know it) is a constant that describes how much light a respective layer of leaves will absorb and how much it will allow to transmit to the next lower leaf level. As such, it is a proxy associated with leaf characteristics such as leaf thickness or SLA more than overall plant stature. If anything, I'd deem LAI closer to stature than the light extinction coefficient, if you do not have height available as state variable.

The reviewer is correct that the light extinction coefficient usually refers to the transmissivity of a leaf layer. In theory, this is represented as one leaf with a given height and SLA per layer. However, LPJmL and other classic DGVMs do not simulate different leaf layers but calculate the light extinction of the entire vegetation layer of one PFT. Therefore, the model actually calculates the light extinction of a stack of leaves. A larger stack of leaves will transmit less light and therefore has a higher light extinction coefficient compared to a smaller stack of leaves. Following this, the height of several leaf layers (or the vegetation layer) can be interpreted as a function of SLA and the light extinction coefficient. As mentioned in the discussion (L663-666) and previous work (Wirth et al., 2021) we think that this is a major limitation and believe that adding plant height as a state variable would be an important model development. As stated in our reply to the related comment in L218f we will improve the model description and refer to this in the discussion.

line(s) 674: In rangelands, mechanical stress through trampling would be another important aspect to consider.

Similar to the missing inclusion of preferential grazing (comment in L294-300), this is related to the representation of grazing. We will add trampling to the discussion of the limitations of the current grazing approach as well.

**Minor editorial comments**

We appreciate the thorough reading and will adopt all minor editorial comments below without responding to each of those separately.

line(s) 10: "… a temperate grassland, a hot and a cold steppe…" => "… a temperate grassland and a hot and a cold steppe…"

line(s) 13: at three grassland sites => at the three grassland sites line(s) 17: Our results show, that => delete comma line(s) 39: high carbon inputs => high carbon sequestration line(s) 61: (examples) => delete, seems to be a leftover note from manuscript writing. Or alternatively replace with the examples you were thinking of…

line(s) 183: "recover slower" – "recover more slowly"

line(s) 184: "the SLA leaf longevity trade-off" – "the SLA v. leaf longevity trade-off"

line(s) 328: "While it remained similar…" – "However, it remained similar…"

line(s) 359 correct typo: resourCe line(s) 420 contribute – contributed line(s) 456: "we present results on above-ground biomass" – "we present results based on above-ground biomass"

line(s) 490: "this allows to assess" – "this allows assessment of", or "this allows assessing"

line(s) 496: we only assess – we only assessed line(s) 533: "this can be explained with" – "this can be explained by"

line(s) 535: "and contributed to the litter layer" –"and increased the input to the litter layer".

line(s) 539: "In addition irrigation led to…" – "In addition, irrigation led to…"

line(s) 619: "…which constituted an additional investment." – Rephrase? "…and therefore, a reduction of investment costs associated with N-fixation."

Hyphenation of two-word combinations that are used in the function of an adjective:

line(s) 69: "disturbance prone environments" – "disturbance-prone environments"

line(s) 73: "multi species communities" – "multi-species communities"

line(s) 181 "stress prone ecosystems" – "stress-prone ecosystems"

l- 203: "age dependent individual mortality" – "age-dependent individual mortality"

line(s) 231 "plant specific resource availability" – "plant-specific resource availability"

line(s) 249 "site specific conditions" – "site-specific conditions"

line(s) 296: bias adjusted data" – "bias-adjusted data"

line(s) 374, 375 "water saving strategy" – "water-saving strategy"

line(s) 397 resource limited – resource-limited line(s) 473: "S dominated community" – "S-dominated community"

line(s) 496: "neither water nor nutrient limited" – "neither water- nor nutrient-limited"

line(s) 542, line(s) 543, line(s) 579 "S dominated" – "S-dominated"

l- 580 "nutrient limited" – "nutrient-limited"

**References**

Scheiter, S., Langan, L. and Higgins, S.I., 2013. Next-generation dynamic global vegetation
models: learning from community ecology. *New Phytologist*, *198*(3), pp.957-969.

Langan, L., Higgins, S.I. and Scheiter, S., 2017. Climate-biomes, pedo-biomes or pyro-
biomes: which world view explains the tropical forest–savanna boundary in South America?.
*Journal of Biogeography*, *44*(10), pp.2319-2330.

**References responses**

Brunel, M., Rammig, A., Furquim, F., Overbeck, G., Barbosa, H.M.J., Thonicke, K., Rolinski, S., 2021.
When do Farmers Burn Pasture in Brazil: A Model-Based Approach to Determine Burning
Date. Rangel. Ecol. Manag. 79, 110–125. https://doi.org/10.1016/j.rama.2021.08.003
Jacobsen, A.L., Pratt, R.B., Venturas, M.D., Hacke, U.G., 2019. Large volume vessels are vulnerable to
water-stress-induced embolism in stems of poplar. IAWA J. 40, 4-S4.
https://doi.org/10.1163/22941932-40190233
Jägermeyr, J., Gerten, D., Heinke, J., Schaphoff, S., Kummu, M., Lucht, W., 2015. Water savings
potentials of irrigation systems: global simulation of processes and linkages. Hydrol. Earth
Syst. Sci. 19, 3073–3091. https://doi.org/10.5194/hess-19-3073-2015
Rolinski, S., Müller, C., Heinke, J., Weindl, I., Biewald, A., Bodirsky, B.L., Bondeau, A., Boons-Prins,
E.R., Bouwman, A.F., Leffelaar, P.A., te Roller, J.A., Schaphoff, S., Thonicke, K., 2018.

Modeling vegetation and carbon dynamics of managed grasslands at the global scale with
LPJmL 3.6. Geosci Model Dev 11, 429–451. https://doi.org/10.5194/gmd-11-429-2018
Rolinski, S., Wirth, S.B., Müller, C., Tietjen, B., 2021. Strategies for assessing grassland degradation,
in: Joint XXIV International Grassland and XI International Rangeland Kenya 2021 Virtual
Congress Oral Papers Proceedings. Presented at the Joint XXIV International Grassland and XI
Rangeland Virtual Congress, Kenya Agricultural and Livestock Research Organisation, Nairobi,
Kenia, pp. 383–387.
Sakschewski, B., Bloh, W. von, Boit, A., Rammig, A., Kattge, J., Poorter, L., Peñuelas, J., Thonicke, K.,
2015. Leaf and stem economics spectra drive diversity of functional plant traits in a dynamic
global vegetation model. Glob. Change Biol. 21, 2711–2725.
https://doi.org/10.1111/gcb.12870
Schaphoff, S., Bloh, W. von, Rammig, A., Thonicke, K., Biemans, H., Forkel, M., Gerten, D., Heinke, J.,
Jägermeyr, J., Knauer, J., Langerwisch, F., Lucht, W., Müller, C., Rolinski, S., Waha, K., 2018.
LPJmL4 – a dynamic global vegetation model with managed land – Part 1: Model description.
Geosci. Model Dev. 11, 1343–1375. https://doi.org/10.5194/gmd-11-1343-2018
Scheiter, S., Langan, L., Higgins, S.I., 2013. Next-generation dynamic global vegetation models:
learning from community ecology. New Phytol. 198, 957–969.
https://doi.org/10.1111/nph.12210
Wirth, S.B., Taubert, F., Tietjen, B., Müller, C., Rolinski, S., 2021. Do details matter? Disentangling the
processes related to plant species interactions in two grassland models of different
complexity. Ecol. Model. 460, 109737. https://doi.org/10.1016/j.ecolmodel.2021.109737

---

## Author Comment (AC2)

This study is predicated on a novel way of quantifying CSR plant functional types (PFTs) for species, and comparing these with frameworks including the leaf economics spectrum. This forms the basis of the entire analysis, and so it is fundamental that the way the PFTs are derived represents CSR theory and can be compared against the LES. There are a number of basic problems with the approach used here, however.

We cordially thank the reviewer for taking the time to review our manuscript. When comparing the reviewer's perception of our topic with our short summary, we gained the impression that there has been a misunderstanding regarding the main focus of our study. In turn, we believe that this led to a number of misconceptions, which we address in detail in our responses. We are confident that our approach is solid and we hope that our suggested changes to the manuscript as well as the explanations in our author response will help to resolve the issues raised by the reviewer.

Using the trait specific leaf area (SLA) to represent both leaf economics and also within the CSR calculations means that to two measures are very likely correlated, potentially leading to a Type 1 statistical error in which the conclusions are accepted despite the statistical test not being sufficient to assign a realistic probability.

Thank you for pointing this out, as it shows that our approach was not clearly described. We agree that Type 1 statistical errors need to be avoided. However, we are not representing a statistical but a functional relationship between SLA and leaf longevity here (L175-186 and 709-714). The connection between SLA and leaf longevity is well established following leaf economics (LES, Wright et al., 2004), but was so far not implemented as such in the LPJmL model for grasslands.
In the original LPJmL model version, SLA was only used to compute the leaf area index (LAI) from the internally computed leaf biomass. In order to represent the establishment of different C-, S- or R-strategists, it is important to represent advantages and disadvantages of the leaf structure in the model. Thinner leaves (high SLA) have a shorter longevity and while they grow quickly to intercept light, they need to be replaced frequently. Neglecting the need to replace thin leaves more frequently would lead to an advantage of high SLA values under all circumstances, which is in contrast to ecological theory and observations (e.g., Díaz et al., 2016; Reich, 2014; Wright et al., 2004). This trade-off had been implemented and applied to tropical (Sakschewski et al., 2015) and European forests (Thonicke et al., 2020) before. The implementation in this study provides a functional relationship of the SLA-LL relationship, as part of the LES, and CSR theory through SLA in grasslands (section 2.3.2 and appendix A2). This newly implemented functional relationship controls the productivity of the different PFTs and the resulting shares of the C-, S- and R-PFTs. However, we do not compare the LES to the C-, S- and R-PFT shares, which would be comparing inputs to outputs and would certainly show a correlation.

We will add a paragraph to section 2.3.2 in which we describe

- the role of the functional relationship between SLA and leaf longevity to distinguish different growth strategies,
- how this functional relationship together with the resource uptake and distribution (section 2.3.1) as well as reproduction and mortality (2.3.3) provides the basis for the dynamic computation of C-, S-, and R-PFTs' productivity, and
- how the productivity determines biomass and cover which are used to determine community composition.

We will further revise and amend section 2.3.2 and appendix A2 to improve the description of
our approach.

With regard to stress, the authors state that "According to CSR theory, the stress gradient
expresses the level of stress a species is exposed to in a certain habitat. It ranges from
unstressed to severely stressed, but does not distinguish individual stress categories (e.g.
temperature, water or nutrient)" thus "different strategies for water-resource use can be used
to distinguish C- and R-strategists (low stress tolerance) from S-strategists". Thus the traits
used here are specific to water stress, and the definition of stress recognised in CSR theory
(constrained metabolic efficiency and thus biomass production) is not cited nor considered.

The reviewer raises a valid point. Of course, stress is not restricted to water stress and other
traits that are related to (too high or too low) temperature or to nutrient stress could be used to
distinguish PFTs. In principle, the LPJmL model also considers stress arising from
temperature and nutrient availability in addition to water stress in its phenology and nitrogen
acquisition routines. However, the grassland steppe sites that we simulated in our study are
predominantly limited by water. Therefore, we decided to focus only on water stress in this
first application of LPJmL-CSR. This allows for a better understanding of the underlying
processes and the resulting pattern. In addition to the traits related to general stress tolerance,
we therefore only include traits related to water stress. However, we agree that the
implications of this simplification should be discussed. We will

• explicitly list the types of stress that are represented or disregarded by LPJmL in
section 2.1,
• add the definition of stress recognized in CSR theory as proposed by the reviewer and
the above reasoning for our focus on water stress to section 2.4.1, and
• discuss the implications of not using traits that are related to temperature and nutrient
stress tolerance for our results - especially for our simulation experiments on resource
limitation - in section 4.3.

Any stress (including water stress - but also factors such as nutrient stress or 'non-resource'
stressors such as temperature) limits metabolic performance and thus growth and biomass
production. Internal, inherent metabolic traits (such as photosynthetic capacity and dark
respiration rate) or growth traits (such as relative growth rate) would have been acceptable to
demonstrate limitation, but the authors provide no evidence that, for instance, that specific
adaptations determining canopy-level conductance can represent the extent of general
tolerance to stress.

We agree with the reviewer that limited metabolic performance is the result of various types
of stress. Depending on the complexity of the model, responses to stress can be computed
internally (reduced growth rate or reduced photosynthetic capacity) if these are implemented
as dynamic functions in the model responding to, e.g., non-optimal temperatures or nutrient
limitations. In LPJmL, SLA is important to determine photosynthetic activity and therefore
affects the growth rate (L175ff). The leaf-to-root ratio affects the photosynthetic activity as
well by controlling the investments into additional leaves. Therefore, we do not only consider
traits related to tolerance to water stress but also traits related to a general tolerance to stress.
We realize that the original version of the manuscript may not have been sufficiently clear in
this regard and will improve the description of the role of the different traits and how they
represent tolerance to stress in section 2.4.1. To achieve this, we will include a short
paragraph for each trait that provides the following:

- a definition of the trait,
- the predominant gradient (stress or disturbance) the trait is associated with through the processes it affects, and
- its use within the computations of LPJmL-CSR including all processes it affects.

We also agree with the reviewer that minimum canopy conductance and maximum transpiration rate do only relate to water stress. However, we selected four traits associated with the stress gradient to represent differences between the strategies. Two traits that are associated with general tolerance to stress through their importance for plant growth and two traits that are specific for water stress. As stated in a previous reply in L62f, we did not select additional traits that specifically relate to other types of stress that are represented in the model (temperature and nitrogen). With a better emphasis of our focus on water stress, the selection of traits relevant for water dynamics is hopefully more comprehensible. As already stated in our reply in L55-64, we agree that our description of the representation of different types of stress and our reasoning to focus on water stress needs to be improved. We made a proposal how this can be achieved at the end of the reply in L65-71.

Line 233: the authors state that "plant stature … can be used to distinguish C- and S-strategists (low disturbance tolerance) from R-strategists". No: S-selected species can be small (e.g. Salix herbacea) but some may become large over a long life-span (i.e. Sequoiadendron giganteum). What matters is the C-selected species get large quickly, S-selected species can become large eventually over a long life-span, and R-selected species cannot. This is more a reflection of longevity and how rapidly plants achieve adult size.

We agree with the reviewer that S-strategists generally show a variety of statures as they nicely illustrated with their examples. This is also clearly stated in Table 2 of Grime (1977) to distinct species of the different strategies. While you can also find tall S-strategists in grasslands (e.g. *Brachiaria brizantha* ), generally grassland plant species are of approximately similar height (Gommers et al., 2013; Pontes et al., 2015).
Still, the reviewer raises an important point. If one would only consider stature, an S-strategist might not be clearly distinguishable from a C- or an R-strategist and our explanation can be misinterpreted this way. However, we are aware of the importance of growth rate and longevity when distinguishing C-, S- and R-strategists. To account for this, the LPJmL model represents the fast-slow economics of the LES as explained in more detail in our reply in L17-47. Furthermore, we do not prescribe plant stature. Instead, we use a parameter that just represents the potential stature a strategist can attain. Depending on abiotic and biotic factors, the C- and S-strategist can become large but the R-strategist cannot. The C-strategists will grow rapidly if sufficient resources are available. The S-strategist will grow slowly but accumulate large amounts of biomass over a longer time or remain small if it is disturbed or outcompeted. We will amend

- section 2.4.1 to more clearly explain the distinction between the C-, S-, and R-PFTs underpinned by the description of the traits and their use within the model as proposed in our reply in L91-94, and
- section 2.1 to include additional details on the growth dynamics including a qualitative description of the photosynthesis, allocation and turnover routines implemented in LPJmL-CSR.

In the present study only juveniles were investigated, so using the leaf area index of a sapling is not going to represent the strategy in the main vegetative phase (seedling CSR strategies are known to be different from adult CSR strategies; Dayrell et al. (2018) Functional Ecology 32, 2730-2741).

We agree with the reviewer that it is important to not only address CSR dynamics of juvenile plants. However, we would like to stress that we do not focus on juvenile plant dynamics. We assume that this misunderstanding originates in the description of establishment where saplings are established on bare ground. Still the model simulates an average individual that typically represents an adult plant (unless the entire plot has been re-established with new plants). We will add the explanation that LPJmL-CSR simulates adult plants to section 2.1 together with the details on the growth dynamics (see details in L131ff).

Also, CSR strategies are phenotypic characters (i.e. attributes of the individual plant that are subject to natural selection), but establishment rate (kest) [line 237] is not a character of an individual (the units of measurement are stated in Table 2 as the number of individuals per metre squared per day – a population measure), and so cannot elucidate the individual phenotype or adaptations of the species (i.e. the plant strategy or PFT).

We agree that CSR strategies can be defined as a phenotypic characteristic of an individual and it may be counterintuitive that a measure that is not reported as being per individual but per meter squared can be used to represent a phenotypic characteristic. However, the establishment rate is just a parameter used within the model to calculate the actual establishment (appendix A3). This calculation considers several variables including the number of individuals and the resulting actual establishment can be reported as individuals per individual (a phenotypic characteristic). We see the point that this may be misunderstood. Also, LPJmL-CSR does simulate trait plasticity as well evolutionary processes. Therefore, phenotypic adaptation is not accounted for and adaptation only occurs at the community level through changes in its composition.
To address the reviewer's concerns, we will

- provide a more thorough qualitative explanation of the establishment in section 2.3.3,
- explain the use of $k_{est}$ within the establishment routine of LPJmL-CSR more detailed in appendix A4, and
- clarify that we do not simulate phenotypic adaptation in section 2.3.3.

In Figure 4, the red, green, blue (RGB) color scheme is used both to represent the extent of C, S and R and the experimental treatments rainfed (red), irrigated (blue) and fertilised (green).

We agree that the coloring is not enhancing clarity and will remove the colors from the axis labels of the ternary plots.

**References**

Díaz, S., Kattge, J., Cornelissen, J.H.C., Wright, I.J., Lavorel, S., Dray, S., Reu, B., Kleyer, M., Wirth, C., Prentice, I.C., Garnier, E., Bönisch, G., Westoby, M., Poorter, H., Reich, P.B., Moles, A.T., Dickie, J., Gillison, A.N., Zanne, A.E., Chave, J., Wright, S.J., Sheremet'ev, S.N., Jactel, H., Baraloto, C., Cerabolini, B., Pierce, S., Shipley, B., Kirkup, D., Casanoves, F., Joswig, J.S., Günther, A., Falczuk, V., Rüger, N., Mahecha, M.D., Gorné, L.D., 2016. The global spectrum of plant form and function. Nature 529, 167–171. https://doi.org/10.1038/nature16489

Gommers, C.M.M., Visser, E.J.W., Onge, K.R.S., Voesenek, L.A.C.J., Pierik, R., 2013.
Shade tolerance: when growing tall is not an option. Trends Plant Sci. 18, 65–71.
https://doi.org/10.1016/j.tplants.2012.09.008
Grime, J.P., 1977. Evidence for the Existence of Three Primary Strategies in Plants and Its
Relevance to Ecological and Evolutionary Theory. Am. Nat. 111, 1169–1194.
https://doi.org/10.1086/283244
Pontes, L. da S., Maire, V., Schellberg, J., Louault, F., 2015. Grass strategies and grassland
community responses to environmental drivers: a review. Agron. Sustain. Dev. 35,
1297–1318. https://doi.org/10.1007/s13593-015-0314-1
Reich, P.B., 2014. The world-wide 'fast–slow' plant economics spectrum: a traits manifesto.
J. Ecol. 102, 275–301. https://doi.org/10.1111/1365-2745.12211
Sakschewski, B., Bloh, W. von, Boit, A., Rammig, A., Kattge, J., Poorter, L., Peñuelas, J.,
Thonicke, K., 2015. Leaf and stem economics spectra drive diversity of functional
plant traits in a dynamic global vegetation model. Glob. Change Biol. 21, 2711–2725.
https://doi.org/10.1111/gcb.12870
Thonicke, K., Billing, M., von Bloh, W., Sakschewski, B., Niinemets, Ü., Peñuelas, J.,
Cornelissen, J.H.C., Onoda, Y., van Bodegom, P., Schaepman, M.E., Schneider, F.D.,
Walz, A., 2020. Simulating functional diversity of European natural forests along
climatic gradients. J. Biogeogr. 47, 1069–1085. https://doi.org/10.1111/jbi.13809
Wright, I.J., Reich, P.B., Westoby, M., Ackerly, D.D., Baruch, Z., Bongers, F., Cavender-
Bares, J., Chapin, T., Cornelissen, J.H.C., Diemer, M., Flexas, J., Garnier, E., Groom,
P.K., Gulias, J., Hikosaka, K., Lamont, B.B., Lee, T., Lee, W., Lusk, C., Midgley, J.J.,
Navas, M.-L., Niinemets, U., Oleksyn, J., Osada, N., Poorter, H., Poot, P., Prior, L.,
Pyankov, V.I., Roumet, C., Thomas, S.C., Tjoelker, M.G., Veneklaas, E.J., Villar, R.,
2004. The worldwide leaf economics spectrum. Nature 428, 821–827.
https://doi.org/10.1038/nature02403

---

## Author Response (AR1)

Review #1

General evaluation of the research paper

The paper presented by the authors addresses a very relevant and important topic in the field
of DGVM model development. For far too long, the representation of grasses and the
herbaceous layer have been given far too little focus in most DGVMs with respect to
structural and functional diversity. Only recently, development of more detailed grass layer
representations in DGVMs are starting to emerge but compared to tree-layer representation
this work is still at a comparatively early stage of development. Grassland ecosystems and
savannas cover a substantial fraction of the land surface and provide important ecosystem
functions and services to a multitude of people while simultaneously being threatened by the
effects of climate change and resource over-exploitation. Therefore, developing vegetation
models that are capable of representing within-grass layer dynamics, diversity and processes
is crucial to assess the impact of different management strategies and environmental change. I
therefore deem the paper a relevant and important scientific contribution.

The CSR theory is a widely known concept and therefore a valid approach to implement
functional diversity and trade-offs within the herbaceous layer of the model. One may
question whether the implementation in its current form using a Bayesian calibration method
to parameterize the new PFTs for three specific sites can be generalized for large-scale
application, but in the given context of the study, the approach seems sound and justified to
me. The shown results in many cases match ecological expectations and improve results
compared to the old model version, further corroborating the chosen approach.

The paper is well-written and clearly structured. I therefore recommend publication pending
minor revisions and clarifications detailed below.

We cordially thank the reviewer for their thorough and constructive feedback as well as the
positive evaluation of our manuscript.

Below we provide a response to all detailed comments including proposals to achieve the
suggested improvements.

Detailed comments

Introduction:

line(s) 36/37: You might also add the role of atmospheric $CO_2$-concentration. $CO_2$-
fertilization effects can shift the competitive balance in grassland communities in locations
where both C3 and C4 grasses are present.

We agree, even though we do not look into the effects of changing CO2 concentrations it
should be a part of this overview.

We added "atmospheric $CO_2$ concentration" to the list in L39 and added the sentence
"Atmospheric $CO_2$ constitutes the basic resource for photosynthesis and its rising
concentration can shift the competitive balance between C3 and C4 grassland species (ref.)."
in L39f line(s) 42: "high temperatures can lead to an increase of microbial decomposition". Only in combination with sufficient moisture. In arid regions, decomposition comes more or less to a stand-still during the dry season due to the water limitation that affects the microbial community. Rains at the beginning of the wet season then lead to peak emissions when microbial decomposition picks up again.

We added "if soil moisture levels are sufficient to permit the formation of microbial community." in L46f line(s) 44/45 "...may be beneficial for grassland productivity depending on its intensity". Maybe add: "by removing moribund plant material and triggering growth (over-)compensation."

We added "by removing moribund plant material and triggering growth (over-)compensation." at the end of the sentence in L50f.

line(s) 49: "for the species" – "for the functional types". I'd rather consistently keep the focus on functional types.

We replaced species with "plant functional types (PFTs) representative of species" in L54 and species with PFT throughout the manuscript when referring to the modelling approach and simulations results. However, we kept the term species when describing the theoretical background and results of field experiments to remain precise.

line(s) 52: "indirectly through alterations of the resource limitations" – add: "…that can cause shifts in the competitive balance between functional types".

We added the suggested phrase in L56.

Methods line(s) 105: "hot-steppe pasture in South Africa": this is a somehow unusual terminology / vegetation classification. The Syferkuil site usually is referred to as savanna rangeland in other publications.

The terminology for the naming of all sites was derived from the Koeppen Geiger climate zones (in this case hot steppe). At the first mention we decided to add the form of grassland management (pasture). We therefore kept the naming as is but added a phrase pointing towards the term savanna rangeland.

We replaced "hot steppe" with "savanna rangeland" in L111 and added "[…], following the Köppen-Geiger climate classification (Kottek et al., 2006)" in L113f.

line(s) 107/108: That means no tests of fertilizer X defoliation intensity combinations? That could be another interesting experiment to add, at least for the simulations.

Thank you for this interesting suggestion. In this part of the manuscript, we only mention the managements for which experimental data were available and that could therefore be used to parameterize the sites. Not knowing experiments including fertilizer X defoliation combinations, we would be grateful for information and very interested to include such data and combinations in further studies. The additional scenarios are described in 2.5. With this separation we distinguish between the scenarios that were predefined by the data and those we selected for further analysis. When defining the scenarios for further analysis, we decided to use extreme cases to test the effect of different limiting resources (e.g. infinite nutrient availability) instead of choosing different fertilizer levels. Regarding the defoliation intensity, we agree that analyzing a gradient of different intensities provides another interesting experiment. However, we decided to put our main focus on the resources and believe that the defoliation intensities of the experiment already cover a sufficient range.

line(s) 115/116: Are the trait values you use to describe the strategies from within a continuous range, or discrete fixed values? For example, if you use SLA as a trait to distinguish between acquisitive and conservative strategies, then you will automatically cover the extremes as well as in-betweens if you allow SLA to be a continuous trait that can range between a minimum and maximum value (see, e.g., Scheiter et al., 2013, Langan et al., 2017).

The reviewer raises a very interesting point. LPJmL-CSR follows the concept of using a small number of PFTs with fixed parameters. Therefore, for example SLA is fixed and each PFT only covers one point of the continuum. We also see the potential for interesting future work following an individual based approach drawing trait values from a continuum similar to LPJ-FIT (Sakschewski et al., 2015) or aDGVM2 (Scheiter et al., 2013). However, the currently implemented management routines of such models are less detailed compared to "classic" DGVMs that include an agricultural component. We therefore see the necessity to continue to improve grassland representation in both model types for the foreseeable future.

We picked this up in the discussion L813-825 (see also reply in L760-783).

line(s) 120 "Overview of managed grasslands in LPJmL" – "Overview of managed grassland representations in LPJmL" seems a more fitting title for this section.

We changed the title of section 2.1 to the reviewer's suggestion.

line(s) 123/124: one polar, one temperate and one tropical grass: C4-type photosynthesis for the tropical grass? Knowing classic LPJ, I deem it likely that this is the case, but good to mention explicitly.

We thank the reviewer for pointing this out. Indeed the tropical grass is a C4-type.

We added the photosynthetic pathway to the listing of the different PFTs in L131.

line(s) 130/131: (no water limitation, ref). – forgot to add the actual reference here.

We added the reference to (Jägermeyr et al., 2015) in L156.

Table 1: Forage supply [MgDM ha-1]: Terminology not entirely clear: Peak standing biomass? Annual withdrawal quantity (through mowing / grazing)? What is the temporal reference frame – annual?

We added the time to the unit in Table 1 and changed the terminology to forage offtake which we define in L35f.

line(s) 166-168: Does this new scheme also account for root biomass distribution in different soil layers, and therefore varying water availability between different soil layers? So that the total water uptake is the biomass-weighted uptake sum across soil layers? Or is it simpler than
that?

We thank the reviewer for pointing out that this could be described more clearly. Root
distribution between different soil layers was already used to determine the water supply from
the different layers in the previous model version (Schaphoff et al., 2018). Our scheme retains
this approach and only distributes the sum over the supply from all soil layers based on the
root biomass. We included this in the explanation of our approach.

We added "First, the PFTs access to water from different soil layers is calculated as described
in Schaphoff et al. (2018)." in L195 and replaced "The new parameter ($k_{root}$), which is a
proxy for root properties associated with morphological properties of the root network (e.g.
branching and spread)." With "Second, the amount of water available for the PFT is
determined considering its root biomass and the new parameter ($k_{root}$), which is a proxy for
root properties associated with morphological properties of the root network (e.g. branching
and spread)." in L196-198.

line(s) 186: I suppose that means that SLA as a trait is a PFT-specific constant? I.e., it cannot
vary over the lifetime of individual, or between different individuals of the same PFT?

Yes, it is a constant but as stated in our reply to a previous comment (L89-97), we agree with
the reviewer that there is great potential in exploring the entire continuum.

We now discuss this in L813-825 (see also reply in 760-783).

line(s) 191/192: Does LPJmL distinguish between forbs and grasses, and if so, how is this
implemented? And for grasses: does it distinguish between C3 and C4 photosynthetic
pathway? Is age-mortality the only reason for mortality, or are there other causes
implemented as well (e.g., due to negative annual C-balance, due to water stress, due to fire,
etc.)?

LPJmL does not distinguish between forbs and grasses and the herbaceous PFTs can include
both. C3 and C4 photosynthetic pathways are distinguished and we added a description in the
methods section. In addition to age mortality, the model checks if a PFTs overall root or leaf
biomass becomes negative and kills the respective PFTs. Excessive water stress from
prolonged drought may be a cause of this. However, additional causes of mortality from water
stress such as embolism (Jacobsen et al., 2019) as well as heat stress are not included. Fire on
managed grassland has been implemented both as a disturbance (unpublished) and a
management practice (Brunel et al., 2021) but is not considered here. We extended the section
on mortality to provide this additional information.

We added the following phrases and sentences:

•   "that do not distinguish between forbs and graminoids:" in L131.
•   "The only additional cause of mortality was negative leaf and/or root biomass after
allocation as a result of prolonged stress. While this may be caused by water stress,
additional causes of mortality from water stress such as embolism (Jacobsen et al.,
2019) as well as heat stress were not considered." in L231ff.
•   "We did not implement additional causes of mortality such as embolism." in L253f.

line(s) 193: "a biomass increase of the average individual dependent on the available area" –
rephrase? "the area-specific biomass increase of the average individual"

Using "area-specific" as suggested by the reviewer is in our opinion less explicit since it does
not define which area. We instead replaced "available area" with bare ground area.

In response to a comment of reviewer 2 we amended substantial parts of section 2.3.3 and
removed the respective phrase. In the updated section 2.3.3 we use the term bare-ground area
as suggested L235, 244 and 246.

section 2.3.3: general question on mortality: does the model distinguish between annual and
perennial herbaceous PFTs? I.e., do you have a PFT with enforced death after one growing
season? Enforcing annual types should implicitly strongly select for fast resource acquisition
at the expense of durable structural components, and a strong focus on reproductive
performance (see, e.g., Pfeiffer et al., 2019).

Currently, LPJmL does not explicitly distinguish perennial and annual PFTs and death is not
enforced at any time. Implicitly, the establishment as well as the mortality rate control the life
cycle of the PFT. High establishment and mortality rates lead to a fast turnover of the
population. We see potential in explicitly distinguishing annual and perennial PFTs for
example through constraining the period of establishment for annuals to the growing season.

We added "Another important aspect in savanna and other dryland ecosystems is the
distinction between annual and perennial plants. In LPJmL, this distinction is not explicitly
made. While the R-PFT has a higher replacement rate of average individuals, it is not
constrained to a specific growing season, after which it is completely killed to be
reestablished the following growing season. Incorporating this distinction into the model is an
option to add additional functional diversity and will likely improve model results." in L842-
846.

line(s) 197: "we retained the approach of establishing saplings instead of seeds" – I assume
that refers to the tree PFTs? A bit unusual to refer to establishing grasses or herbs as
"saplings". I assume that you must have excluded tree PFTs from the simulations of the
grassland sites, allowing grasses/forbs only? Otherwise, it is likely that a forest type or
savanna type would have established as potential natural vegetation at least at the German and
South African sites. You should add the information of how you handled the tree component
of the model in the section where you describe your simulation protocol. Also clarify how
establishment is done specifically for the grasses / herbaceous layer.

Indeed, only herbaceous PFTs are allowed to establish on managed grassland stands. We
added this to the model description. We agree with the reviewer that the term sapling is
misleading in this context and replaced it with the term seedling throughout the manuscript. In
addition, since this may create some confusion regarding the sapling LAI parameter, for
which we had to keep the term, but explained the origin of the parameter name and its
purpose.

We added

•   "Tree PFTs, which are also part of LPJmL, were not allowed to establish on managed
grasslands and all further descriptions provided here of or related to PFTs only
concern herbaceous PFTs." in L132ff.

• "While seedling is the more intuitive term for herbaceous plants and we will use it
throughout the manuscript, the subscript in the parameter name refers to saplings
because it was adopted from the tree PFTs in the past. " in L312ff.
• And replaced "sapling" with "seedling" throughout the manuscript.

line(s) 199/200: So just to make clear that I understand correctly: the average individuals are
clones, i.e., all of the same PFT, but you introduced the clone-concept to be able to account
for PFT-specific reproduction aspects, such as seed numbers, germination rates, and seedling
survival probability? If so, you should make it clearer than it is currently. It goes in the
direction of the problems faced by models that simulate actual, true individual plants and their
reproduction and establishment.

The reviewer raises an important point here. Indeed, the concept of the average individual
should be explained in more detail to prevent confusion with individual based approaches. We
added a section in the methods explaining that each PFT can be seen as a representative for a
population with certain attributes that describe the population (e.g. number of average
individuals, individual biomass). In addition, we discussed our approach in comparison to an
individual based approach to show advantages and disadvantages.

We added "In LPJmL, each PFT represent an entire population of adult plants using the
concept of average individuals. The PFT describes the carbon and nitrogen stocks of the
leaves and roots of an average individual and the number of average individuals in a
population. It follows, that the carbon and nitrogen stocks of the population can be determined
by multiplying the average individual stocks with the number of average individuals.". in
L136-139. We replaced "As a consequence, all grasslands that are not located at the border
between climatic regions were simulated using only one of these PFTs to represent
herbaceous vegetation." with "Carbon and nitrogen stocks as well as the number of average
individuals are dynamically calculated each day from the simulated processes which are:" in
L139f and added "Prior to our implementation, each herbaceous PFT was represented by one
average plant individual." to L150.

line(s) 203: age-dependent mortality: hard set (at a specific age), or based on an age-
dependent likelihood? And: the age-dependency differs between the different strategy types?

Thank you for this comment. Actually neither is the case. Depending on the growth
efficiency, the number of average individuals is reduced (Appendix A3 L912-951). Actual
mortality is derived from the maximum mortality rate - which is the same for all strategy
types - and the growth efficiency. The growth efficiency is dependent on SLA, which differs
between the strategy types (Appendix A3 Eq. A10). We extended the description in Appendix
A3.

We replaced "[…] that day." In L943-947 with "since the last allocation and $C_{ind,PFT}$ is the
biomass increment from photosynthesis since the last allocation. The growth efficiency $\Delta bm \cdot$
$C_{ind,leaf,PFT}^{-1}$ is the ratio of the net carbon change and the carbon stock of the leaves, which is
lower for old plants. The SLA influences the maximum age of the different strategies
assuming that plants with a low SLA and faster metabolism reach a lower age compared to
high SLA plants.".

And what is the allowed maximum number of average individuals, and the maximum number
of grass-layer PFTs that can now coexist within one grid cell?

We thank the reviewer for this question. It made us realize that we did not include this in
Appendix A3. There is no hard maximum number of individuals. However, if the total
number of individuals exceeds 250 /ind/m2, 5% of the individuals die. We added a qualitative
description in the method section and update the equations and explain the underlying
reasoning in Appendix A3. The number of PFTs per grid cell is in theory not limited, however
we decided to use one PFT for each main strategy for the purpose of this study. For future
studies this number can be increased, however this will also increase the computation
requirements.

We added

• "In theory, however, the number of PFTs that could coexist within a grid cell is not
limited." In L135f.
• "In grasslands with a high growth efficiency and frequent defoliation establishment
may lead to a continuous increase of the number of average individuals. To avoid
numerical errors that could results from this, we prohibit the number of average
individuals to exceed 250 $Ind.\cdot m^{-2}$." In L950f.

line(s) 205/206: "It can be assumed that few individuals that maintain a high cover and
biomass must be larger…" – I assume all individuals that are part of one PFT have the same
size and biomass, given that you are still using the average individual concept? So, adding
new young individuals will lower the size and decrease the age of all clone individuals within
the PFT due to the averaging. But this implies that a strongly reproduction-oriented PFT
strategy would automatically have a smaller average individual size, a young average age, and
a larger number of clone individuals representing the PFT. This has implications for the age-
dependent mortality, as highly reproductive strategy types are then less likely to reach the age
where age-dependent mortality hits. Did you consider this aspect?

The reviewer raises an important point. We do not simulate the age of the average individual.
Our implementation of mortality depends on the growth efficiency. This describes the change
in carbon from photosynthesis and turnover per average individual compared to the average
individual carbon pools. In this ratio, the number of average individuals cancels out and the
key aspect is the GPP to turnover ratio, which should be smaller in older populations leading
to a higher mortality. We included this explanation in the method section on the mortality.

We replaced "age dependent individual mortality" with "age-mortality" and added "The
growth efficiency is the ratio of the net change in the individual carbon stocks (the result of
net photosynthesis and turnover) and the individual carbon stocks. Assuming that old plants
grow more slowly this is used as a proxy for population age and resulting age-mortality." in
L251ff.

Table 2: Maybe add a column that specifies the predominant gradient associated with the
parameter. You mention it in the text of this section, but it would be helpful to also have it as
a brief overview in the table. I find the distinction between biotic and abiotic dimension a bit
arbitrary/confusing with respect of the definition. Referring directly to the respective gradient
(stress gradient for biotic, disturbance gradient for abiotic) would seem more intuitive for me.

We abandoned the terminology abiotic and biotic gradient. When writing the original draft,
we found that it provides a clear distinction between the parameters related to each gradient.
However, as the reviewer correctly noted, this creates an additional layer of terminology to
understand when reading the manuscript.

We abandoned the terms "abiotic dimension" and "biotic dimension" and instead now directly
refer to the "stress gradient" and the "disturbance gradient" throughout the manuscript.
Additionally, we replaced the dimension column in Table 2 with a column that provides
information on the predominant gradient.

Table 2: Hierarchy: How did you determine the hierarchy? Based on your expert assessment?

We added "based on our expertise" in L378.

Table 2: Light extinction coefficient: Independent from SLA, or correlated? High-SLA leaves
should have more transmission than low-SLA leaves.

We agree with the reviewer that transmissivity of single leaves and their SLA are correlated.
However, we had to deal with the challenge that LPJml does not simulate multiple leaf layers
and cannot distinguish between the transmission of single leaves and the entire vegetation
layer. To account for the difference between leaf and entire vegetation transmission at least
implicitly, here the light extinction coefficient is not a measure of the transmissivity of a
single leaf. Instead it is the transmissivity of the entire vegetation layer of a PFT. Therefore,
we assume that PFTs, which have a high SLA can still have a high light extinction if many
high transmissivity leaves are stacked. In the current version of the manuscript this is only
touched upon in the discussion (L663-666). We now describe this in more detail in the
methods section.

We added "We assumed all parameters to be independent from each other. While we are
aware that $SLA$ and the light extinction coefficient $k_{beer}$ are correlated in reality because the
transmissivity of leaves increases with $SLA$ we have to treat them as independent because in
LPJmL, the light extinction coefficient does not describe the transmissivity of a single leaf but
of the entire vegetation layer. Stacking a high number of high transmissivity leaves may result
in the same light extinction compared to a lower number of low transmissivity leaves. In
LPJmL-CSR, a similar $k_{beer}$ would be assigned for both cases because it represents the light
extinction coefficient of the entire vegetation layer." in L338-344.

Table 2: Maximum transpiration unit [mm] – if this is to be a rate, then the time part of the
unit is missing. [mm/day]?

We changed the unit to $[\mathrm{mm\,d^{-1}}]$.

line(s) 237/238: The root efficiency coefficient does affect the competitiveness between plants
(biotic interaction), but it also relates to the stress gradient (abiotic) with respect to water
uptake capacity. This is an example illustrating why using "biotic" and "abiotic" as
dimensions is maybe not the best way to make the distinction.

We agree that there are cases were the distinction between biotic and abiotic is not so clear.
As already stated earlier (reply in L281-288) we abandoned the terms and only retain the
terms stress and disturbance gradient.

line(s) 240/241: The light extinction coefficient describes the fraction of light intercepted by
each additional leaf layer, right? As the amount of light that can transmit a leaf layer depends
on the thickness of the leaf, one would expect kbeer to be correlated with SLA, which, unlike
kbeer, you define as abiotic dimension. It would be good if you sort this out more clearly.

We agree and refer to our proposal from the related comment in the reply in L293-311. We
now also describe more clearly, which parameters play a role for the stress or the disturbance
gradient or for both gradients.

In addition to the changes described in our reply in L293-311, we added a column for the
subsidiary gradient in Table 2.

line(s) 241/242: the leaf area index of a sapling represents the offspring size - What do you
define as "offspring size"? The height of the offspring, or its starting biomass, or its projected
foliar coverage? I'm not sure LAIsap is a good description of offspring size, as its meaning is
rather vague without a clearer definition. Whether a seedling/sapling of given leaf biomass
has a high or low LAI is a function of its SLA, so LAIsap for a given unit of leaf biomass
essentially is nothing else as another way to refer to SLA.

In LPJmL, the leaf area index of a sapling is only used to calculate the sapling biomass using
SLA. So instead of assuming a given leaf biomass, we assume a given SLA and calculate the
leaf biomass. Using the same SLA, a higher sapling LAI is equal to a higher sapling biomass.
We changed offspring size to offspring biomass and added an explanation of the relationship
to SLA. We also revised the discussion to reflect both SLA and sapling LAI when discussing
offspring biomass.

We incorporated a more accurate description in the overview of the parameters in section
2.4.1 which contains the following sentence "In LPJmL 5 and in LPJmL-CSR, it is used to
calculate the above-ground biomass of a seedling using the PFTspecific SLA". in L333f.

Table 3: Flip order of columns "variable" and "site", as site is unique and variable is tied to
site and non-unique.

We swapped columns variable and site.

line(s) 287/288: "the current representation of some processes within the model" – which
processes specifically?

We here refer to section 4.1.2 where these processes are listed. We changed "some processes
within the model" to "the processes, listed in sect. 4.1.2," and removed the reference to
section 4.1.2 at the end of the sentence (L389f).

line(s) 299: 390 years - your spin-up duration? Did you add a transient simulation period after
the spin-up (how long? For what time-period?). One can only guess based on the time-axis
labeling in the figures that follow in the results section. Please specify this with some more
detail.

We agree that additional information is needed. We first conducted a potential natural
vegetation spin-up simulation of 30000 years followed by a spin-up including land use of 390
years after which the transient simulation start. We added the following to the modelling
protocol section.

"Before simulating managed grasslands, the model was run for 30000 years with natural
vegetation to obtain an equilibrium of the carbon and nitrogen cycle during a spinup
simulation. Afterwards, a second spinup of 390 years was conducted to account for the effects of historical land-use change on soil conditions." in L402ff and replaced "390 years" with
"the second spinup period" in L406.

Modelling protocol: What is the temporal resolution the CSR-model version runs on?
Monthly, or daily?

All processes are executed on a daily time scale. We also compute the outputs on a daily
timescale but aggregate to a monthly or annual resolution for some of the results.

We added "LPJmL-CSR simulates all processes and provides all outputs with a daily
resolution. If necessary, outputs are aggregated to a monthly or annual resolution in the
postprocessing." in L401f.

How do you initialize community composition with respect to present PFTs and shares of
PFTs at the beginning of the simulation? Based on the field-based observations? If so, how
would you do it in a situation where you did not know the field situation of sites, e.g., for a
large-scale or global simulation? (Question for the discussion, I guess).

Upon initialization, each PFT is established dependent on the respective establishment rate
and biomass (derived from sapling LAI, SLA and leaf to root ratio). Therefore, initially a PFT
with high values in both has a higher share in the community. However, if its strategy is not
suitable this will change over time. This means, that no data on initial community
composition or similar is needed. We added this explanation to the model description.

We added

• "The initial community composition is not prescribed. Instead, upon initialisation,
each PFT is established based on the PFT-specific establishment rate and offspring
biomass (sect. 2.3.3 and 2.4.1). The community composition during each time step
emerges from the competition for resources dependent on the processes described
above." in L150-153.
• "Furthermore, in LPJmL-CSR the initial community composition is not dependent on
additional data which facilitates the application at different sites or at larger scales." in
L606f.

Results

Figure 1: Please specify temporal reference frame for panels a, d, and g - is it the annual sum
(yield), the peak season leaf biomass (leaf biomass), the grazing period duration offtake
(grazing offtake)?

We added the units to the caption and added the temporal dimension of the unit to the
subtitles in the figure.

General question on all scenarios that included animal grazing: Is preferential grazing, i.e.,
selection of more palatable over less palatable PFTs, accounted for by the new CSR model
version? Unlike mowing or biomass removal by fire that is indiscriminate, biomass removal
by herbivores can alter community composition quite substantially, especially under high
grazing pressure. If preferential grazing is not yet implemented, this should be added as a
limitation in the respective section of the discussion, and could be pointed out as a future need
for development.

The reviewer raises an important point. Indeed the current implementation (Rolinski et al.,
2018) does not consider preferences for specific PFTs. We now briefly mention this when
describing the model and discuss this in the section on future need for development.

We added

• "In this study, we use the mowing and the daily grazing option. The daily grazing
option does not account for animal preferences (Rolinski et al., 2018)." in L154f.
• "Plant species have adapted to grazers in manifold ways, one of which is grazing
avoidance by being less or even unpalatable. This is a successful strategy in grazing
systems because in contrast to mowing, which is indiscriminate, grazing animals show
preferences for plants with a higher palatability. Selective grazing and grazing
avoidance through palatability are currently not represented in LPJmL but can have a
strong effect on the community composition (Newman et al., 1995; Parsons et al.,
1994). Including preferences for example for high SLA PFTs may improve simulation
results further." in L851-856.

line(s) 365-368: Ecologically, the shift towards more investment into above-ground biomass
(growth (over-)compensation) and towards a more resource-exploitative strategy
(construction of "cheaper" leaves with reduced life duration is plausible. However, I do not
see right away why the minimum canopy conductance should decrease due to grazing?

We agree that the decrease of the minimum canopy conductance is unlikely to be related to
grazing directly. More likely, the high and similar minimum canopy conductance of the
ungrazed scenario (C0) is an artefact of the parameterization. All parameters can be assigned
primary and secondary processes that they affect. The leaf to root ratio and the SLA are
different in the two scenarios and act as a compensation of defoliation from grazing (primary
process). However, to some extent these parameters also control access to and distribution of
resources (secondary processes). In the ungrazed scenario, these do not need to be adjusted to
compensate for the defoliation but can still play a role in the competition for water. Therefore,
more parameters can control resource access and distribution and it is likely that this will
affect the parameterisation of minimum canopy conductance.

We included a thorough description of the processes controlled by each parameter in section
2.4.1, L284-306 and L315-335 (see also reply in L1019-114). We extended section 3.2.1 in
L477-482 by "However, this is likely an artefact of the parameterization. As stated in sect.
2.4.1, both $SLA$ and $lmro$ do not only underpin the compensation of defoliation but can also
play a role for resource uptake and distribution. In the ungrazed scenario (C0), no defoliation
has to be compensated and both parameters are only needed for their secondary role for
resource uptake and distribution which likely affected the selection of $g_{min}$. In contrast in the
grazed scenario (C1), they are needed for their primary role and $g_{min}$ and $E_{max}$ become more
important for resource uptake and distribution.".

line(s) 406/407: How does the relative contribution of the S- and R-PFT to the forage supply
compare to their relative abundance or relative contribution to FPC? I.e., did they contribute
more or less than could be expected according to their relative abundance within the
community?

Thank you for the interesting question. We did not look into this in detail. Since biomass is an
important variable when calculating FPC, we believe it is likely that forage supply and
growing season FPC are similar. However, there might be differences when averaging over the entire year. We analyzed our results regarding this and the results confirm our hypothesis
that above-ground biomass and FPC are similar. This is not surprising since above-ground
biomass is used to calculate the FPC. Proportional differences between the PFTs' FPC closely
resemble differences in their above-ground biomass. Deviations are a results of the PFTs'
different SLA and $k_{beer}$ values. We believe that adding this will not provide any additional
value to the manuscript and therefore did not make any changes.

line(s) 442/443: "In the irrigated scenario, only the S-PFT contributed to forage supply." -
That is a bit surprising? One would expect that irrigation reduces stress resulting from water
limitation, therefore opening the community more strongly for the C-PFT.

This was also surprising and counterintuitive to us. We already provide an explanation in the
discussion in L706-710, which we now reference to in the sentence in L555f.

line(s) 473/474: "…still dominated by the S-PFT." - Is this a legacy effect from the pre-
irrigation time period's community composition? If run long enough without resource
limitation (i.e., with irrigation on), would the S-PFT type be replaced by the C-PFT type, and
if yes, how long do you expect this would take? Can be part of the discussion, if not already
discussed there.

We already touch upon this in L666-669 but agree that this can be discussed in more detail.
We added a reference in L589 and extended the discussion in section 4.1.2.

We added

•   "Whether or not this is the new equilibrium state or the community is still
 transitioning is crucial (sect. 4.1.2)" in L589.
•   "Less than two years is a very fast transition and while the shares of the leaf biomass
 seem to have reached a new equilibrium after one or two years of irrigation, it is likely
 that the soil carbon and nitrogen pools are not in equilibrium yet. This is especially
 interesting when considering that the overall increase in leaf biomass may promote
 litterfall and the formation of inorganic nitrogen. This in turn may lead to reduced
 nitrogen limitation and additional changes in the community composition.
 Furthermore, biological nitrogen fixation is dependent on soil moisture and may
 therefore also contribute to decreasing nitrogen stress under irrigation. However,
 irrigation also leads to increased leaching and could therefore also decrease inorganic
 nitrogen availability." in L673-679.

Discussion

General remark: how do you intend to use the CSR-model in the future, if you ideally need an
a-priori determination of the ideal PFT parameterization depending on site, community, and
management? And how can communities respond to changing management or environmental
conditions if the parameterization of the PFTs cannot be dynamically adjusted during the
simulation based on a selection mechanim that filters for the best-suited parameterization
under the given circumstances?

The reviewer raises several interesting questions that go beyond this study. We are currently
working on a globally applicable set of PFTs, which will form the basis of another study in
the near future. For that study, we retain the fixed PFT parameterization of classic DGVMs.
However, we are generally open and very much interested in further developing the model.

As already mentioned in the reply in L89-97, it would be very interesting to combine the
approach of LPJmL-CSR and aDGVM2 or LPJ-FIT.

line(s) 494/495: "IN LPJmL-CSR, growth of the vegetation was faster than in LPJmL 5.2,
which led to higher yields for all cuts." – Elaborate briefly on the causes for the faster growth
in the new model version.

The faster growth compared to LPJmL 5 has two reasons: First, the new implementation of
biological nitrogen fixation led to less nitrogen stress and higher photosynthesis. Second, this
is also a result of the new parameterization, which was tailored to this site.

We added "We identified two reasons for the faster growth. First, the new implementation for
biological nitrogen fixation (Appendix A4) reduced nitrogen stress and promoted higher
photosynthesis rates. Second, while the parameters used for LPJmL-CSR were tuned for
performance under the site specific environmental conditions and management, the
parameters used in LPJmL 5 were defined for large scale simulations with different
management." in L611-614.

line(s) 504: "but selected a livestock density of 1.0 cows ha-1" – use "livestock units" rather
than cows (how about steers, heifers, etc.); And: Is this to determine the amount of manure
input? The temperate grassland was not grazed but mowed, so livestock density does not
make much sense with respect to grazing off-take?

The livestock density refers only to the spin-up and the historical periods for which no data on
actual land use were available. Therefore, it is entirely unrelated to the transient simulations
that reproduce the mowing experiments.

We replaced cows with LSUs and "[…] that […]" with " […] for the land use spinup
simulation (see Sect. 2.5 and SI) to prescribe a fixed grazing pressure, which […]" in L623f.

line(s) 506: Briefly describe the processes / mechanisms that lead to increased carbon input to
the soil in the CSR-version compared to the old version.

We identified three causes for the increased carbon input: First, the SLA longevity trade-off
we implemented led to an increase in turnover supplying more carbon to the litter layer.
Second, implementing explicit mortality of average individuals created an additional input
into the litter layer. Third, accounting for the carbon added through the application of manure
fertilizer also constituted an additional carbon input into the system.

We added "The increased soil carbon input had three reasons. First, the trade-off between
SLA and leaf longevity lead to higher turnover rates and in turn higher litterfall compared to
LPJmL 5. Second, accounting for mortality explicitly constituted an additional input into the
litter layer. Third, our simulation included manure application which provided an additional
carbon input into the system." in L626-629.

line(s) 526/527: Here finally the information that I was missing in the methods section. You
should add this information to the modeling protocol section (that you did exclude the tree
PFTs from your site-scale simulations.

We adopted this suggestion see reply in L188-201.

line(s) 528/529: You should try to give a reason for the "why" of this, instead of simply repeating the result. For example, an explanation could be that grazing was not the only / the main stress for herbaceous vegetation at this savanna site. The site has a pronounced dry-vs-wet season dynamics, and therefore water limitation as a stress factor, maybe also N-limitation, may be causes for the dominance of the S-type irrespective of the grazing management.

We agree with the reviewer that this should be explained and share their opinion of the underlying reasons. We added a sentence to explain the dry wet dynamics of the site and that these are independent of grazing, which therefore does not affect the water stress level allowing the S-PFT to remain advantageous.

We added "The dominance of the S-PFT independent of grazing is plausible considering the pronounced dry vs. wet season dynamics at the site that impose water stress and potentially also nitrogen stress." in L651ff.

line(s) 540/541: You could test this by specifically allowing no other PFT than the S-type to enforce a monoculture.

We discussed the possibility to investigate this further, but decided against because LPJmL would limit us to simulating an S-PFT monoculture already before the beginning of the irrigation, which would likely lead to different initial conditions when starting irrigation. This would make it difficult to interpret the results.

line(s) 544/545: Was your simulation time period with irrigation long enough to allow establishment of a new steady state with respect to community composition? In my experience, community composition shifts are one of the slower processes and can take quite a number of years before reaching a new steady-state after a change in forcing has occurred.

We touch upon this in section 3.4.2 L587f by saying that "the transition occurred within the first one to two years", which is much faster than we would expect. We mention this when discussing the change in soil organic carbon (L656-662) but we agree that this is very brief. We now added more detail and highlighted the transition time more prominently. We also provided an explanation for the fast transition, which was related to the removal of competition for water. In a water scarce environment, the S-PFT as a water saver was advantageous and the C- and R-PFT were subordinate. Under irrigation, the S-PFT's slow growth becomes a disadvantage and the C- and R-PFT can exploit resources more efficiently. Both increase their biomass rapidly until a different limitation prevents further increase, while the biomass of the S-PFT remains similar. This is comparable to real world ecosystems. However, existing individuals cannot grow infinitely and need to reproduce producing new individuals. This process of reproduction and dispersal may slow down the transition. In LPJmL, the PFTs increase their biomass independent from the establishment of additional individuals which speeds up the transition.

We added

- "LPJmL does not simulate seed bank formation and reproduction is not limited by the amount of seeds available in a seed bank. Instead, the establishment depends on the bare ground area and the PFT-specific establishment rate." in L234f.
- "Regardless of the finality of the transition, its velocity is likely overestimated by LPJmL for two reasons. First, the C- and R-PFT can establish quickly despite their limited presence before the onset of irrigation because LPJmL does not simulate a
seed bank which would in reality be small at least for the C-PFT limiting its
establishment. Second, in reality growth of established individuals is limited and
reproduction and dispersal, which slow down population biomass increase, are needed
for such a transition. In LPJmL, already established individuals continue to grow and
the population biomass increases even without additional establishment." in L680-685.

line(s) 545/546: "However, periods of drought can induce and additional disturbance." –
Correct, but not in this case, because due to the irrigation you had drought eliminated.

The reviewer is correct. A plausible explanation is that the parameterization allows the R-PFT
to coexist with the C-PFT if the main resource limitation is removed.

We replaced "However, periods of drought can induce an additional disturbance (Wang et al.,
2019) creating a niche for R strategists (Kooyers, 2015; Norton et al., 2016)." with "The
success of both the C- and the R-PFT is likely determined by the similarity of their $SLA$, $k_{beer}$
and $lmro$ which become more important compared to $E_{max}$ and $g_{min}$ if there is no water
limitation. Potentially larger differences in these parameter would lead to the success of one
of the two instead." in L669-672.

line(s) 549: "LPJmL 5.3 underestimated the observed forage supply…" – I'm not sure about
your usage of the term "forage supply" (generally throughout the manuscript) - is forage
supply, according to your definition, the potentially available biomass offered by the
rangeland, or do you actually rather mean "the amount of feed required by the animals"
(which should then be termed as "forage demand"?

We agree that our use of forage supply was ambiguous because we use it to define the amount
of biomass removed through mowing or grazing for the temperate grassland and the cold
steppe but also for the amount of leaf biomass available for grazing for the hot steppe. This
was an attempt to use common terms for all sites, which appears to be confusing instead of
helpful. We therefore changed the term forage supply to forage offtake for the temperate
grassland and the cold steppe and use the term leaf biomass for the hot steppe.

We added a definition for forage offtake in L34ff and replaced "supply" with "offtake" for the
temperate grassland and the cold steppe and "forage supply" with "leaf biomass" for the hot
steppe.

line(s) 552/553: I do not understand: how does feed demand change forage supply? Forage
supply is a biomass potential offered by the plant community. Increased feed demand, as
described here by your correction, should not increase the forage supply of the plant
community (unless through growth overcompensation), but rather reduce the supply due to
the increased demand from the animal side?

As in the previous comment we acknowledge that using the term forage supply creates some
confusion which we resolved as stated in the reply in L601-609.

line(s) 554/555: The fact that animal demand could not be met AND above-ground biomass
collapsed is a rather clear indication of over-grazing / exceeding of rangeland carrying
capacity. In this context, maybe also discuss changes in the PFT community composition, i.e.,
changes in the prevailing strategy types. It can be expected that such a shift in strategy types
occurs under such circumstances.

We agree with the reviewer that the model results provide strong evidence for overgrazing and added a phrase explicitly stating so. We also added a sentence discussing the change in community composition which shows an increase of the C-PFT (and also to some extent the R-PFT) as shown in Fig SI 9 and 12.

We added "indicating overgrazing" in L693 and "Additionally, LPJmL simulates a different community composition compared to the low grazing intensity. The relative share of the C- and to some extent also the R-PFT is higher for the high grazing intensity (Fig. SI 9 b and 12 h) because such strategies are better suited to tolerate grazing." in L694ff.

line(s) 562/563: You did not combine fertilization with irrigation, right? Do you expect that fertilization in combination with irrigation would increase leaf biomass beyond the level reached with irrigation alone?

Generally, irrigation alone already affects processes related to inorganic N inputs and losses. Biological N fixation and mineralization increase with increasing soil moisture. However, irrigation also leads to higher leaching. We therefore expect that the PFTs are still N limited even though irrigation may already increase but could also decrease inorganic N availability. Additional inorganic N from fertilization may remove the N limitation leading to an additional leaf biomass increase but may also lead to higher maintenance respiration limiting leaf biomass growth. Therefore, we cannot give an unambiguous answer. We added this explanation in section 4.1.3.

We added "Similar to the hot steppe, it is possible, that our time frame is too short for the soil pools to have reached a new equilibrium. As described in Sect. 4.1.2, irrigation alone already affects processes that could increase nitrogen supply by biological nitrogen fixation and litterfall, but also decrease it by leaching. Both biological nitrogen fixation and mineralisation are dependent on soil moisture as well as on temperature which is low in the cold steppe limiting the increase of inorganic nitrogen. Therefore, it is possible that only an intermediate state emerges during our simulation period. Especially when also considering the increased leaching, we expect that the cold steppe is still nitrogen limited under irrigation, therefore combining irrigation with fertilisation could further reduce nitrogen limitation leading to increased productivity and changes in the community composition. However, the leaf biomass increase may also be limited by higher maintenance respiration which is connected to leaf nitrogen content. Additional analysis is needed to enhance the understanding of these complex interactions." in L723-731.

line(s) 575: "Fertilization had no effect on SOC" – Not surprising, given that fertilization without irrigation did not increase leaf biomass and therefore C-input to the soil.

We agree and added "because leaf biomass and in turn carbon inputs into the soil did not increase" in L715f.

line(s) 580/581: "it seems that an S-strategy remained advantageous" - Again, I wonder about the turnover time required by the model to let a community transition from one steady-state to a new steady-state.

While for the hot steppe we can provide clear evidence, that a new steady state was reached, for the cold steppe the reviewer raises an interesting point. Increased soil moisture from irrigation may lead to an increase of the NO3 and NH4 pools from mineralization and biological nitrogen fixation which may take longer than the simulated time frame (see also
reply in L558-585). We added this to the discussion.

See reply in L634-655.

line(s) 600: And it may be interesting how grass-tree coexistence (typical for savanna sites as
the one one in South Africa) will affect grass layer community composition compared to a
situation where trees are excluded from the simulation.

Indeed an improved representation of Savannahs would be a major step for DGVMs. In order
to achieve this, we see the need for additional model development as discussed in Rolinski et
al., (2021).

We added "Furthermore, the coexistence of tree and grass species, which is typical for
savanna sites, is not implemented in the LPJmL model. However, this is crucial to adequately
represent such ecosystems (Rolinski et al., 2021) and should be a focus of future model
development." in L840ff.

line(s) 606/607: "Generally, a change in resource availability does only change the conditions
for the establishment of a community but does not directly affect the established vegetation" –
Environmental filtering can also affect the established community by increasing mortality for
specific strategy types within the community, not only by changing establishment success of
given strategy type. Since you seem to have no other mortality causes aside from age-
dependent mortality in the model (at least not for the grass layer), you will not see this effect,
but it does exist, nonetheless.

We agree with the reviewer and extended this sentence to reflect the limitation of our model
to age mortality and to discuss potential effects of other causes of mortality.

We

• replaced "Generally" with "In LPJmL-CSR" in L754
• added "In reality however, a change in resource availability may also increase the
mortality for specific strategy types affecting the already established community as
well." in L756f.
• added "LPJmL-CSR only represents age mortality, i.e. the effects of mortality from
other causes such as frost, heat and embolism are not represented. Especially under
changing climatic conditions, specific strategy types may show increased mortality
and lose their advantage to the advantage of other strategy types. Including additional
causes of mortality may introduce additional trade-offs and enhance the differentiation
between strategy types." in L847-850.

line(s) 614: Why are N-fixers not separate PFTs in the model? I'm a bit surprised that they are
not.

Facing the challenge of adding new PFTs to a classic DGVM, our aim was to reduce
complexity as much as possible at first. This included restricting ourselves to add as little
PFTs as possible. Grouping N-fixers with non-fixers halved the number of PFTs. We believe
this is reasonable because the model will only fix additional N if the demand is not fulfilled.
In an approach with two separate PFTs, this would mean a change in community composition
and an increase of the N-fixer PFT at the expense of the non-fixer. In our approach, this simply means an increase in biological nitrogen fixation. One could say, that implicitly the
PFT is a fixer if needed and not if not needed and could determine this status using the
biological nitrogen fixation output. We added the necessary detail to the description of
biological N fixation in Appendix A4.

We added "While in reality, biological nitrogen fixation is a feature restricted to legume
species, in LPJmL we decided to not distinguish in fixing and non-fixing PFTs to keep the
number of PFTs as small as possible. This is reasonable because a PFT can be representative
of multiple species and will only fix additional nitrogen if its demand cannot be fulfilled by
other sources of nitrogen uptake and if its NPP is sufficient. One could say, the PFT has the
ability to fix nitrogen only if needed comparable to a community containing legumes only if
they are advantageous." in L970-974.

line(s) 622/623: So the assumption is that grazing is non-preferential, correct? I.e., grazers do
not favor one PFT over another, for example based on criteria that characterize palatability /
nutrition value. This is a simplification in the model that should be discussed briefly, as
herbivores usually do not function the same way as mowing (or fire) that removes biomass
indiscriminatingly.

Yes, grazing is not preferential. As stated in our reply in L406-419 we included this in the
model description and briefly discuss the limitations of the current approach.

line(s) 624: "tolerance or avoidance" – Avoidance would for example (aside from temporal
avoidance) be realized by being unpalatable. As your grazing is non-preferential, being a
grazing avoider type based on palatability would not make a difference in your model as the
animals would not discriminate against the avoider. This is a limitation you should mention.

We thank the reviewer for raising this point and included grazing avoidance through
palatability in the limitations together with preferential grazing (reply in L723f).

line(s) 629/630: "… and the PFTs had to follow a grazing-tolerance strategy." - The fact that
grazing avoidance can only be achieved through life cycle adaptation and not through
palatability likely causes a bias in your community composition. You should at least mention
this possibility.

We thank the reviewer for their suggestion.

We added "Because LPJmL does not account for differences in the palatability of different
strategy types the parameterization could not select for such likely successful strategies
leading to a potentially biased community composition." in L781ff.

line(s) 632/633: "At the cold steppe site, grazing only happened during the growing season
and both grazing tolerance an avoidance could be useful strategies." – Well, likely not
avoidance in the way you can represent it in the model (temporal avoidance). If grazing
happens during the growing season, and your only way to implement avoidance is through life
cycle adaptation, i.e., temporal avoidance, this will push avoiders to the non-growing season
as time when no grazing happens. But I don't see how avoiders could succeed by shifting their
existence focus to exactly the season when growth is not possible?

We added "However, grazing avoidance in time, which is the only type simulated by LPJmL
will not be successful as it would mean shifting biomass production to the non-growing season where the environmental conditions do not allow growth." in L785ff to acknowledge
that the model is not able to simulate the type of avoidance that is likely successful.

line(s) 643-645: This challenge could be circumvented by moving away from a PFT-concept
with fixed pre-defined parameter values for each PFT, which implicitly limits the number of
strategies that can be realized, for example by defining typical value ranges for the given
parameters of a strategy type. Within these continuous ranges, a strategy type can assume
many trait value combinations that define its location within the trait space occupied by the
strategy type, and therefore allows more plasticity within a strategy type, e.g., a plant could be
a moderate, intermediate, or extreme S-strategy type.

We agree with the reviewer, that moving away from the fixed PFT approach is a suitable way
to circumvent many of these issues. As discussed in previous comments one necessity is to
follow an individual based approach as in aDGVM2 or LPJ-FIT. We see this as a promising
and intriguing topic for future model development of LPJmL-CSR and emphasize this more in
the discussion.

We added "Generally, the approach of using a small number of PFTs with a fixed set of
parameters has been criticised (Quillet et al., 2010) leading to the development of next
generation DGVMs that apply an individual based approach such as LPJmL-FIT
(Sakschewski et al., 2015) or aDGVM (Scheiter et al., 2013). These models simulate the
competition between individual plants for which parameter values are drawn from predefined
ranges upon establishment. Given sufficient time, only successful strategies will survive. Such
models provide a much more nuanced representation of function diversity compared to classic
DGVMs with their coarse division into fixed PFTs but are also more computationally
substantially more expensive because of the high number of individuals for which all
processes have to be calculated. Past studies have therefore often focused on specific regions
such as the Amazon rainforest (Sakschewski et al., 2015), European forests (Thonicke et al.,
2020) or South African semi-arid rangelands (Pfeiffer et al., 2019). In contrast, classic
DGVMs are still widely applied on the global scale for example to calculate the global carbon
budget (Friedlingstein et al., 2022) and we see the need to continue their development for the
foreseeable future. Combining our approach of distinguishing between PFTs that follow the
main strategies of the CSR theory with an individual based approach making use of the full
parameter range instead of single points provides an interesting opportunity for future
research of diverse grasslands." in L813-825.

line(s) 645/646: The challenge will be to expand this site-scale-focused approach to a
generalized large-scale / global approach, because it will not be possible to parameterize
suitable PFTs for all imaginable locations and circumstances. I think the value of what you
show in this study is to prove that the CSR-concept can work within a DGVM and is
ecologically sound in many points. But to make it general, you will have to move away from
the discrete parameterization of your PFT approach, for example by allowing an evolutionary
approach that self-selects successfull strategies via environmental filtering from a pool of
potential trait value combinations, where each trait is represented by a continuous range of
allowed values.

The generalization for a global application indeed poses a challenge. However, for the tree
PFTs, researchers managed to find a set for classic DGVMS that represents the broad range of
environmental conditions possible. We believe that for herbaceous PFTs it will also be
possible to find a suitable set that will improve the representation of grasslands in current
DGVMs We hope to present this in a separate study in the near future. In the long term, additional model development including the step towards dynamic PFTs will further improve
the representation of different growth strategies.

line(s) 664/665: I do not really agree with this approach. The light extinction coefficient (as I
know it) is a constant that describes how much light a respective layer of leaves will absorb
and how much it will allow to transmit to the next lower leaf level. As such, it is a proxy
associated with leaf characteristics such as leaf thickness or SLA more than overall plant
stature. If anything, I'd deem LAI closer to stature than the light extinction coefficient, if you
do not have height available as state variable.

The reviewer is correct that the light extinction coefficient usually refers to the transmissivity
of a leaf layer. In theory, this is represented as one leaf with a given height and SLA per layer.
However, LPJmL and other classic DGVMs do not simulate different leaf layers but calculate
the light extinction of the entire vegetation layer of one PFT. Therefore, the model actually
calculates the light extinction of a stack of leaves. A larger stack of leaves will transmit less
light and therefore has a higher light extinction coefficient compared to a smaller stack of
leaves. Following this, the height of several leaf layers (or the vegetation layer) can be
interpreted as a function of SLA and the light extinction coefficient. As mentioned in the
discussion (L663-666) and previous work (Wirth et al., 2021) we think that this is a major
limitation and believe that adding plant height as a state variable would be an important model
development. As stated in our reply to the related comment in L218f we amended the model
description and refer to this in the discussion.

We added "We here deviate from the common interpretation of the light extinction
coefficient, which is usually defined as the light absorption of a layer of leaves. However, as
explained in Sect. 2.4.1, LPJmL represents the entire vegetation as a single layer and we
therefore define the light extinction coefficient not for a single leaf but a stack of leaves.
Taller plants likely produce more layers of leaves corresponding to a larger stack and a thicker
vegetation layer with a higher light extinction. However, thickness of the vegetation layer is
not explicitly represented in LPJmL and we represent the described differences by using
lower light extinction coefficients for small stature plants for which we assume a lower
thickness of the vegetation layer and higher light extinction coefficients for large stature
plants." in L832-837.

line(s) 674: In rangelands, mechanical stress through trampling would be another important
aspect to consider.

Similar to the missing inclusion of preferential grazing (comment in L294-300), this is related
to the representation of grazing.

We added "Additionaly, LPJmL-CSR does not consider mechanical stress caused by
trampling of animals and potential strategy dependent damage. Incorporating this may add
another dimension of stress to distinguish different PFTs." in L856f.

Minor editorial comments

We appreciate the thorough reading adopted all minor editorial comments below without
responding to each of those separately.

line(s) 10: "… a temperate grassland, a hot and a cold steppe…" => "… a temperate grassland
and a hot and a cold steppe…"

line(s) 13: at three grassland sites => at the three grassland sites line(s) 17: Our results show, that => delete comma line(s) 39: high carbon inputs => high carbon sequestration line(s) 61: (examples) => delete, seems to be a leftover note from manuscript writing. Or
alternatively replace with the examples you were thinking of…

line(s) 183: "recover slower" – "recover more slowly"

line(s) 184: "the SLA leaf longevity trade-off" – "the SLA v. leaf longevity trade-off"

line(s) 328: "While it remained similar…" – "However, it remained similar…"

line(s) 359 correct typo: resourCe line(s) 420 contribute – contributed line(s) 456: "we present results on above-ground biomass" – "we present results based on
above-ground biomass"

line(s) 490: "this allows to assess" – "this allows assessment of", or "this allows assessing"

line(s) 496: we only assess – we only assessed line(s) 533: "this can be explained with" – "this can be explained by"

line(s) 535: "and contributed to the litter layer" –"and increased the input to the litter layer".

line(s) 539: "In addition irrigation led to…" – "In addition, irrigation led to…"

line(s) 619: "…which constituted an additional investment." – Rephrase? "…and therefore, a
reduction of investment costs associated with N-fixation."

Hyphenation of two-word combinations that are used in the function of an adjective:

line(s) 69: "disturbance prone environments" – "disturbance-prone environments"

line(s) 73: "multi species communities" – "multi-species communities"

line(s) 181 "stress prone ecosystems" – "stress-prone ecosystems"

l- 203: "age dependent individual mortality" – "age-dependent individual mortality"

line(s) 231 "plant specific resource availability" – "plant-specific resource availability"

line(s) 249 "site specific conditions" – "site-specific conditions"

line(s) 296: bias adjusted data" – "bias-adjusted data"

line(s) 374, 375 "water saving strategy" – "water-saving strategy"

line(s) 397 resource limited – resource-limited line(s) 473: "S dominated community" – "S-dominated community"

line(s) 496: "neither water nor nutrient limited" – "neither water- nor nutrient-limited"

line(s) 542, line(s) 543, line(s) 579 "S dominated" – "S-dominated"

l- 580 "nutrient limited" – "nutrient-limited"

Review #2

This study is predicated on a novel way of quantifying CSR plant functional types (PFTs) for
species, and comparing these with frameworks including the leaf economics spectrum. This
forms the basis of the entire analysis, and so it is fundamental that the way the PFTs are
derived represents CSR theory and can be compared against the LES. There are a number of
basic problems with the approach used here, however.

We cordially thank the reviewer for taking the time to review our manuscript. When
comparing the reviewer's perception of our topic with our short summary, we gained the
impression that there has been a misunderstanding regarding the main focus of our study. In
turn, we believe that this led to a number of misconceptions, which we address in detail in our
responses. We are confident that our approach is solid and we hope that our changes to the
manuscript resolved the issues raised by the reviewer.

Using the trait specific leaf area (SLA) to represent both leaf economics and also within the
CSR calculations means that to two measures are very likely correlated, potentially leading to
a Type 1 statistical error in which the conclusions are accepted despite the statistical test not
being sufficient to assign a realistic probability.

Thank you for pointing this out, as it shows that our approach was not clearly described. We
agree that Type 1 statistical errors need to be avoided. However, we are not representing a
statistical but a functional relationship between SLA and leaf longevity here (L175-186 and
709-714). The connection between SLA and leaf longevity is well established following leaf economics (LES, Wright et al., 2004), but was so far not implemented as such in the LPJmL
model for grasslands.
In the original LPJmL model version, SLA was only used to compute the leaf area index
(LAI) from the internally computed leaf biomass. In order to represent the establishment of
different C-, S- or R-strategists, it is important to represent advantages and disadvantages of
the leaf structure in the model. Thinner leaves (high SLA) have a shorter longevity and while
they grow quickly to intercept light, they need to be replaced frequently. Neglecting the need
to replace thin leaves more frequently would lead to an advantage of high SLA values under
all circumstances, which is in contrast to ecological theory and observations (e.g., Díaz et al.,
2016; Reich, 2014; Wright et al., 2004). This trade-off had been implemented and applied to
tropical (Sakschewski et al., 2015) and European forests (Thonicke et al., 2020) before. The
implementation in this study provides a functional relationship of the SLA-LL relationship, as
part of the LES, and CSR theory through SLA in grasslands (section 2.3.2 and appendix A2).
This newly implemented functional relationship controls the productivity of the different
PFTs and the resulting shares of the C-, S- and R-PFTs. However, we do not compare the
LES to the C-, S- and R-PFT shares, which would be comparing inputs to outputs and would
certainly show a correlation.

We

[revised manuscript text omitted]

With regard to stress, the authors state that "According to CSR theory, the stress gradient
expresses the level of stress a species is exposed to in a certain habitat. It ranges from
unstressed to severely stressed, but does not distinguish individual stress categories (e.g.
temperature, water or nutrient)" thus "different strategies for water-resource use can be used
to distinguish C- and R-strategists (low stress tolerance) from S-strategists". Thus the traits
used here are specific to water stress, and the definition of stress recognised in CSR theory
(constrained metabolic efficiency and thus biomass production) is not cited nor considered.

The reviewer raises a valid point. Of course, stress is not restricted to water stress and other
traits that are related to (too high or too low) temperature or to nutrient stress could be used to
distinguish PFTs. In principle, the LPJmL model also considers stress arising from
temperature and nutrient availability in addition to water stress in its phenology and nitrogen
acquisition routines. However, the grassland steppe sites that we simulated in our study are
predominantly limited by water. Therefore, we decided to focus only on water stress in this
first application of LPJmL-CSR. This allows for a better understanding of the underlying
processes and the resulting pattern. In addition to the traits related to general stress tolerance,
we therefore only include traits related to water stress. However, we agree that the
implications of this simplification should be discussed.

We

- 980 • added "LPJmL represents the response of the vegetation to temperature, water and
- 981 nitrogen stress but disregards additional causes of stress such as other nutrient
- 982 deficiencies, salt, heavy metals, ozone or UV radiation." in L142ff.

- We replaced "According to CSR theory, the stress gradient expresses the level of stress a species is exposed to in a certain habitat." with "According to CSR theory, stress is defined as constrained metabolic efficiency limiting biomass production and can be caused by a variety of factors (Grime, 1977)." in L270f
- added "Since the LPJmL model only represents a subset of possible stress factor (Sect. 2.1), only stress arising from temperature and water as well as nitrogen availability can be considered. Within LPJmL-CSR, some traits are linked to a more general response to stress, while other are used to represent adaptation to specific stressors. Since the grassland steppe sites that we simulated in our study are predominantly limited by water, we decided to focus on water stress in this first application of LPJmL-CSR. This allows for a better understanding of the underlying processes and the resulting patterns." in L276-280.

Any stress (including water stress - but also factors such as nutrient stress or 'non-resource' stressors such as temperature) limits metabolic performance and thus growth and biomass production. Internal, inherent metabolic traits (such as photosynthetic capacity and dark respiration rate) or growth traits (such as relative growth rate) would have been acceptable to demonstrate limitation, but the authors provide no evidence that, for instance, that specific adaptations determining canopy-level conductance can represent the extent of general tolerance to stress.

We agree with the reviewer that limited metabolic performance is the result of various types of stress. Depending on the complexity of the model, responses to stress can be computed internally (reduced growth rate or reduced photosynthetic capacity) if these are implemented as dynamic functions in the model responding to, e.g., non-optimal temperatures or nutrient limitations. In LPJmL, SLA is important to determine photosynthetic activity and therefore affects the growth rate (L175ff). The leaf-to-root ratio affects the photosynthetic activity as well by controlling the investments into additional leaves. Therefore, we do not only consider traits related to tolerance to water stress but also traits related to a general tolerance to stress. We realize that the original version of the manuscript may not have been sufficiently clear in this regard and improved the description of the role of the different traits and how they represent tolerance to stress in section 2.4.1.

[revised manuscript text omitted]

We also agree with the reviewer that minimum canopy conductance and maximum
transpiration rate do only relate to water stress. However, we selected four traits associated
with the stress gradient to represent differences between the strategies. Two traits that are
associated with general tolerance to stress through their importance for plant growth and two
traits that are specific for water stress. As stated in a previous reply in L992f, we did not
select additional traits that specifically relate to other types of stress that are represented in the
model (temperature and nitrogen). With a better emphasis of our focus on water stress, the
selection of traits relevant for water dynamics is hopefully more comprehensible. As already
stated in our reply in L985-995, we agree that our description of the representation of
different types of stress and our reasoning to focus on water stress needs to be improved. We
included this in the changes we made regarding our reply in L1042-1013.

See reply L1020-1038.

Line 233: the authors state that "plant stature … can be used to distinguish C- and S-
strategists (low disturbance tolerance) from R-strategists". No: S-selected species can be
small (e.g. Salix herbacea) but some may become large over a long life-span (i.e.
Sequoiadendron giganteum). What matters is the C-selected species get large quickly, S-
selected species can become large eventually over a long life-span, and R-selected species
cannot. This is more a reflection of longevity and how rapidly plants achieve adult size.

We agree with the reviewer that S-strategists generally show a variety of statures as they
nicely illustrated with their examples. This is also clearly stated in Table 2 of Grime (1977) to
distinct species of the different strategies. While you can also find tall S-strategists in
grasslands (e.g. *Brachiaria brizantha* ), generally grassland plant species are of approximately

| 1114 | similar height (Gommers et al., 2013; Pontes et al., 2015). |
| 1115 | Still, the reviewer raises an important point. If one would only consider stature, an S-strategist |
| 1116 | might not be clearly distinguishable from a C- or an R-strategist and our explanation can be |
| 1117 | misinterpreted this way. However, we are aware of the importance of growth rate and |
| 1118 | longevity when distinguishing C-, S- and R-strategists. To account for this, the LPJmL model |
| 1119 | represents the fast-slow economics of the LES as explained in more detail in our reply in |
| 1120 | L896-977. Furthermore, we do not prescribe plant stature. Instead, we use a parameter that |
| 1121 | just represents the potential stature a strategist can attain. Depending on abiotic and biotic |
| 1122 | factors, the C- and S-strategist can become large but the R-strategist cannot. The C-strategists |
| 1123 | will grow rapidly if sufficient resources are available. The S-strategist will grow slowly but |
| 1124 | accumulate large amounts of biomass over a longer time or remain small if it is disturbed or |
| 1125 | outcompeted. |

| 1126 | We restructured and amended section 2.4.1 and hope that the distinction between the PFTs is |
| 1127 | clearer now. |

| 1128 | We added "At the core of the model is the representation of growth dynamics including the |
| 1129 | assimilation and allocation of new biomass through photosynthesis and turnover of senescent |
| 1130 | tissue. Each day, the GPP is calculated dependent on radiation, temperature, water and |
| 1131 | nitrogen limitations for each PFT. Subsequently, NPP is computed by subtracting growth and |
| 1132 | maintenance respiration from GPP. In a third step, the assimilated carbon is distributed |
| 1133 | between leaves and roots to approach the prescribed optimal leaf mass to root mass ratio. |
| 1134 | Finally, senescent leaf and root tissue is transferred to the litter layer." in L144-149. |

| 1135 | In the present study only juveniles were investigated, so using the leaf area index of a sapling |
| 1136 | is not going to represent the strategy in the main vegetative phase (seedling CSR strategies are |
| 1137 | known to be different from adult CSR strategies; Dayrell et al. (2018) Functional Ecology 32, |
| 1138 | 2730-2741). |

| 1139 | We agree with the reviewer that it is important to not only address CSR dynamics of juvenile |
| 1140 | plants. However, we would like to stress that we do not focus on juvenile plant dynamics. We |
| 1141 | assume that this misunderstanding originates in the description of establishment where |
| 1142 | saplings are established on bare ground. Still the model simulates an average individual that |
| 1143 | typically represents an adult plant (unless the entire plot has been re-established with new |
| 1144 | plants). |

| 1145 | We added "In LPJmL each PFT represent an entire population of adult plants using the |
| 1146 | concept of average individuals."in L136f. |

| 1147 | Also, CSR strategies are phenotypic characters (i.e. attributes of the individual plant that are |
| 1148 | subject to natural selection), but establishment rate (kest) [line 237] is not a character of an |
| 1149 | individual (the units of measurement are stated in Table 2 as the number of individuals per |
| 1150 | metre squared per day – a population measure), and so cannot elucidate the individual |
| 1151 | phenotype or adaptations of the species (i.e. the plant strategy or PFT). |

| 1152 | We agree that CSR strategies can be defined as a phenotypic characteristic of an individual |
| 1153 | and it may be counterintuitive that a measure that is not reported as being per individual but |
| 1154 | per meter squared can be used to represent a phenotypic characteristic. However, the |
| 1155 | establishment rate is just a parameter used within the model to calculate the actual |
| 1156 | establishment (appendix A3). This calculation considers several variables including the |
| 1157 | number of individuals and the resulting actual establishment can be reported as individuals |

per individual (a phenotypic characteristic). We see the point that this may be misunderstood.
Also, LPJmL-CSR does simulate trait plasticity as well evolutionary processes. Therefore,
phenotypic adaptation is not accounted for and adaptation only occurs at the community level
through changes in its composition.

We added "Each day new the number of average individuals of each PFT is increased if there
is bare ground area available. The bare-ground area is distributed between established PFTs
depending on their establishment rate $k_{est}$. The total amount of seedlings established is
calculated based on $k_{est}$, accounting for the bare ground area. Subsequently, the number of
average individuals is increased and the size of the individual specific carbon and nitrogen
stocks is adjusted. LPJmL-CSR does not consider trait plasticity or evolutionary processes
and therefore does not account for phenotypic adaptation This also means, that already
established and newly establish average individuals share the same traits." in L244-249.

In Figure 4, the red, green, blue (RGB) color scheme is used both to represent the extent of C,
S and R and the experimental treatments rainfed (red), irrigated (blue) and fertilised (green).

We agree that the coloring is not enhancing clarity and removed the colors from the axis and
labels of the ternary plots.

We removed the ambiguous coloring from the figure.

Additional changes:

• We added "the original" in L13.
• We deleted "availability of" in L40.
• We replaced "the time frames" with "periods" in L74.
• We added a comma between Sect. 2.3.2 and (Wright et al., 2004) in L182.
• We added "Names, descriptions and usage of the model parameters are based on the
model versions LPJmL4 (Schaphoff et al., 2018) and 5 (von Bloh et al., 2018)." in
L268f.
• We rephrased the caption of Table 2 as follows: "Parameter names, units, ranges,
associated CSR gradient(s) and the hierarchy of the parameters for the C-, S- and R-
PFTs."
• We added "[…]so that different CSR strategies can be represented by the extended set
of PFTs. The selected traits affect a variety of processes within the model and
differentiate the C- , S- and R-PFT along the stress and disturbance gradients. The
selected traits affect a variety of processes within the model and differentiate the C- ,
S- and R-PFT along the stress and disturbance gradients" in L336ff.
• We added "between" in L515.
• We replaced the reference to Fig. SI 7 with the reference to Fig. SI 8 in L520.
• We corrected the reference to the panels of Fig. SI 11 and 12 in L572, L575 and L588.
• We replaced "In LPJmL, herbaceous plants are represented as a number of average
individuals […]" with "In LPJmL, herbaceous plants are represented as average
individuals of a number of different PFTs […]" in L830.

References responses

[revised manuscript text omitted]

Onoda, Y., Wright, I.J., Evans, J.R., Hikosaka, K., Kitajima, K., Niinemets, Ü., Poorter, H., Tosens, T., Westoby, M., 2017. Physiological and structural tradeoffs underlying the leaf economics spectrum. New Phytologist 214, 1447–1463. https://doi.org/10.1111/nph.14496

Parsons, A.J., Newman, J.A., Penning, P.D., Harvey, A., Orr, R.J., 1994. Diet Preference of Sheep: Effects of Recent Diet, Physiological State and Species Abundance. Journal of Animal Ecology 63, 465–478. https://doi.org/10.2307/5563

Pfeiffer, M., Langan, L., Linstädter, A., Martens, C., Gaillard, C., Ruppert, J.C., Higgins, S.I., Mudongo, E.I., Scheiter, S., 2019. Grazing and aridity reduce perennial grass abundance in semi-arid rangelands – Insights from a trait-based dynamic vegetation model. Ecological Modelling 395, 11–22. https://doi.org/10.1016/j.ecolmodel.2018.12.013

Pontes, L. da S., Maire, V., Schellberg, J., Louault, F., 2015. Grass strategies and grassland community responses to environmental drivers: a review. Agron. Sustain. Dev. 35, 1297–1318. https://doi.org/10.1007/s13593-015-0314-1

Quillet, A., Peng, C., Garneau, M., 2010. Toward dynamic global vegetation models for simulating vegetation–climate interactions and feedbacks: recent developments, limitations, and future challenges. Environmental Reviews 18, 333–353. https://doi.org/10.1139/A10-016

Reich, P.B., 2014. The world-wide 'fast–slow' plant economics spectrum: a traits manifesto. Journal of Ecology 102, 275–301. https://doi.org/10.1111/1365-2745.12211

Rolinski, S., Müller, C., Heinke, J., Weindl, I., Biewald, A., Bodirsky, B.L., Bondeau, A., Boons-Prins, E.R., Bouwman, A.F., Leffelaar, P.A., te Roller, J.A., Schaphoff, S., Thonicke, K., 2018. Modeling vegetation and carbon dynamics of managed grasslands at the global scale with LPJmL 3.6. Geosci. Model Dev. 11, 429–451. https://doi.org/10.5194/gmd-11-429-2018

Rolinski, S., Wirth, S.B., Müller, C., Tietjen, B., 2021. Strategies for assessing grassland degradation, in: Joint XXIV International Grassland and XI International Rangeland Kenya 2021 Virtual Congress Oral Papers Proceedings. Presented at the Joint XXIV International Grassland and XI Rangeland Virtual Congress, Kenya Agricultural and Livestock Research Organisation, Nairobi, Kenia, pp. 383–387.

Sakschewski, B., Bloh, W. von, Boit, A., Rammig, A., Kattge, J., Poorter, L., Peñuelas, J., Thonicke, K., 2015. Leaf and stem economics spectra drive diversity of functional plant traits in a dynamic global vegetation model. Global Change Biology 21, 2711–2725. https://doi.org/10.1111/gcb.12870

Schaphoff, S., Bloh, W. von, Rammig, A., Thonicke, K., Biemans, H., Forkel, M., Gerten, D., Heinke, J., Jägermeyr, J., Knauer, J., Langerwisch, F., Lucht, W., Müller, C., Rolinski, S., Waha, K., 2018. LPJmL4 – a dynamic global vegetation model with managed land – Part 1: Model description. Geoscientific Model Development 11, 1343–1375. https://doi.org/10.5194/gmd-11-1343-2018

Scheiter, S., Langan, L., Higgins, S.I., 2013. Next-generation dynamic global vegetation models: learning from community ecology. New Phytol. 198, 957–969. https://doi.org/10.1111/nph.12210

Teng, Y., Zhan, J., Agyemang, F.B., Sun, Y., 2020. The effects of degradation on alpine grassland resilience: A study based on meta-analysis data. Global Ecology and Conservation 24, e01336. https://doi.org/10.1016/j.gecco.2020.e01336

Thonicke, K., Billing, M., von Bloh, W., Sakschewski, B., Niinemets, Ü., Peñuelas, J., Cornelissen, J.H.C., Onoda, Y., van Bodegom, P., Schaepman, M.E., Schneider, F.D., Walz, A., 2020. Simulating functional diversity of European natural forests along climatic gradients. Journal of Biogeography 47, 1069–1085. https://doi.org/10.1111/jbi.13809

von Bloh, W., Schaphoff, S., Müller, C., Rolinski, S., Waha, K., Zaehle, S., 2018. Implementing the Nitrogen cycle into the dynamic global vegetation, hydrology and crop growth model LPJmL (version 5). Geoscientific Model Development.

Wang, Q., Yang, Y., Liu, Y., Tong, L., Zhang, Q., Li, J., 2019. Assessing the Impacts of Drought on
Grassland Net Primary Production at the Global Scale. Sci Rep 9, 14041.
https://doi.org/10.1038/s41598-019-50584-4
Wirth, S.B., Taubert, F., Tietjen, B., Müller, C., Rolinski, S., 2021. Do details matter? Disentangling the
processes related to plant species interactions in two grassland models of different
complexity. Ecological Modelling 460, 109737.
https://doi.org/10.1016/j.ecolmodel.2021.109737
Wright, I.J., Reich, P.B., Westoby, M., Ackerly, D.D., Baruch, Z., Bongers, F., Cavender-Bares, J.,
Chapin, T., Cornelissen, J.H.C., Diemer, M., Flexas, J., Garnier, E., Groom, P.K., Gulias, J.,
Hikosaka, K., Lamont, B.B., Lee, T., Lee, W., Lusk, C., Midgley, J.J., Navas, M.-L., Niinemets, U.,
Oleksyn, J., Osada, N., Poorter, H., Poot, P., Prior, L., Pyankov, V.I., Roumet, C., Thomas, S.C.,
Tjoelker, M.G., Veneklaas, E.J., Villar, R., 2004. The worldwide leaf economics spectrum.
Nature 428, 821–827. https://doi.org/10.1038/nature02403

---

## Author Response (AR2)

Dear Prof. Dr. Stoy,

Thank you very much for guiding this review process. Below you find the answers to the reviewers' comments as well as some additional minor wording changes we implemented in the new manuscript version.

We also thank the reviewers for evaluating our manuscript.

Best regards on behalf of all co-authors,

Stephen Wirth

Reviewer 3
Functional diversity plays an important role in the resistance and resilience of an ecosystem towards the impacts of changing conditions and might be essential to maintaining the ecosystem functions of permanent grasslands under climate change. This study implemented a representation of functional diversity based on the CSR theory and the global spectrum of plant form and function into the LPJmL dynamic global vegetation model forming LPJmL-CSR. The paper is a relevant and important scientific contribution, and the authors have addressed the comments from the first round of revision very well. I would encourage the authors to address the following minor comments before accepting the paper for publication.

Abstract: spell out each acronym when it appears for the first time in the paper.

Thank you for pointing this out. We added the missing explanations in the abstract (L7ff).

Line 40: Some recent studies also reported the role of rainfall in C3 and C4 grass compositions, e.g., Xie, Qiaoyun, et al. "Satellite-observed shifts in C3/C4 abundance in Australian grasslands are associated with rainfall patterns." Remote Sensing of Environment 273 (2022): 112983.

Thank you for bringing this up. We added rainfall patterns as a driver of shifts in the C3/C4 composition in L41f.

Line 110: "cold steppe pasture in Inner Mongolia (China)" is contrary to "Mongolia" in Table 1. Mongolia is a different country, whilst Inner Mongolia is in China.

Thank you. We corrected this in Table 1

Figures 1 and 3: some x-axis labels overlap and are hard to read.

Thank you. We have updated the x-axis labels of figure 3.

The paper is too long and would benefit from more concise language.

We agree that the paper is long but believe that in order to reach scientists beyond the vegetation modelling community, the level of detail we provided is necessary. Nevertheless, we made some small adjustments listed within the additional changes section.

Reviewer 4

Grasses and herbaceous vegetation are typically represented only by C3 and C4 grass PFTs in DGVM, while trees are represented by multiple PFTs. The presented study aims to increase the level of grass diversity in LPJmL by including additional grass PFTs. Those additional PFTs represent CSR strategies. The updated model is calibrated by field data for three different study sites and showed higher data-model agreement than the original model version. Overall, I agree with the authors that the number of the grass PFTs should be increased in DGVMs and I like the idea of using the well-established CSR concept for such model improvements.

General comments.

Eight different traits are used to represent different CSR strategies and the traits and the hierarchy of those traits for different strategies are presented. Yet, I think it not clear for all of these traits what high and low values mean and why different values represent different strategies. I suggest the be more explicit to make clear why a low or high value of a trait represents CSR strategies. Ideally, this could be done in a schematic figure or a table.

Thank you for pointing out that this is still not sufficiently clear. We added an overview figure (Fig. 1) that shows the CSR triangle and connects the three strategies, the two gradients and the eight traits. Additionally, we explain what kind of strategy is represented by a low or a high trait value in section 2.4.1 (L296,304,308,312,325f, 332, 338 and 341f).

It is mentioned that only a low number of PFTs (ie three PFTs) was added. But if I understood the parametrization process correctly, trait values for each PFT were calibrated for three sites and two scenarios at each site, which means that 18 PFTs (3 CRS PFTs x 3 sites x 2 treatments) were parameterized. This parametrization shows how optimal PFTs should look like for these different scenarios and sites. However, I was wondering how to move from those site-specific parameters to larger scales and additional sites. Is it necessary to reparametrize for each site? Would it be possible to calibrate the models such that there are only 3 PFTs for CRS strategies per site or for C3 and C4 grasses, and then fit some competition parameters such that the introduction of grazing or other disturbances modifies the fractional covers of those strategies in agreement with observations? This may provide a more general parametrization of the model I suggest to be more explicit about such aspects in the discussion.

The reviewer raises an interesting point. For this study we indeed calibrated 18 PFTs in total and doing so for a large number of sites and scenarios is infeasible. However, LPJmL has successfully been calibrated at the global scale in the past using a genetic optimization algorithm and remote sensing data (Forkel et al., 2014, 2019) and we believe this to be a promising approach.

We added "While separate calibrations are feasible for a small number of sites and scenarios, for large scale or global assessments the lack of data and the computational requirement for the calibration make a site-specific calibration infeasible. However, using a more efficient calibration method and remote sensing data instead of on-site experiments can be used to derive a set of PFTs which are representative of the entire globe or at least climatic regions. For LPJmL, a genetic optimisation algorithm has been used to successfully calibrate the phenology (Forkel et al., 2014) and vegetation dynamics (Forkel et al., 2019) of natural ecosystems. Following this approach, we believe it is possible to identify C-, S- and R-PFTs for the tropical, temperate and polar regions ending up with nine PFTs in total." in L839-845.

One variable analyzed in the study is forage offtake. To me, this notation suggests that it is the biomass removed by mowing or grazing. This would then be a fixed value prescribed by the observations but not a modeled variable (unless grazing animals and their demand are simulated

dynamically). Yet, in the study offtake seems to be considered as variable simulated by the model. Does offtake represent the maximum amount of biomass that can be removed (representing maximum sustainable yield), or whether the demand of animals can be met? Please clarify.

Thank you for pointing this out. We defined forage offtake as the amount of biomass removed through mowing or grazing (L36f). In LPJmL mowing removes all biomass above a defined threshold while daily grazing removes biomass dependent on the feed demand of the livestock unit. Therefore, forage offtake is variable.

We added "[…] to determine forage offtake. While mowing removes all biomass above a threshold of 50 $gCm^2$, the forage offtake from daily grazing depends on the livestock units' feed demand (details in Appendix A5 and Rolinski et al., 2018)." in L157f to provide a brief overview and a reference to further literature.

Minor comments.
L4 delete "Especially"

Thank you. We accept the suggestion.

L54 consistent instead of uniform?

We made the replacement.

L62 traits instead of means?

Thank you. We accept the suggestion.

L 63 maybe "that influence the performance of different species"

Thank you. We accept the suggestion.

L 70 overall strategy of a community: reword? I'm not sure if a community has a strategy? Maybe say community composition?

In line with the CSR theory we changed "overall" to "average" (L71).

L 84 what is meant by "compare the CSR strategies"? Compare fractional cover of different strategies? Or classify species into one of these strategies?

To clarify, we changed "[…] a method to compare the CSR strategies of vascular plants […]" to "[…] a method to classify and compare the CSR strategies of different vascular plants […]" in L84.

L 99 not sure if it's necessary to highlight the soil module here (and not other important components such as ecophysiology or biogeochemical cycles)

Thank you. We accept the suggestion.

L103 management scenarios?

Thank you. We accept the suggestion.

L124 in the fractional cover of ...?

Thank you. We accept the suggestion.

L150, 151 mixture of "was" and "is", I suggest to check for consistency of tenses in ms

We reviewed the manuscript and changed the tense at several places (L152, L245, 250 and 254).

L170 delete "here"

Thank you. We accept the suggestion.

L170 dominant instead of advantageous?

The sentence the reviewer refers to is now in L174f. However, it already contained the verb "dominate" and did not make use of "advantageous". We believe the reviewer wanted to refer to L183 and replaced "advantageous" with "dominant" here.

L189 based on PFC?

Thank you. We accept the suggestion.

L243 share the same properties: so all individuals are identical? Does recruitment imply heterogeneity of the population and some age and height structure?

No, as explained in L138ff, LPJmL uses an average individual approach which does not include heterogeneity of the population. To make this more explicit we changed "[…] that share the same properties […]" to "assuming a homogeneous population (i.e. individuals of the same PFT share the same properties)" in L247.

L250 prohibit infinite growth: but if space and resources are limited, plants shouldn't grow infinite. Why is it necessary to constrain growth? Isn't there saturation of growth as plant get taller, competition gets more intense or all cells ore occupied?

Thank you, we realized that the term "growth" is misleading in this context as it may easily be interpreted as plant growth. However, what we mean is that we limited the number of individuals per grid cell. We replaced "growth" with "increase" to avoid such a misconception (L256).

L271-272 if there is a stress gradient, then there is a large number of stress categories, why is it important that stress categories are not distinguished?

The stress gradient expresses the level of stress and is therefore not a gradient across stressors but across stress intensities caused by one or several stressors. We replaced "level" with "intensity" in L277 and rephrased "[…] but does not distinguish individual stress categories" to "[…] and can include the combined impacts of several stressors." in L278 to increase clarity.

L277 a more general response to stress: this is vague, what is meant by a more general response?

We rephrased "[...] more general response to stress, […]" to "[…] general response to stress independent of the stressor, […]" in L284 to increase clarity.

L327 seedlings additional to what?

We mean additional to the already established plants. However, this is already clearly described by the term establishment and the word additional may lead to the interpretation that the model represent multiple different establishment processes. To avoid the misconception we deleted additional (L334).

L377 but ensured

Thank you. We accept the suggestion.

L390 made it impossible

Thank you. We accept the suggestion.

Fig 1 forage offtake instead of supply in panel a? Labels in c, d, i overlap (also in other figures). How were bias, phase and variance calculated? This should be mentioned in the caption or the methods.

We have updated the labels of figure 2 and added the equations for the components of the mean square error in the appendix (L996-1003) to which we refer in L393.

L368, 477 and elsewhere, the words "shifted" and "change". These words suggest to me that traits change during the simulations eg after switching on grazing. But these differences in trait values are the outcome of the calibration process and are therefore equilibrium or optimal trait values, right? I suggest to reword for example by saying the algorithm selected for less explorative strategies or that traits were lower/higher for grazing treatments.

We agree with the reviewer that this might be confusing to some reader and have reworded the ambiguous phrases in section 3.2 and 4.2.2. (L475ff, 479, 481f, 492f, 494 and 763)

Fig 2 replace abiotic and biotic by disturbance and stress

Thank you. We have updated the labels of figure 2.

L480, 481 reword secondary and primary role of traits?

We rephrased the respective sentences removing the terms primary and secondary role (L486f).

Fig 3 forage offtake instead of supply in a? Panel b) means that the same amount of biomass is removed each month in each year, do I interpret this correctly? Wy are the red an black line not identical in 2005 as the lines in e and f? Years in d and f overlap and can't be read.

Thank you for pointing this out. We have updated the label and x axis labels of figure 3. Indeed, in panel b) the values are the same for each timestep. As we explained in L549, the animal forage demand is always met in both scenarios. In panels d) to f) the colors show the different scenarios. In panel d) the red and black line represent the different fertilization scenarios which are already different before 2005. In panel e), black and blue distinguish the grazed and the ungrazed scenario which are also already different in 2005. Only at the cold steppe (f), both scenarios only start to differ in 2005. To make this clear, we added "For the temperate grassland and the hot steppe, the different management of the unfertilised and fertilised and ungrazed and grazed scenario led to differences in soil carbon before the first year shown in Fig. 4." in L519f.

L680 equilibrium state instead of finality?

Thank you. We accept the suggestion.

L871 delete "existence of a"

Thank you. We accept the suggestion.

**Additional changes**
We replaced ecosystem functions with ecosystem services in L1, 4, 24, 26 and 36 and with ecosystem functioning and service provision in L34, 86 and 889.

Deleted "of permanent grasslands" in L26.

Added (Guuroh et al., 2018) in L35.

Added plant in L39.

Added active in L48.

Replaced (Milchunas and Lauenroth, 1993; Oesterheld and Loreti, 1999; Semmartin and Oesterheld, 1996) with (Ruppert et al., 2015) in L51.

Added the reference to (Buzhdygan et al., 2020) in L67

Removed pasture in L112.

Rephrased "[…]between large investments in reproduction but a small stature (R) and small investment […]" to "between plant species with large investments in reproduction but a small stature (R) from plant species with small investments" in L120f.

Added a reference to section 2.3 in L124.

Added (Scheiter et al., 2023) in Table 1.

Added "of a given grassland area" in L212f.

Replaced leave with leaf in L214.

Added resource in L217.

Added and "[…] age-related mortality is common in natural grasslands (Zimmermann et al., 2010),[…]"in L255f.

Replaced "such as embolism." with "such as drought or fire (Zimmermann et al., 2010)" in L259f.

Replaced "inherited" with "retained" in L265.

Changed functional plant traits to plant functional traits in L273f.

Replaced "that can be caused" by "and can be caused" in L276.

Replaced stress categories with stressors in L278.

Replaced leaf economic spectrum with LES in L280.

Replaced "that we simulated in our study" with "simulated by us" in L285.

Deleted "in this first application of LPJmL-CSR" in L285.

Added "standing" to the caption of figure 5.

Replaced "contained" with "also included" in L651.

Added (Scheiter et al., 2023) in L657 and L661.

Replaced "The R-PFT was better suited to withstand the grazing and out-competed the C-PFT in the grazed scenario due to its higher ability to deal with disturbances." with "The R-PFT was more tolerant towards grazing disturbances and gained dominance in the grazed scenario, replacing the C-PFT which had a lower ability to deal with disturbance." In L662f.

Replaced "[…] would increase the competition for light and space […]" with "would increase the competition for light due to self-shading effects (Zimmermann et al., 2010)[…]" in L 671.

Replaced "determined" with "led to biased" in L761.

Added "plant" in L889.

Replaced "including intermediate habitats" with "considering habitats with intermediate environmental conditions" in L892f.

References

Buzhdygan, O.Y., Meyer, S.T., Weisser, W.W., Eisenhauer, N., Ebeling, A., Borrett, S.R., Buchmann, N., Cortois, R., De Deyn, G.B., de Kroon, H., Gleixner, G., Hertzog, L.R., Hines, J., Lange, M., Mommer, L., Ravenek, J., Scherber, C., Scherer-Lorenzen, M., Scheu, S., Schmid, B., Steinauer, K., Strecker, T., Tietjen, B., Vogel, A., Weigelt, A., Petermann, J.S., 2020. Biodiversity increases multitrophic energy use efficiency, flow and storage in grasslands. Nat Ecol Evol 4, 393–405. https://doi.org/10.1038/s41559-020-1123-8

Forkel, M., Carvalhais, N., Schaphoff, S., v. Bloh, W., Migliavacca, M., Thurner, M., Thonicke, K., 2014. Identifying environmental controls on vegetation greenness phenology through model–data integration. Biogeosciences 11, 7025–7050. https://doi.org/10.5194/bg-11-7025-2014

Forkel, M., Drüke, M., Thurner, M., Dorigo, W., Schaphoff, S., Thonicke, K., von Bloh, W., Carvalhais, N., 2019. Constraining modelled global vegetation dynamics and carbon turnover using multiple satellite observations. Sci Rep 9, 18757. https://doi.org/10.1038/s41598-019-55187-7

Guuroh, R.T., Ruppert, J.C., Ferner, J., Čanak, K., Schmidtlein, S., Linstädter, A., 2018. Drivers of forage provision and erosion control in West African savannas—A macroecological perspective. Agriculture, Ecosystems & Environment 251, 257–267. https://doi.org/10.1016/j.agee.2017.09.017

Milchunas, D.G., Lauenroth, W.K., 1993. Quantitative Effects of Grazing on Vegetation and Soils Over a Global Range of Environments. Ecological Monographs 63, 327–366. https://doi.org/10.2307/2937150

Oesterheld, M., Loreti, J., 1999. Grazing, fire, and climate effects on primary produc- tivity of grasslands and savannas, in: Walker, L. (Ed.), Ecosystems of Disturbed Ground. Elsevier Science, Oxford, pp. 287–306.

Rolinski, S., Müller, C., Heinke, J., Weindl, I., Biewald, A., Bodirsky, B.L., Bondeau, A., Boons-Prins, E.R., Bouwman, A.F., Leffelaar, P.A., te Roller, J.A., Schaphoff, S., Thonicke, K., 2018. Modeling vegetation and carbon dynamics of managed grasslands at the global scale with LPJmL 3.6. Geosci. Model Dev. 11, 429–451. https://doi.org/10.5194/gmd-11-429-2018

Ruppert, J.C., Harmoney, K., Henkin, Z., Snyman, H.A., Sternberg, M., Willms, W., Linstädter, A., 2015. Quantifying drylands' drought resistance and recovery: the importance of drought intensity, dominant life history and grazing regime. Global Change Biology 21, 1258–1270. https://doi.org/10.1111/gcb.12777

Scheiter, S., Pfeiffer, M., Behn, K., Ayisi, K.K., Siebert, F., Linstädter, A., 2023. Managing southern African rangeland systems in the face of drought - a synthesis of observation, experimentation, and modelling for policy and decision support, in: von Maltitz, G.P., Midgley, G.F., Veitch, J., Brümmer, C., Rötter, R.P., Viehberg, F.A., Veste, M. (Eds.), Sustainability of Southern African Ecosystems under Global Change, Ecological Studies. Springer.

Semmartin, M., Oesterheld, M., 1996. Effect of Grazing Pattern on Primary Productivity. Oikos 75, 431–436. https://doi.org/10.2307/3545883

Zimmermann, J., Higgins, S.I., Grimm, V., Hoffmann, J., Linstädter, A., 2010. Grass mortality in semi-arid savanna: The role of fire, competition and self-shading. Perspectives in Plant Ecology, Evolution and Systematics 12, 1–8. https://doi.org/10.1016/j.ppees.2009.09.003